# BLITZRANK: Principled Zero-shot Ranking Agents with Tournament Graphs

Sheshansh Agrawal [* 1]   Thien Hang Nguyen [* 1]   Douwe Kiela [1]

## Abstract

Selecting the top $m$ from $n$ items via expensive $k$-wise comparisons is central to settings ranging from LLM-based document reranking to crowdsourced evaluation and tournament design. Existing methods either rely on heuristics that discard comparison information, or exploit it at prohibitive cost. We introduce a *tournament graph* framework that provides a principled foundation for $k$-wise ranking. Our key observation is that each $k$-item comparison reveals an induced tournament of $\binom{k}{2}$ pairwise preferences; aggregating these into a global preference graph and computing its transitive closure yields many additional orderings without further oracle calls. We formalize when the current top-$m$ output is *certifiably determined* and design a greedy query schedule that maximizes information gain towards identifying the top-$m$ items. The framework also gracefully handles non-transitive preferences – cycles induced by real-world oracles – by collapsing them into equivalence classes that yield principled *tiered rankings*. Applied to LLM reranking across 14 benchmarks and 5 models, BLITZRANK achieves Pareto dominance over existing approaches: matching or exceeding accuracy while requiring 25–40% fewer tokens than comparable methods; against pairwise reranking, it achieves near-identical quality with $7\times$ fewer tokens. Code available at https://github.com/ContextualAI/BlitzRank.

## 1. Introduction

Consider the classic *25 horses' race* puzzle: *given 25 horses where 5 can race at a time, what is the minimum number of races needed to identify the 3 fastest?*

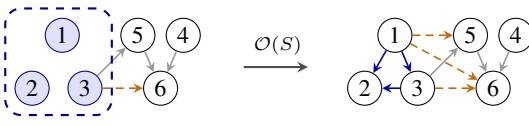

Figure 1. A $k$-wise oracle query on $n{=}6$ candidates. **Left:** A query set $S$ of $k{=}3$ candidates (shaded) is selected. **Right:** The oracle returns a tournament on $S$, revealing $\binom{3}{2}{=}3$ new edges (blue). Combined with prior edges (gray), additional preferences are inferred transitively (orange dashed).

The answer – 7 races – is well known (Zhou & Jiu, 2008), but the reasoning reveals a deeper principle. A naive tournament bracket uses each race only to identify a winner. The optimal strategy extracts *all* the information each race provides: a race among 5 horses reveals a complete ordering – 10 pairwise preferences – and accumulating these transitively certifies the top 3 with far fewer races than a bracket requires.

This paper develops a principled framework for this class of problems: *top-$m$ selection from $n$ items via $k$-wise comparison queries*. Each query to an oracle reveals the induced tournament on up to $k$ items – $\binom{k}{2}$ pairwise preferences – and the goal is to identify the top-$m$ items using as few queries as possible. The problem arises naturally whenever comparisons are expensive: LLM-based document reranking, crowdsourced preference judgments, human-in-the-loop evaluation, and tournament design. In all these settings, each query incurs cost in time, money, or human attention, making query efficiency paramount.

Existing methods leave information on the table. Classical sorting algorithms adapted for $k$-wise comparisons – heapsort (Zhuang et al., 2024b; Qin et al., 2024), tournament brackets (Chen et al., 2025), sliding windows (Sun et al., 2023) – typically focus on selection or identifying a winner at each stage, underutilizing the $\binom{k}{2}$ pairwise relationships revealed. To our knowledge, no existing approach provides a framework for accumulating comparison outcomes and a criterion to certify the top-$m$.

We propose a *tournament graph* framework and an algorithm BLITZRANK[1] that addresses these limitations. By

*Equal contribution [1]Contextual AI. Correspondence to: Sheshansh Agrawal and Thien Hang Nguyen <blitzrank.icml@gmail.com>.

*Proceedings of the 43rd International Conference on Machine Learning*, Seoul, South Korea. PMLR 306, 2026. Copyright 2026 by the author(s).

[1]The name BLITZRANK evokes blitz chess, where rapid tournament play efficiently determines rankings.

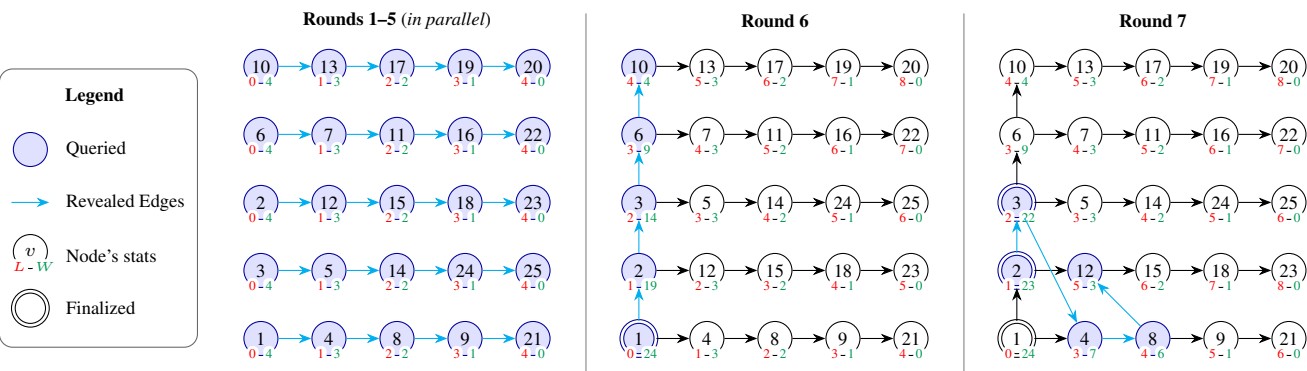

*Figure 2.* Illustration of BLITZRANK (Algorithm 1) achieving the optimal 7 queries on the classic *25 horses puzzle*, where $(n, k, m) = (25, 5, 3)$. Each node shows the horse ID with $L(u) := |R_G^-(u)|$ (cf. (1)) bottom left and $W(u) := |R_G^+(u)|$ (cf. (2)) at bottom right. Blue nodes indicate horses queried in that round. In this transitive instance, $\kappa_G(u) = L(u) + W(u)$, (cf. (3)) and double circles indicate resolved horses (where $\kappa_G(u) = 24$).

Note: For reproducibility, the initial ordering was generated with Python's `random.shuffle` on $[1, 2, \ldots, 25]$ with `seed=42`. The first five rounds are grouped as shown.

aggregating these local tournaments into a global preference graph, the transitive closure yields additional orderings without further queries. We formalize when a node is *resolved* – its relationship to all $n-1$ others determined – and design an algorithm that terminates once the current top-$m$ are certified. Applied to the 25 horses puzzle, the algorithm discovers the optimal 7-race strategy without being given any problem-specific knowledge (Figure 2).

The framework also addresses a phenomenon that prior work largely overlooks: *non-transitive preferences*. Real-world oracles – LLMs, crowdworkers, domain experts – sometimes produce cyclic judgments: $A \succ B \succ C \succ A$. Rather than treating cycles as noise, we treat them as structure indicating that the oracle cannot consistently separate those items. Our framework captures this via *strongly connected components* (SCCs) of the preference graph, which collapse into equivalence classes to yield a DAG of "relevance tiers." The *same algorithm* handles both settings – returning a total ordering when preferences are consistent, and a principled tiered ranking otherwise.

**Contributions.**

1. **A unifying theoretical framework.** We formalize top-$m$ selection via $k$-wise comparison oracles using tournament graphs (§2). The framework captures how transitive closure amplifies each query's information yield, defines when items are *resolved*, and unifies transitive and non-transitive preferences.

2. **A provably correct algorithm.** We present BLITZRANK, a greedy algorithm that schedules queries among minimally-resolved SCCs to guarantee progress (§2.4). We prove correctness and termination in both transitive and non-transitive settings.

3. **Empirical validation.** Across 14 benchmarks and 5 LLM oracles, our approach achieves Pareto dominance:

matching or exceeding baseline accuracy while requiring 25–40% fewer tokens than methods with comparable structure, and $7\times$ fewer than pairwise reranking at near-identical quality (§4). Convergence is predictable, and our analysis confirms that cycles capture genuinely ambiguous documents rather than noise.

**Overview.** Section 2 presents the tournament graph framework, Section 3 discusses related work, and Section 4 evaluates BLITZRANK empirically on document reranking benchmarks. Full proofs, theoretical analysis, and additional experiments appear in the appendices.

## 2. Tournament Graph Framework

### 2.1. Problem Setup

Let $V$ be a set of $n$ items and $G^* = (V, E^*)$ be a fixed but unknown tournament, representing the oracle's latent preferences: for every pair $u \neq v$, exactly one of $(u, v)$ or $(v, u)$ belongs to $E^*$. We access $G^*$ through a $k$-wise comparison oracle $O_{G^*}$: given any queried set $S \subseteq V$ with $|S| \leq k$, returns all directed edges of $G^*$ induced by $S$.

Our ranking target is reachability-based. For a directed graph $G$, write $u \rightsquigarrow_G v$ if there is a directed path from $u$ to $v$, and define the *in-reach* of $v$ by

$$R_G^-(v) := \{u \in V \setminus \{v\} : u \rightsquigarrow_G v\}. \tag{1}$$

The task is to identify $m$ vertices with smallest in-reach using as few oracle queries as possible. When $G^*$ is transitive, in-reach coincides with in-degree, so this reduces to the usual loss-count ordering; in general, the reachability view extends naturally to cyclic preferences. Formal definitions and discussion appear in Appendix F.

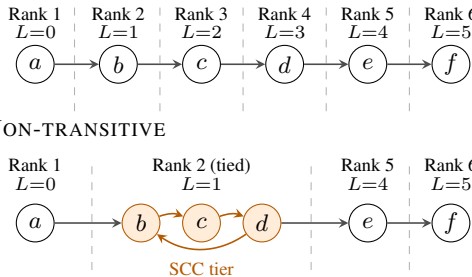

TRANSITIVE

NON-TRANSITIVE

*Figure 3.* Tournament graphs with $n=6$ candidates. **Top (transitive):** Consistent preferences yield a total ordering; each node has a unique rank determined by $L(u) := |R_G^-(u)|$, the number of vertices that reach it. **Bottom (non-transitive):** A cycle $b \succ c \succ d \succ b$ forms a strongly connected component (orange). Nodes in the SCC share the same tier since no consistent ordering exists among them, but the partial order $a \succ \{b, c, d\} \succ e \succ f$ is still recovered.

## 2.2. Revealed Graph and Transitive Inference

We maintain a *revealed graph* $G = (V, E)$ that accumulates edges returned by the oracle. Initially $E = \emptyset$; after querying a set $S$, we add the revealed edges $O_{G^*}(S)$ to $E$. By construction, $G$ is always a subgraph of $G^*$.

The key point is that $G$ contains more information than its explicit edges. If $u \leadsto_G v$, then the transitive preference $u \succ v$ is already certified in the unknown tournament, even if the edge $(u, v)$ was never queried directly. In this sense, each query is amplified by reachability: a newly revealed edge can combine with existing paths to certify many additional pairwise relations.

For a vertex $v$, we also consider its out-reach

$$R_G^+(v) := \{u \in V \setminus \{v\} : v \leadsto_G u\}. \quad (2)$$

The vertices whose relation to $v$ is already determined form the known-relationship set

$$K_G(v) := R_G^-(v) \cup R_G^+(v), \qquad \kappa_G(v) := |K_G(v)|. \quad (3)$$

When $\kappa_G(v) = n - 1$, the comparison between $v$ and every other vertex is already implied by the revealed graph; we call such a vertex *resolved*. At that point, its discovered in-reach is already its true in-reach in $G^*$, so further queries cannot change its rank. We use this as a practical stopping rule for algorithm design: we terminate once the current top-$m$ vertices by discovered in-reach are resolved. The full finalization framework is presented in Appendix G.

## 2.3. Non-Transitive Preferences

When the oracle produces cycles, a global total order may not exist. Our framework captures these intransitivities

via *strongly connected components* (SCCs). Vertices in the same SCC reach one another in both directions, so they behave as a tied tier under reachability-based ranking. Collapsing each SCC to a supernode gives the *condensation* $[G]$, which is always a DAG; for tournaments, the condensation is in fact a transitive tournament. This induces a total order on SCC tiers even when the original graph contains cycles.

Accordingly, our output in the non-transitive case is a *tiered ranking*: SCCs are ordered from best to worst, while vertices inside the same SCC are tied (see Figure 3). When all SCCs are singletons, this reduces to the transitive case and yields a total ordering. For top-$m$ selection, if the boundary tier is larger than the remaining quota, any subset of the required size from that tier is a valid output.[2]

At a high level, the algorithm operates on the discovered SCC structure while maintaining the same vertex-level resolution criterion from the previous subsection. Part II of the appendix develops the full theory, including why discovered SCCs refine the true ones and why reasoning on the condensation is enough to certify correctness.

## 2.4. Algorithm: BLITZRANK

BLITZRANK (Algorithm 1) realizes the tournament graph framework. At each iteration, the algorithm recomputes the discovered in-reach $|R_G^-(v)|$ and known-relationship count $\kappa_G(v)$ for every vertex, forms the current top-$m$ set $T$ by smallest discovered in-reach, and stops once every vertex in $T$ is resolved. If not, it computes the condensation $[G]$ and greedily queries representatives from the unresolved SCCs with the smallest in-reaches in the condensation graph.

**Why greedy scheduling works.** The greedy step operates on SCCs ordered by ascending in-reach in the condensation. The progress argument has two parts. First, at any non-terminal iteration, the earliest unresolved SCCs are tied in condensation in-reach; this follows from the general structural lemma on tied candidates after the finalization threshold (Lemma 22), applied to $[G]$, with the fact that these tied SCCs indeed contain unresolved vertices (Lemma 59). Second, once two SCCs are tied, the forced-tie property (Lemma 46) implies that there is no edge between them in the condensation and thus no revealed edge between any of their members in $G$. Querying representatives from such SCCs therefore uncovers at least one new edge and progress is guaranteed at every round. This is the key behind the termination proof for BLITZRANK; see Theorem 60 in Appendix J.

**Parallelization.** When $k$ is small relative to $n$, multiple disjoint groups can be queried in parallel. The algorithm

---

[2]If a total order is required, ties inside a tier can be broken by a secondary signal such as the original retrieval score.

**Algorithm 1** BLITZRANK

1: **Input:** vertex set $V$ with $|V| = n$, oracle $O_{G^*}$, maximum query size $k$, target count $m$
2: **Output:** a valid top-$m$ output for the underlying tournament $G^*$
3: Initialize $E \leftarrow \emptyset, G \leftarrow (V, E)$
4: **loop**
5:    // *Compute vertex metrics*
6:    **for** each $u \in V$ **do**
7:       Compute $|R_G^-(u)|$ and $\kappa_G(u)$
8:    **end for**
9:    // *Identify resolved vertices and current top-$m$ candidates*
10:    $F \leftarrow \{u \in V : \kappa_G(u) = n - 1\}$ {resolved vertices}
11:    $T \leftarrow$ the $m$ vertices with smallest $|R_G^-(\cdot)|$
12:    // *Termination condition: current top candidates are resolved*
13:    **if** $T \subseteq F$ **then**
14:       **return** $T$ sorted by ascending $|R_G^-(\cdot)|$
15:    **end if**
16:    // *Greedy Schedule*
17:    Compute condensation $[G]$ of the revealed graph $G$
18:    $\mathcal{C} \leftarrow$ SCCs containing $\geq 1$ vertex with $\kappa_G(u) < n - 1$, ordered by ascending in-reach in $[G]$
19:    $Q \leftarrow \{\text{rep}(C) : C \in \mathcal{C}[1 : k']\}$ where $k' = \min(k, |\mathcal{C}|)$ {one representative per SCC}
20:    // *Query and update*
21:    $E \leftarrow E \cup O_{G^*}(Q), \quad G \leftarrow (V, E)$
22: **end loop**

naturally supports this because unresolved SCCs are handled independently until they become connected by newly revealed edges.

**Variable window size.** The framework places no constraint on $k$ being fixed across rounds. In each iteration, $k$ can be chosen adaptively based on the candidate documents' lengths, the oracle's context window, or model-specific capabilities. This is a practical advantage for heterogeneous document collections where a fixed window size either truncates long documents or underutilizes the context on short ones.

## 2.5. Proof Sketch

BLITZRANK stops when the current top-$m$ vertices by discovered in-reach are all resolved. We sketch why this stopping rule yields a correct output and why it is always reached; the full proofs appear in Appendix J.

**Output correctness.** The correctness argument rests on the following property: if $u$ is resolved (i.e. $\kappa_G(u) = n-1$), every $w \neq u$ is either in $R_G^-(u)$ or $R_G^+(u)$. We show in

Lemma 54 that whenever $u$ is resolved and has discovered in-reach no larger than that of some vertex $v$, then $u$ truly ranks at least as well as $v$ in the unknown tournament $G^*$ – that is, $|R_{G^*}^-(u)| \leq |R_{G^*}^-(v)|$.

At termination, every vertex $u$ in the returned set $T$ is resolved and has discovered in-reach at most that of every vertex outside $T$. Applying the lemma to each such pair certifies both the internal ordering within $T$ and its rank dominance over all remaining vertices (Corollary 56, Theorem 57).

**Termination.** Every non-terminal round discovers at least one new edge, so the algorithm halts in at most $\binom{n}{2}$ rounds. The argument works on the condensation $[G]$, which is always a DAG. In any DAG, the $i$-th vertex by in-reach has in-reach at most $i-1$; the longest prefix where this bound is tight defines the *finalization threshold* $\widetilde{m}$. Non-terminality forces $\widetilde{m} \leq n'-2$, and the two positions just past the threshold must share the same in-reach (otherwise finalization would extend further). These "tied" SCCs have no revealed edge between them, so the greedy schedule – which prioritizes them (§2.4) – is guaranteed to discover a new edge (Theorem 60).

## 2.6. Tie-breaking and Query Complexity

Algorithm 1 leaves two choices open: the ordering among SCCs with equal condensation in-reach, and the representative selected from each SCC. The correctness and termination guarantees hold for any choice, but query efficiency depends on them. A natural heuristic for both is to *prioritize the least-resolved entity*. Among SCCs with equal condensation in-reach, we query those with lower out-reach (highest positional uncertainty) and select the representative vertex with the smallest $\kappa_G$ (fewest established relationships), concentrating effort where information gain is highest. We denote this fully specified schedule BLITZRANK† (defined in Appendix K) and use it in all experiments (Section 4).

The termination proof gives a worst-case bound of $\binom{n}{2}$ queries, but in practice BLITZRANK terminates much sooner: Figure 2 shows it achieves the optimal 7 queries on the 25-horses instance. For top-1 selection ($m = 1$), each query eliminates at least $k - 1$ candidates, giving at most $\lceil (n - 1)/(k - 1) \rceil$ queries (Proposition 62). For general $m$, we conjecture that BLITZRANK† achieves $O((n-1)/(k-1) + (m-1)/(k-1) \cdot \log_k m)$, decomposing into a *candidate reduction* term and a *frontier refinement* term (Conjecture 63). Empirically, observed query counts remain within a factor of 1.25 of this form across $n$ from 100 to 800 and $k$ from 5 to 50 (Figure 10), suggesting the bound is tight up to lower-order terms. A formal proof for $m > 1$ remains open; see Appendix K for details.

## 3. Related Work

**LLM-based document reranking.** LLM reranking methods fall into three paradigms. *Pointwise* methods score documents independently, enabling parallelism but discarding relative information; zero-shot LLM approaches include query likelihood (Sachan et al., 2022) and relevance classification (Zhuang et al., 2024a), while trained cross-encoder models (Contextual AI, 2025) dominate in efficiency. *Pairwise* methods recover relative preferences: Qin et al. (2024) introduced Pairwise Ranking Prompting with heapsort aggregation at $O(n \log n)$ comparisons, noting that "pairwise comparisons are not guaranteed to be transitive." *Listwise* methods compare multiple documents per call: RankGPT (Sun et al., 2023) and LRL (Ma et al., 2023) concurrently established the sliding-window paradigm, processing windows of 20 documents with stride 10, while subsequent work distilled this into open-source models (Pradeep et al., 2023a;b). AcuRank (Yoon et al., 2025) maintains Gaussian distributions over document relevance and performs Bayesian updates via TrueSkill, selectively reranking uncertain documents until confidence criteria are met. All of these methods use a fixed window size; our framework accommodates variable $k$ per round (§2.4), adapting to heterogeneous document lengths and model-specific context limits.

The *setwise* approach (Zhuang et al., 2024b) bridges pairwise and listwise by prompting the LLM to select the most relevant from $k$ candidates, using this primitive within heapsort. While more efficient than pairwise, setwise extracts only the *winner*, discarding the remaining $\binom{k}{2} - (k-1)$ pairwise relationships. Our framework differs by extracting the *complete tournament* from each $k$-wise comparison and accumulating edges where transitive closure amplifies each query's information yield.

**Handling inconsistency in LLM rankings.** LLM judgments frequently violate transitivity. Zeng et al. (2024) address this via LLM-RankFusion, measuring inconsistency patterns and resolving them through rank aggregation. Tang et al. (2024) show that shuffling and Kemeny-optimal aggregation mitigate positional biases; similarly, TourRank (Chen et al., 2025) runs multiple parallel tournaments with different random seeds and aggregates scores. Most related to our approach, ELSPR (Yu et al., 2025) uses SCC analysis to quantify non-transitivity via structural entropy, but applies it to filter training examples for evaluator LLMs rather than for query-efficient selection. REALM (Wang et al., 2025) takes a probabilistic approach, modeling relevance as Gaussian distributions updated via Bayesian inference. In contrast, our framework is deterministic and treats cycles as structure (tiered rankings) rather than a defect to be aggregated away.

**Tournament theory and multi-wise comparisons.** Our framework builds on tournament graph theory (Brandt et al., 2016; Laslier, 1997; Landau, 1953). The query complexity of tournament solutions under a pairwise oracle was studied by Dey (2017), who proved that finding the Copeland set, Slater set, and most other solutions requires $\Omega(n^2)$ queries in the worst case. When the top cycle has bounded size $c$, all solutions can be found in $O(nc + n \log n / \log(1 - 1/c))$ queries; however, this identifies only the first SCC (the top cycle) and does not recover the full condensation ordering over all components.

On the multi-wise comparison side, Ren et al. (2021) establish that full-ranking feedback from $k$-wise comparisons improves sample complexity by a factor of $k \log k$ over winner feedback, and Saha & Gopalan (2019) show similar gains under the Plackett–Luce model. These results, along with near-optimal top-$m$ algorithms under parametric models (Jang et al., 2017; Chen et al., 2018), assume stochastic comparison models and target total rankings rather than structural objectives like SCC recovery.

Our setting differs from all of the above in three respects: (i) we assume a deterministic tournament rather than a stochastic comparison model, (ii) we use a $k$-wise oracle ($k \geq 2$) that reveals complete induced tournaments rather than restricting to pairwise queries or winner-only feedback, and (iii) we target the SCC decomposition ordering – a tiered ranking that declares cyclic items tied – rather than a total ranking or a single tournament solution set; no existing algorithm addresses this combination. A more detailed discussion appears in Appendix F.3.

## 4. Experiments

We evaluate BLITZRANK (Algorithm 1) on standard document reranking benchmarks within the established retrieve-then-rerank pipeline, using an LLM as the zero-shot comparison oracle. Our experiments address two primary questions: (1) Does extracting complete tournaments and propagating information via transitive closure translate to practical efficiency gains? (2) Do these efficiency gains compromise ranking quality, or can our framework match existing approaches with fewer oracle calls? We compare against baseline reranking strategies across 14 datasets and 5 LLM oracles.

### 4.1. Setup

**Datasets.** We evaluate on 14 datasets: six TREC Deep Learning tracks (DL19–DL23, DL-Hard) (Craswell et al., 2020; 2021; 2025a;b;c; Mackie et al., 2021) and eight BEIR datasets (TREC-COVID, NFCorpus, Signal-1M, News, Robust04, Touché, DBPedia, SciFact) (Thakur et al., 2021). For each dataset, BM25 (Robertson et al., 2009) retrieves the top-100 candidates per query.

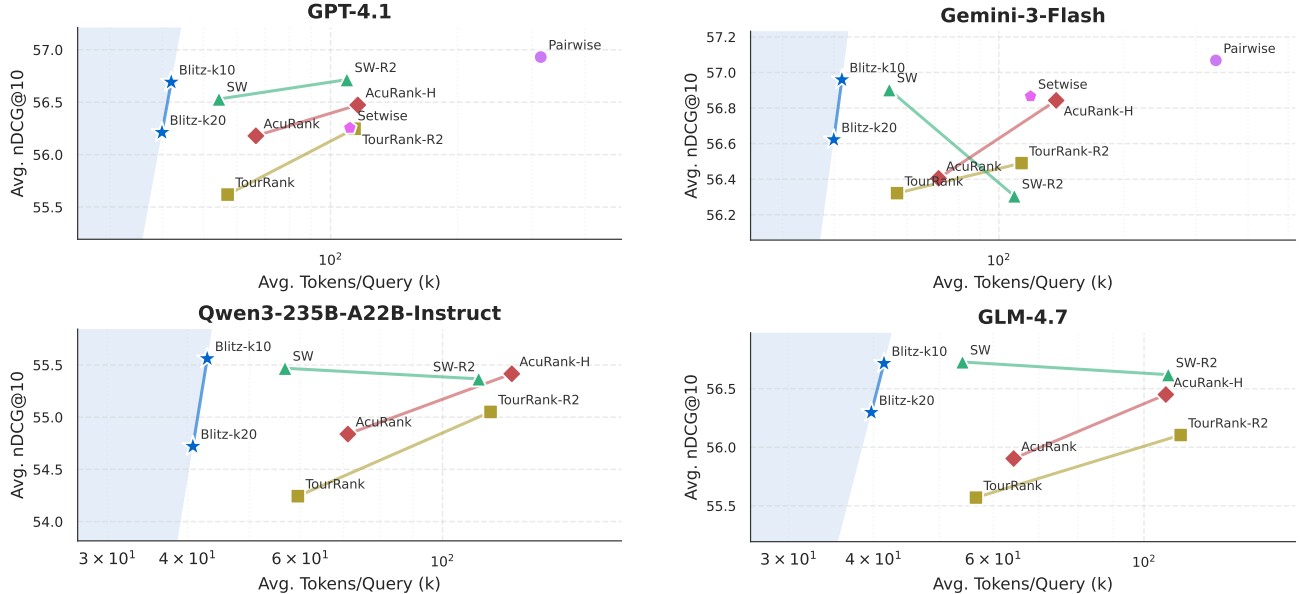

*Figure 4.* Pareto frontiers showing the accuracy-efficiency trade-off across LLM oracles. BLITZRANK (Algorithm 1) consistently occupies the upper-left region, achieving competitive accuracy with 25–40% fewer tokens than methods with comparable structure (§B).

**Metrics.** We measure ranking quality via nDCG@$k$ (Järvelin & Kekäläinen, 2002) and efficiency via input tokens per query – isolating algorithmic efficiency from implementation details.

**Baselines.** We compare against five LLM-based reranking methods: Sliding Windows (Sun et al., 2023) (single-pass and two-pass), TourRank (Chen et al., 2025) (single and two-round), AcuRank (Yoon et al., 2025) (standard and high-precision), Setwise (Zhuang et al., 2024b), and Pairwise (Qin et al., 2024). See Appendix A for detailed comparison rationale.

**Oracles.** All methods use the same underlying LLM with the RankGPT prompt format (Sun et al., 2023). We evaluate five diverse LLMs: GPT-4.1, Gemini-3-Flash, GLM-4.7 (Team et al., 2025), DeepSeek-V3.2 (DeepSeek-AI et al., 2025), and Qwen3-235B-A22B-Instruct (Team, 2025), allowing us to assess whether our efficiency gains generalize across models of varying capabilities.

**Our method.** We evaluate BLITZRANK, or simply BLITZ, with two window sizes: $k = 10$ and $k = 20$. We set target $m = 10$, so BLITZRANK terminates once the current top-10 documents are resolved.

### 4.2. Main Results: Accuracy-Efficiency Frontier

Our main results (summarized in Table 1 and Figure 4; full per-dataset breakdown in Table 6) show that the tournament graph framework achieves comparable or superior ranking quality while consuming substantially fewer tokens, validating the hypothesis that transitive

*Table 1.* Summary of reranking quality and efficiency, macro-averaged across 14 datasets and 2 LLM oracles (GPT-4.1 & Gemini-3-Flash). Relative cost is measured against BLITZ-k20.

| Method | nDCG@10 | Tokens | Rel. Cost |
|---|---|---|---|
| BM25 (no rerank) | 41.1 | 0 | — |
| Pairwise | **57.0** | 324k | 8.1× |
| Setwise | 56.6 | 115k | 2.9× |
| TourRank | 56.0 | 57k | 1.4× |
| TourRank-R2 | 56.4 | 114k | 2.8× |
| SW | 56.7 | 54k | 1.4× |
| SW-R2 | 56.5 | 109k | 2.7× |
| AcuRank | 56.3 | 69k | 1.7× |
| AcuRank-H | 56.6 | 127k | 3.2× |
| BLITZ-k20 | 56.4 | **40k** | **1.0×** |
| BLITZ-k10 | 56.9 | 42k | 1.1× |

closure yields practical efficiency gains.

**Efficiency gains.** BLITZRANK requires significantly fewer tokens than all baselines across every oracle. With GPT-4.1, BLITZ-k10 and BLITZ-k20 consume 42k and 40k tokens per query, respectively – a 22–26% reduction compared to SW (54k) and TourRank (57k), and 37–40% fewer than AcuRank (67k). The gap widens dramatically against comparison-based methods: Pairwise requires 315k tokens (7.5× more than BLITZ-k10) and Setwise requires 111k tokens (2.6× more). Higher-computation variants exacerbate these differences: SW-R2 (109k), TourRank-R2 (114k), and AcuRank-H (116k) all consume 2.6–2.9× more tokens than BLITZ-k10.

*Table 2.* Macro-averaged nDCG@$k$ across 14 datasets × 5 models.

| Method | nDCG@1 | nDCG@5 | nDCG@10 |
|---|---|---|---|
| BM25 retrieval | 46.5 | 43.0 | 41.1 |
| TourRank | 63.3 | 58.8 | 55.5 |
| TourRank-R2 | 63.5 | 59.3 | 56.1 |
| SW | **66.2** | 59.8 | **56.6** |
| SW-R2 | 65.7 | 59.6 | 56.4 |
| AcuRank | 64.5 | 59.5 | 55.9 |
| AcuRank-H | 65.6 | 59.8 | 56.4 |
| **BLITZ-k20** | 66.1 | 59.7 | 56.0 |
| **BLITZ-k10** | 66.0 | **60.1** | 56.5 |

*Table 3.* nDCG@10 at matched window sizes (GPT-4.1, TREC DL19 and DL20). Sliding Window's quality degrades sharply at $k=10$ while BLITZRANK maintains comparable performance.

| Method | k | DL19 | DL20 |
|---|---|---|---|
| Sliding Window | 20 | 74.0 | 70.8 |
| Sliding Window | 10 | 56.4 | 53.2 |
| BLITZRANK | 20 | 74.6 | 70.7 |
| BLITZRANK | 10 | 73.6 | 72.4 |

*Table 4.* SCC statistics at convergence on DL19 (43 queries) using GPT-4.1. Larger $k$ requires fewer rounds but produces more cycles. This reflects a pattern of increased cycle occurrences when comparing many documents simultaneously.

| Config | Queries | #Rounds | Final #SCCs | Avg SCC Size |
|---|---|---|---|---|
| $k = 10$ | 43 | 13.4 | 93.7 | 1.069 |
| $k = 20$ | 43 | 6.6 | 85.3 | 1.180 |

**Ranking quality and Model Generalization.** The efficiency-accuracy trade-off generalizes robustly across all five LLM oracles (Table 1). Despite using fewer oracle calls, BLITZRANK matches or exceeds baseline accuracy across almost all configurations. With GPT-4.1, BLITZ-k10 achieves 56.7 average nDCG@10 – matching the best sliding-window variant (SW-R2: 56.7) at less than 40% of its token cost, and outperforming SW (56.5), TourRank (55.6), AcuRank (56.2), and Setwise (56.3). Only Pairwise is marginally higher (56.9) but at $7.5×$ the cost. With Gemini-3-Flash, BLITZ-k10 achieves 57.0 nDCG@10, within 0.1 points of Pairwise (57.1) while using $8×$ fewer tokens (42k vs. 334k). With GLM-4.7, BLITZ-k10 (56.7) matches SW (56.7) and exceeds TourRank (55.6) at 24% lower token cost (41k vs. 54k). With DeepSeek-V3.2, BLITZ-k10 (56.6) still outperforms TourRank (55.5) and AcuRank (56.0) at lower cost. With Qwen3-235B – the weakest oracle – BLITZ-k10 (55.6) exceeds all baselines except SW (55.5, within noise), again at reduced token consumption (43k vs. 57k). Table 2 shows that trends for nDCG@1 and nDCG@5 are consistent.

**Pareto dominance.** Across all oracles, BLITZRANK consistently achieves quality comparable to the most accurate methods at a fraction of their cost (Figure 4). Methods that match BLITZ-k10's accuracy (e.g., SW-R2, AcuRank-H) require $2–3×$ more tokens; methods that match its efficiency achieve lower accuracy. Appendix B provides further efficiency details.

### 4.3. Effect of Window Size

BLITZRANK's primary user-facing hyperparameter is the window size $k$. Across all 5 oracles (Table 6), BLITZ-k10 consistently outperforms BLITZ-k20 in ranking quality: by 0.5 with GPT-4.1 (56.7 vs. 56.2), 0.4 with Gemini-3-Flash (57.0 vs. 56.6), and 0.9 with Qwen3-235B (55.6 vs. 54.7). The token cost difference between configurations is modest (40–43k), suggesting that smaller windows, with more queries but finer-grained comparisons, provide better

accuracy. We note that $k=10$ outperforms $k=20$ despite comparing fewer documents per query; as analyzed in §4.4, larger windows cause LLMs to produce more cyclic judgments among similar documents due to positional attention limitations. Additionally, convergence is highly predictable: for $k=10$, BLITZRANK terminates in 12–15 rounds (mean 13.6, std 0.58), reflecting the progress guarantee of the greedy SCC schedule (Theorem 60) and enabling reliable cost estimation. See Appendix C for detailed convergence analysis.

**Comparison with Sliding Window at $k=10$.** Table 3 compares both methods at matched window sizes on DL19 and DL20. Sliding Window's quality degrades sharply from $k=20$ to $k=10$ (74.0→56.4 on DL19), because with stride 5 it propagates only the top-5 documents per pass which is insufficient to identify the top 10. In contrast, BLITZRANK at $k=10$ maintains quality comparable to $k=20$ (73.6 vs. 74.6 on DL19), because correctness is certified by the resolution criterion rather than window coverage.

### 4.4. Analysis of Strongly Connected Components

We analyze SCCs formed by BLITZRANK on DL19 using GPT-4.1 to understand when and why cycles arise, and what they reveal about ranking difficulty.

**Evolution of Strongly Connected Components.** Table 4 summarizes SCC statistics at convergence, and Figure 5 traces their evolution across rounds. Both configurations begin with 100 singleton SCCs (one per document). With $k=20$, merging begins earlier – around rounds 5–7 – as BLITZRANK compares larger document sets where cyclic judgments are more likely. By convergence, $k=20$ produces 85.3 SCCs with average size 1.18, while $k=10$ produces 93.7 SCCs with average size 1.07.

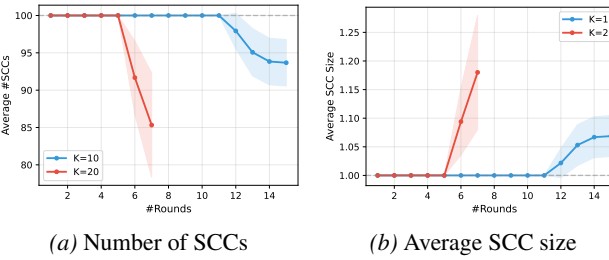

*(a)* Number of SCCs     *(b)* Average SCC size

*Figure 5.* Evolution of SCCs on DL19 with GPT-4.1. Solid lines and shaded regions show means and variance across queries, respectively. **(a)** Both $k$'s begin with 100 singleton SCCs. $k$=20 forms cycles earlier (rounds 5–7), ending with ∼85 SCCs. $k$=10 forms fewer cycles and also later. **(b)** Average SCC size follows a similar pattern: $k$=20 reaches 1.18 average size by round 7, while $k$=10 reaches only 1.07 by round 15.

*Table 5.* BM25 score variance within SCCs vs. neighboring documents on DL19 using GPT-4.1. Documents within SCCs exhibit ∼40% lower BM25 variance than equal-sized neighbor groups (ratio ∼0.6), confirming that cycles capture genuinely similar documents rather than arbitrary ties.

| Config | SCCs ($\geq 2$) | Avg Size | Within-SCC Std | Neighbor Std | Ratio |
|--------|-----------------|----------|----------------|--------------|-------|
| $K = 10$ | 100 | 3.72 | 0.605 | 1.032 | 0.59 |
| $K = 20$ | 126 | 6.01 | 0.695 | 1.125 | 0.62 |

The shaded regions in Figure 5 indicate variance across queries. The $k$=20 configuration exhibits higher variance in both SCC count and size, reflecting greater sensitivity to query-specific document similarity. This explains the accuracy gap between configurations (Table 6): larger windows induce more ties, and ties at the top-$m$ boundary directly impact nDCG@10.

**Larger windows produce larger SCCs.** At every rank position, $k$=20 produces larger SCCs than $k$=10. The effect is most pronounced in mid-ranks: at rank 15, $k$=20 averages 8.86 documents per SCC compared to 2.74 for $k$=10. This likely reflects the "lost in the middle" phenomenon (Liu et al., 2024): when comparing 20 documents simultaneously, LLMs struggle to attend to items in the middle of the list, producing inconsistent judgments that manifest as cycles. Smaller windows ($k$=10) reduce this burden, yielding more consistent orderings.

**SCCs Capture Genuinely Similar Documents.** We use BM25 score variance as a proxy for document similarity: documents with similar lexical relevance to the query should have similar BM25 scores. For each SCC of size $\geq 2$, we compute the standard deviation of BM25 scores within the SCC and among equal-sized groups of neighboring (non-tied) documents. Table 5 shows that documents within SCCs have substantially lower BM25 variance than their neighbors. For $k$=10, within-SCC standard deviation is 0.605 compared to 1.032 for neighbors – a ratio of 0.59. For $k$=20, the ratio is 0.62 (0.695 vs.

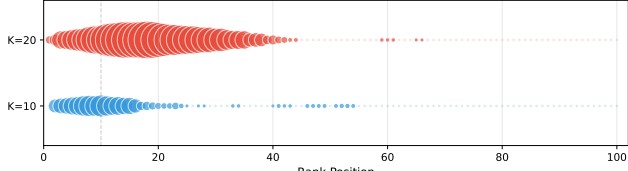

*(a)* SCC locations by rank position (bubble size ∝ SCC size)

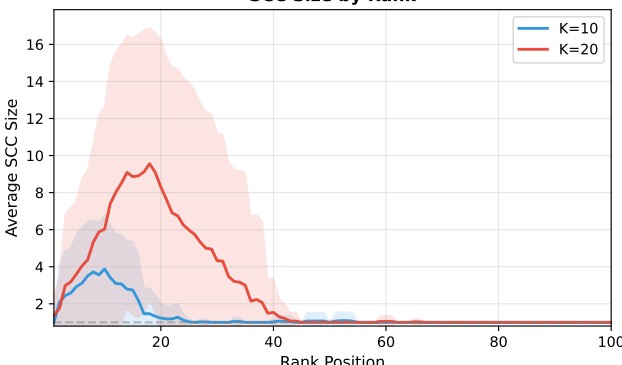

*(b)* Average SCC size by rank position

*Figure 6.* Spatial distribution of strongly connected components after convergence on DL19 using GPT-4.1.
**(a)** Each bubble represents an SCC; size indicates number of documents. The dashed vertical line marks rank 10 (top-$m$ boundary). $k$=20 produces larger SCCs concentrated in ranks 10–40, while $k$=10 produces smaller SCCs focused in ranks 1–20. **(b)** Average SCC size exhibits a wave pattern: both configurations show peaks in mid-ranks rather than at the tail.

1.125). This ∼40% reduction in variance confirms that cycles capture documents that are genuinely similar in lexical relevance, not arbitrary ties. Notably, $k$=10 SCCs have lower within-SCC variance (0.605 vs. 0.695) despite smaller average SCC size (3.72 vs. 6.01). This suggests that $k$=10 successfully resolves easier ambiguities through finer comparisons, leaving only the most difficult cases as unresolved cycles – explaining its consistent accuracy advantage.

**Oracle noise.** Our framework assumes a deterministic oracle (§2.1), but LLM oracles are inherently noisy. The SCC analysis above provides indirect evidence that this noise is well-behaved: cycles capture genuinely similar documents (∼40% lower BM25 variance), suggesting that noisy edges are concentrated among hard-to-distinguish items rather than corrupting long transitive chains. The consistent performance across five oracles of varying capability further indicates that the efficiency gains are robust to oracle-specific noise characteristics.

## 5. Conclusion

We presented a tournament graph framework for top-$m$ selection via $k$-wise comparison oracles, alongside

BLITZRANK, a greedy algorithm with provable correctness and termination guarantees. Empirically, BLITZRANK achieves Pareto dominance across 14 benchmarks and 5 LLM oracles, matching or exceeding accuracy while reducing token consumption by 25–40% compared to comparable methods, and up to $7\times$ compared to pairwise approaches. Moreover, convergence is highly predictable, enabling reliable cost estimation. Finally, our analysis of strongly connected components confirms that cycles capture genuine lexical similarity rather than arbitrary noise, validating the interpretation of non-transitivity as structure.

### 5.1. Future Work

**Noisy oracle.** Our framework assumes a deterministic oracle: each pairwise comparison has a fixed ground-truth outcome. Real oracles – LLMs, crowdworkers, human experts – are noisy, and this noise interacts asymmetrically with transitive inference: in principle, a single erroneous edge can collapse a long chain into an SCC. While our empirical analysis suggests such catastrophic collapse is rare with capable models, a principled treatment of noise remains a key challenge. We also leave the explicit incorporation of prior retrieval scores into query selection to future work.

**Query complexity.** We showed $\lceil (n-1)/(k-1) \rceil$ query complexity for top-1 selection and conjectured a tight bound for general $m$ (Conjecture 63). A formal proof or matching lower bound remains open.

**Noisy and probabilistic oracles.** Extending the framework to handle oracle noise is a significant open problem. One approach models each edge as a random variable whose confidence accumulates across repeated or corroborating observations. A key challenge is that edges differ in *structural importance*: an edge along a long chain discriminates many pairs transitively, so its corruption is catastrophic, while a leaf edge affects only one comparison. Models that account for this asymmetry – weighting edges by transitive reach, or using soft SCCs that resist collapse from isolated errors – could yield algorithms that degrade gracefully under noise.

**Incorporating priors.** First-stage retrieval scores provide a natural prior over the ranking. Algorithms that initialize edge beliefs with priors, or prioritize queries where the prior is uncertain, could improve both efficiency and robustness. This connects to active learning formulations where queries maximize information gain relative to prior belief.

We hope the deterministic framework developed here provides a foundation for these extensions, much as noiseless sorting algorithms underpin the study of noisy comparison models.

## Impact Statement

This paper develops algorithmic methods for query-efficient ranking using expensive comparison oracles. By reducing the number of oracle calls required to identify top candidates, our framework lowers computational costs when LLMs serve as oracles – contributing to more sustainable use of large-scale models. The techniques are domain-agnostic and apply broadly to ranking problems with costly comparisons (crowdsourcing, human evaluation, tournament design). We do not foresee specific negative societal consequences beyond those common to advances in information retrieval and ranking systems.

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

# Part I

# Additional Experimental Details

*Table 6.* Reranking quality (nDCG@10) and efficiency (input tokens per query in thousands).

| Method | TREC-DL | | | | | | BEIR | | | | | | | | Avg. | |
|---|---|---|---|---|---|---|---|---|---|---|---|---|---|---|---|---|
| | DL19 | DL20 | DL21 | DL22 | DL23 | DLHard | COVID | NFC | Signal | News | R04 | Touche | DBP | Scif | nDCG | Tok |
| *No Reranking* | | | | | | | | | | | | | | | | |
| BM25 | 50.6 | 48.0 | 44.6 | 26.9 | 26.2 | 28.5 | 59.5 | 33.7 | 33.0 | 39.5 | 40.7 | 44.2 | 31.8 | 67.9 | 41.1 | 0 |
| *GPT-4.1* | | | | | | | | | | | | | | | | |
| Pairwise | 74.8 | 72.3 | 73.0 | 52.6 | 49.3 | 40.0 | 83.1 | 39.8 | 33.3 | 51.4 | 67.1 | 35.4 | 45.0 | 80.0 | **56.9** | 315k |
| Setwise | 73.5 | 70.4 | 72.0 | 51.9 | 48.2 | 39.6 | 83.3 | 39.9 | 34.0 | 49.6 | 66.6 | 35.6 | 44.4 | 78.4 | 56.3 | 111k |
| TourRank | 72.6 | 69.4 | 71.2 | 51.8 | 49.3 | 37.6 | 84.3 | 40.1 | 31.5 | 50.3 | 66.3 | 32.2 | 44.5 | 77.6 | 55.6 | 57k |
| TourRank-R2 | 74.5 | 70.9 | 70.3 | 51.8 | 50.3 | 38.6 | 84.5 | 40.4 | 33.2 | 50.0 | 66.3 | 32.4 | 45.5 | 78.7 | 56.2 | 114k |
| SW | 74.0 | 70.8 | 70.5 | 51.4 | 49.5 | 37.4 | 82.7 | 40.9 | 34.3 | 51.5 | 66.7 | 36.9 | 45.9 | 79.0 | 56.5 | 54k |
| SW-R2 | 73.4 | 72.0 | 70.5 | 52.0 | 49.6 | 39.4 | 82.6 | 40.5 | 34.0 | 51.9 | 66.0 | 36.4 | 46.4 | 79.1 | 56.7 | 109k |
| AcuRank | 74.2 | 70.5 | 70.5 | 52.1 | 49.9 | 38.2 | 83.5 | 40.4 | 32.3 | 50.0 | 66.8 | 33.0 | 45.6 | 79.4 | 56.2 | 67k |
| AcuRank-H | 73.7 | 70.8 | 71.1 | 51.5 | 51.0 | 37.9 | 83.5 | 40.5 | 32.6 | 51.6 | 67.0 | 32.9 | 45.9 | 80.5 | 56.5 | 116k |
| BLITZ-k20 | 74.6 | 70.7 | 70.4 | 51.4 | 48.9 | 37.3 | 82.4 | 39.9 | 33.7 | 50.1 | 66.4 | 36.5 | 45.3 | 79.2 | 56.2 | 40k |
| BLITZ-k10 | 73.6 | 72.4 | 71.3 | 52.0 | 50.2 | 37.7 | 83.8 | 40.3 | 33.0 | 50.3 | 66.8 | 37.4 | 45.6 | 79.4 | 56.7 | 42k |
| *Gemini-3-Flash* | | | | | | | | | | | | | | | | |
| Pairwise | 74.9 | 72.4 | 72.7 | 53.3 | 50.2 | 38.9 | 82.7 | 40.8 | 34.2 | 48.8 | 67.9 | 36.3 | 45.5 | 80.4 | **57.1** | 334k |
| Setwise | 73.9 | 73.1 | 72.6 | 52.8 | 49.5 | 38.9 | 82.5 | 40.7 | 33.0 | 47.6 | 67.4 | 37.9 | 44.7 | 81.6 | 56.9 | 119k |
| TourRank | 74.9 | 72.4 | 71.9 | 51.0 | 49.8 | 36.6 | 82.7 | 40.6 | 33.8 | 49.7 | 66.4 | 35.1 | 45.6 | 77.9 | 56.3 | 57k |
| TourRank-R2 | 73.0 | 72.1 | 72.5 | 51.4 | 50.5 | 38.5 | 82.7 | 40.5 | 33.5 | 49.6 | 66.7 | 35.5 | 46.0 | 78.3 | 56.5 | 113k |
| SW | 74.0 | 73.1 | 73.0 | 52.1 | 51.1 | 38.9 | 82.6 | 40.8 | 32.6 | 49.4 | 66.5 | 36.1 | 45.7 | 80.6 | 56.9 | 54k |
| SW-R2 | 73.3 | 71.6 | 72.7 | 50.6 | 51.5 | 38.1 | 81.0 | 40.1 | 32.3 | 47.0 | 65.9 | 37.7 | 46.3 | 80.1 | 56.3 | 109k |
| AcuRank | 73.8 | 71.6 | 72.6 | 51.3 | 51.0 | 37.5 | 81.9 | 40.8 | 32.3 | 48.3 | 67.2 | 35.0 | 45.6 | 80.9 | 56.4 | 71k |
| AcuRank-H | 73.9 | 71.7 | 71.9 | 51.9 | 51.6 | 38.6 | 82.5 | 41.3 | 34.1 | 49.0 | 66.9 | 35.1 | 46.0 | 81.3 | 56.8 | 138k |
| BLITZ-k20 | 74.9 | 72.1 | 72.4 | 51.4 | 49.7 | 38.1 | 82.3 | 41.1 | 33.0 | 48.2 | 65.8 | 37.9 | 45.3 | 80.4 | 56.6 | 40k |
| BLITZ-k10 | 74.8 | 72.0 | 73.0 | 50.8 | 50.9 | 39.4 | 82.3 | 41.0 | 33.9 | 47.4 | 66.6 | 39.6 | 45.9 | 79.9 | 57.0 | 42k |
| *GLM-4.7* | | | | | | | | | | | | | | | | |
| TourRank | 74.1 | 71.2 | 69.8 | 51.0 | 49.4 | 38.4 | 83.4 | 40.0 | 30.4 | 49.1 | 66.0 | 31.8 | 45.9 | 77.5 | 55.6 | 57k |
| TourRank-R2 | 74.8 | 71.0 | 71.4 | 52.3 | 49.2 | 38.7 | 83.3 | 40.4 | 31.9 | 49.0 | 66.4 | 32.6 | 46.1 | 78.2 | 56.1 | 113k |
| SW | 74.2 | 71.6 | 71.5 | 51.7 | 48.3 | 38.5 | 82.9 | 39.8 | 34.1 | 50.4 | 66.2 | 37.6 | 47.6 | 80.0 | **56.7** | 54k |
| SW-R2 | 74.5 | 71.9 | 71.5 | 52.3 | 49.4 | 37.1 | 82.9 | 40.0 | 33.6 | 50.4 | 65.9 | 34.9 | 47.2 | 81.0 | 56.6 | 109k |
| AcuRank | 74.0 | 70.2 | 70.8 | 51.8 | 48.3 | 37.5 | 82.5 | 40.1 | 31.7 | 49.8 | 66.0 | 33.0 | 46.5 | 80.4 | 55.9 | 64k |
| AcuRank-H | 75.1 | 71.4 | 71.6 | 52.4 | 48.4 | 39.3 | 83.4 | 40.5 | 32.1 | 50.3 | 66.7 | 31.7 | 46.5 | 80.9 | 56.4 | 108k |
| BLITZ-k20 | 75.7 | 71.2 | 70.4 | 51.0 | 47.7 | 38.5 | 83.3 | 39.9 | 33.3 | 49.9 | 65.1 | 35.8 | 46.5 | 79.9 | 56.3 | 40k |
| BLITZ-k10 | 72.8 | 72.4 | 71.3 | 52.4 | 49.2 | 38.6 | 81.8 | 39.9 | 33.9 | 50.0 | 66.5 | 38.4 | 46.5 | 80.3 | **56.7** | 41k |
| *Qwen3-235B-A22B-Instruct* | | | | | | | | | | | | | | | | |
| TourRank | 71.3 | 69.0 | 69.0 | 49.7 | 47.3 | 36.6 | 84.8 | 38.7 | 28.3 | 50.3 | 64.0 | 34.5 | 42.1 | 73.8 | 54.2 | 59k |
| TourRank-R2 | 72.8 | 71.2 | 69.0 | 51.5 | 48.3 | 37.4 | 83.6 | 39.8 | 30.7 | 49.5 | 65.6 | 33.0 | 43.0 | 75.3 | 55.1 | 119k |
| SW | 73.5 | 69.7 | 70.6 | 49.9 | 47.8 | 39.2 | 83.8 | 39.6 | 32.5 | 50.5 | 64.5 | 32.5 | 45.0 | 77.2 | 55.5 | 57k |
| SW-R2 | 73.5 | 71.1 | 70.2 | 51.5 | 48.4 | 37.8 | 83.6 | 39.9 | 31.6 | 47.9 | 65.3 | 33.0 | 45.1 | 76.3 | 55.4 | 114k |
| AcuRank | 73.9 | 69.4 | 69.8 | 50.7 | 48.9 | 37.3 | 82.1 | 39.2 | 29.0 | 50.3 | 65.4 | 32.2 | 43.2 | 76.3 | 54.8 | 71k |
| AcuRank-H | 74.2 | 69.6 | 70.2 | 51.3 | 49.5 | 39.0 | 82.7 | 39.6 | 30.5 | 50.3 | 65.9 | 32.1 | 44.2 | 76.5 | 55.4 | 128k |
| BLITZ-k20 | 71.9 | 70.6 | 69.2 | 50.5 | 48.6 | 39.1 | 83.2 | 38.9 | 30.4 | 49.2 | 63.8 | 32.7 | 43.3 | 74.5 | 54.7 | 41k |
| BLITZ-k10 | 74.1 | 71.1 | 70.3 | 49.0 | 47.3 | 39.7 | 83.1 | 39.8 | 31.9 | 50.9 | 65.3 | 35.9 | 43.5 | 75.9 | **55.6** | 43k |
| *DeepSeek-V3.2* | | | | | | | | | | | | | | | | |
| TourRank | 72.7 | 71.2 | 69.5 | 49.6 | 47.8 | 38.8 | 83.3 | 40.0 | 31.4 | 50.0 | 65.2 | 33.5 | 45.0 | 79.6 | 55.5 | 56k |
| TourRank-R2 | 74.5 | 70.9 | 71.0 | 51.5 | 48.6 | 38.2 | 83.9 | 40.9 | 32.8 | 52.7 | 66.1 | 32.7 | 46.1 | 79.6 | 56.4 | 112k |
| SW | 74.6 | 71.4 | 71.2 | 52.5 | 47.5 | 40.8 | 84.3 | 40.7 | 34.1 | 51.4 | 65.6 | 40.0 | 47.2 | 80.7 | **57.3** | 54k |
| SW-R2 | 74.9 | 72.0 | 70.5 | 52.2 | 48.1 | 38.5 | 84.5 | 40.3 | 33.7 | 52.4 | 65.4 | 36.0 | 47.2 | 80.5 | 56.9 | 108k |
| AcuRank | 73.6 | 71.2 | 70.9 | 52.1 | 48.5 | 37.2 | 83.5 | 40.5 | 31.0 | 53.1 | 66.3 | 31.3 | 45.6 | 79.3 | 56.0 | 69k |
| AcuRank-H | 74.5 | 71.4 | 71.3 | 52.4 | 48.8 | 39.4 | 84.5 | 41.2 | 32.2 | 52.2 | 66.8 | 32.6 | 46.4 | 80.9 | 56.8 | 131k |
| BLITZ-k20 | 74.2 | 71.6 | 69.6 | 52.2 | 47.4 | 39.1 | 82.4 | 40.2 | 32.9 | 52.4 | 64.3 | 35.1 | 45.4 | 80.0 | 56.2 | 39k |
| BLITZ-k10 | 74.0 | 71.6 | 70.9 | 52.3 | 47.1 | 40.6 | 84.6 | 40.6 | 32.1 | 50.3 | 65.4 | 37.0 | 46.2 | 80.1 | 56.6 | 41k |

## A. Baseline Comparison Rationale

Each baseline tests a specific aspect of our tournament graph framework.

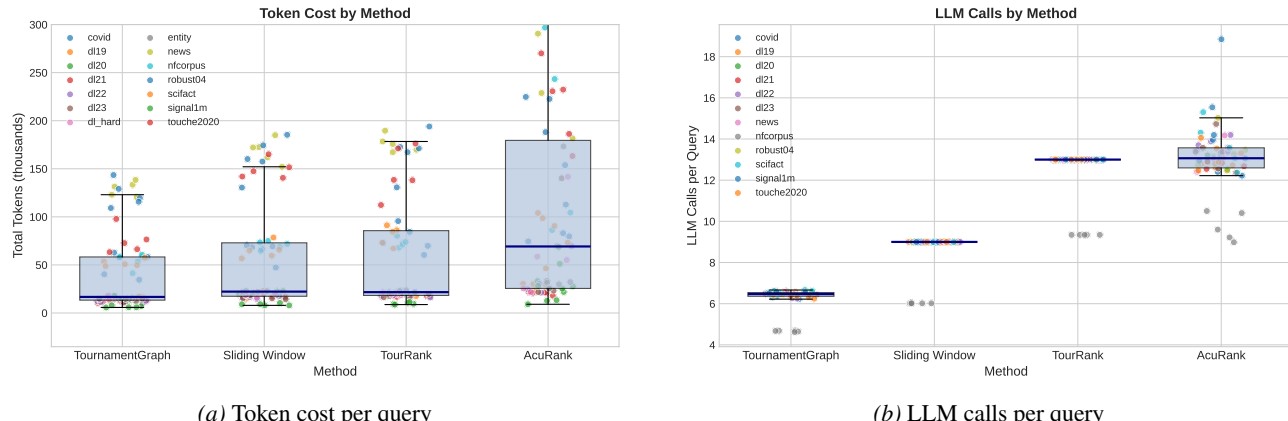

*(a)* Token cost per query                                      *(b)* LLM calls per query

*Figure 7.* Efficiency comparison across listwise reranking methods on 14 datasets (GPT-4.1). Each point represents one dataset; box plots show the distribution. **(a)** Token consumption varies across datasets due to differing document lengths, but BLITZRANK consistently achieves the lowest median cost. AcuRank exhibits the highest variance due to its adaptive computation. **(b)** LLM call counts isolate algorithmic efficiency from document length effects. BLITZRANK requires ∼6.5 calls on average – fewer than Sliding Window (9), TourRank (13), and AcuRank (13). Outliers at lower call counts correspond to NFCorpus, where many queries have fewer than 100 retrieved documents (see Figure 8).

*Table 7.* LLM calls per query across reranking methods. Statistics shown for all 14 datasets and separately excluding NFCorpus (13 datasets). BLITZRANK makes the fewest calls (∼6.5) while Sliding Window and TourRank are deterministic (9 and 13 calls respectively). AcuRank averages ∼13 calls with slight variation due to adaptive refinement.

| | All Datasets | | | | | | Excl. NFCorpus | | | | | |
| Method | Mean | Std | Min | Max | Std% | Rng% | Mean | Std | Min | Max | Std% | Rng% |
|---|---|---|---|---|---|---|---|---|---|---|---|---|
| BLITZRANK | 6.3 | 0.6 | 5 | 7 | 9.0 | 32.0 | 6.5 | 0.1 | 6 | 7 | 2.0 | 7.0 |
| Sliding Window | 8.8 | 0.8 | 6 | 9 | 9.0 | 34.0 | 9.0 | 0.0 | 9 | 9 | 0.0 | 0.0 |
| TourRank | 12.7 | 1.0 | 9 | 13 | 8.0 | 29.0 | 13.0 | 0.0 | 13 | 13 | 0.0 | 0.0 |
| AcuRank | 13.1 | 1.4 | 9 | 19 | 11.0 | 75.0 | 13.4 | 1.1 | 12 | 19 | 8.0 | 49.0 |

**Sliding Windows (Sun et al., 2023)**   This baseline directly tests our central hypothesis: sliding windows process overlapping document sets, yet discard comparison information from previous windows rather than accumulating it. Our framework captures these relationships via transitive closure, potentially achieving equivalent quality with fewer queries.

**TourRank (Chen et al., 2025)**   While TourRank uses a static tournament structure to schedule matches, it does not employ tournament graphs to model comparison outcomes or exploit graph-theoretic properties such as transitivity. This comparison isolates the value of maintaining a tournament graph and propagating information via transitive closure versus using tournaments purely as a scheduling mechanism.

**AcuRank (Yoon et al., 2025).**   Both AcuRank and BLITZRANK are adaptive – terminating when sufficiently confident about the top-$m$ – but employ different certification criteria: AcuRank uses Bayesian score distributions, while our framework certifies via graph-theoretic finalization through transitive closure.

**Setwise (Zhuang et al., 2024b).**   This baseline provides the most direct contrast to our approach: both methods issue $k$-wise comparison queries, but Setwise extracts only the winner – a single comparison – while our framework captures the complete tournament of $\binom{k}{2}$ pairwise relationships.

**Pairwise (Qin et al., 2024).**   Pairwise reranking serves as an upper bound on comparison granularity: each query yields exactly one pairwise relationship. This makes it highly accurate – reducing the cognitive load on LLMs to simple binary judgments – but computationally expensive, requiring 7–8× more tokens than our approach.

*Figure 8.* Distribution of retrieved documents per query in NFCorpus. Unlike other datasets where BM25 retrieves 100 documents per query, NFCorpus exhibits a bimodal distribution with many queries having fewer than 20 candidates (mean: 68.2). This explains the lower LLM call counts for NFCorpus in Table 7.

*Table 8.* Number of rounds as a function of window size $k$, aggregated over 70 configurations (14 datasets $\times$ 5 LLMs). The algorithm exhibits deterministic convergence: for fixed $k$, round count remains nearly constant across datasets and oracles, enabling reliable cost estimation.

| $k$ | Min | Max | Mean | Std | Avg. Tokens |
|---|---|---|---|---|---|
| 10 | 12 | 15 | 13.59 | 0.58 | 44,814 |
| 20 | 6 | 7 | 6.73 | 0.45 | 42,139 |

## B. Efficiency Comparison Details

Setwise and Pairwise consume substantially more tokens than BLITZRANK due to their small comparison windows ($k=3$ and $k=2$, respectively), requiring many more queries to establish order. More interesting is the comparison against methods that share our oracle structure – AcuRank, TourRank, and Sliding Window all prompt the LLM to rank $k=20$ documents per call. Despite this structural similarity, BLITZRANK achieves lower cost through principled information extraction.

Figure 7 visualizes token consumption and LLM call counts across the 14 datasets. Token costs vary across datasets due to differing document lengths, but BLITZRANKconsistently achieves the lowest median (Figure 7a). To isolate algorithmic efficiency from document length effects, we also compare LLM call counts (Figure 7b). BLITZRANKrequires only ∼6.5 calls per query on average, compared to 9 for Sliding Window, 13 for TourRank, and 13 for AcuRank (Table 7).

The outliers at lower call counts in Figure 7b correspond to NFCorpus, where many queries have fewer than 100 retrieved documents. Figure 8 shows that NFCorpus exhibits a bimodal distribution with many queries having fewer than 20 candidates (mean: 68.2), explaining why all methods require fewer calls on this dataset. We report statistics both with and without NFCorpus in Table 7 to isolate this effect.

**Comparison with AcuRank.** AcuRank exhibits both higher cost and less predictable cost than BLITZRANK. While BLITZRANKhas a standard deviation of just 2% in call counts (excluding NFCorpus), AcuRank's standard deviation reaches 8%, with a range spanning 49% of its mean (Table 7). This variance stems from AcuRank's design: it models each document with an independent score distribution and adaptively allocates comparisons based on uncertainty. In contrast, our framework maximizes information extraction by capturing complete tournaments from each query and propagating relationships via transitive closure, yielding deterministic convergence behavior.

AcuRank's adaptivity does offer a potential advantage: spending more computation on genuinely difficult queries. However, modern LLMs produce largely consistent judgments, limiting the practical benefit of this flexibility. Whether combining adaptive allocation with tournament graph structure can yield further gains remains an open question.

## C. Convergence Analysis

Table 8 and Figure 9 characterize the algorithm's convergence behavior across window sizes. For $k=10$, convergence requires 12–15 rounds (mean 13.6, std 0.58); for $k=20$, only 6–7 rounds (mean 6.7, std 0.45).

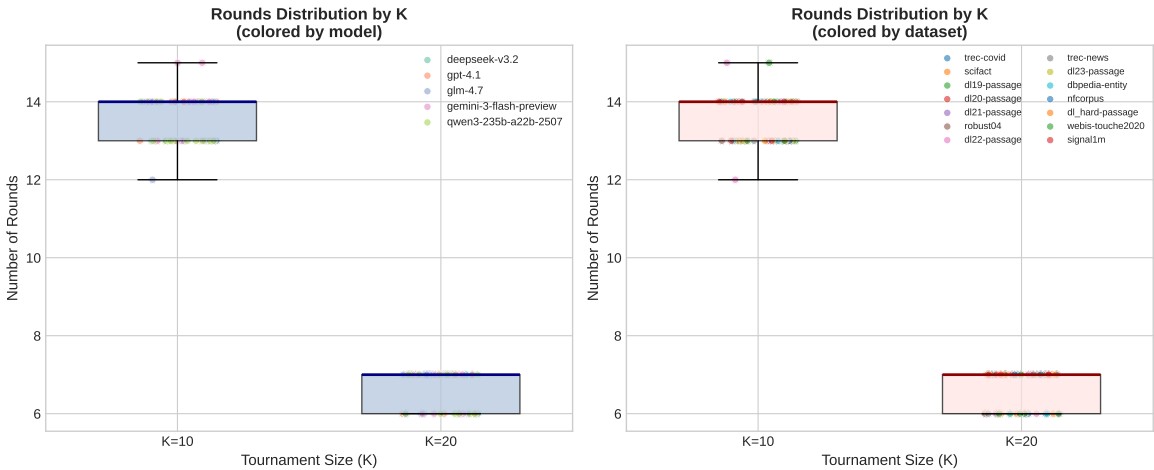

*Figure 9.* Distribution of rounds until convergence for $k=10$ and $k=20$, aggregated across 70 configurations (14 datasets $\times$ 5 LLMs). Left panel colors by model, right panel by dataset. The algorithm exhibits highly predictable convergence: $k=10$ requires 12–15 rounds (mean 13.6), while $k=20$ requires only 6–7 rounds (mean 6.7). Variance across models and datasets is minimal, indicating that convergence depends primarily on problem structure rather than oracle-specific factors.

This predictability is notable: despite variation in dataset characteristics, query difficulty, and oracle capabilities, round count remains nearly constant for fixed $k$. The deterministic convergence stems from the progress guarantee (Lemma 46): each round reveals at least one new edge, and the algorithm terminates once the top-$m$ candidates are finalized.

Token consumption also remains stable across oracles despite differences in their ranking capabilities, indicating that the algorithm's query complexity depends primarily on the problem structure ($n$, $m$, $k$) rather than oracle-specific factors. This contrasts sharply with adaptive methods like AcuRank, which exhibit 8% variance in call counts – the deterministic nature of our framework enables reliable cost estimation before execution.

## D. SCC Analysis

*Table 9.* Average SCC size by rank position on DL19 using GPT-4.1. SCCs concentrate in mid-ranks where the algorithm focuses queries, peaking around rank 15 for $k=20$ (avg 8.86) and rank 5–10 for $k=10$ (avg $\sim$4). Beyond rank 50, SCC size drops to 1.0 as lower-ranked documents are excluded via transitive closure.

| Rank | 1 | 2 | 3 | 5 | 10 | 15 | 20 | 25 | 50 |
|---|---|---|---|---|---|---|---|---|---|
| $K = 10$ | 1.00 | 2.09 | 2.44 | 2.93 | 3.88 | 2.74 | 1.23 | 1.02 | 1.00 |
| $K = 20$ | 1.35 | 1.77 | 2.98 | 3.60 | 6.02 | 8.86 | 8.30 | 5.98 | 1.00 |

**SCCs concentrate in mid-ranks, not at the tail.** One might expect SCCs to cluster among low-ranked documents, where items are uniformly irrelevant and difficult to distinguish. Instead, Figure 6b shows a wave pattern: SCC size peaks around ranks 10–20 for $k=20$ and ranks 5–10 for $k=10$, then drops to 1.0 beyond rank 50. This pattern reflects the algorithm's greedy scheduling (§2.4): queries target non-finalized SCCs with smallest in-reach, concentrating comparisons near the decision boundary. Lower-ranked documents are often excluded via transitive closure after a single comparison – losing to a document that loses to a top-$m$ candidate finalizes their position without further queries.

## E. Scheduling Ablation

To isolate the contribution of greedy SCC scheduling (§2.4), we compare three scheduling strategies on synthetic transitive instances where oracle calls can be counted exactly. We run BLITZRANK with the default greedy schedule and two random baselines: (1) *Random (cands.)*, which selects a random $k$-subset from the current candidate set, and (2) *Random (all)*, which selects a random $k$-subset from all available nodes. Values in Table 10 are mean $\pm$ std of total oracle calls over 20 seeds. The greedy schedule is 2.5–5.8$\times$ more efficient than random scheduling from the candidate set, and 8.6–28.6$\times$ more efficient than random scheduling from all nodes.

*Table 10.* Scheduling ablation on synthetic transitive instances ($m = n$). Oracle calls (mean $\pm$ std over 20 seeds). Ratio = Random / Greedy.

| Config | Greedy | Random (cands.) | Ratio | Random (all) | Ratio |
|---|---|---|---|---|---|
| $n{=}100,\ k{=}10$ | $37.7 \pm 1.2$ | $132.1 \pm 9.6$ | $3.5\times$ | $541.5 \pm 102.4$ | $14.4\times$ |
| $n{=}100,\ k{=}20$ | $14.3 \pm 0.6$ | $35.9 \pm 4.3$ | $2.5\times$ | $123.5 \pm 26.3$ | $8.6\times$ |
| $n{=}200,\ k{=}10$ | $89.2 \pm 1.6$ | $520.9 \pm 32.1$ | $5.8\times$ | $2554.6 \pm 432.4$ | $28.6\times$ |
| $n{=}200,\ k{=}20$ | $33.8 \pm 0.7$ | $129.4 \pm 8.9$ | $3.8\times$ | $653.5 \pm 184.5$ | $19.3\times$ |

# Part II

# Theory

The most direct proof of Algorithm 1's correctness and termination can be made short, as the proof sketch in §2.5 demonstrates: output correctness follows from a single ordering lemma on resolved vertices, and termination from the tied-SCC progress argument on the condensation DAG. We nonetheless present the full development through the transitive special case (Algorithm 2), the general non-transitive case (Algorithm 3), and finally the resolution-based variant (Algorithm 1) for two reasons. First, the progression builds intuition for why the algorithm works and how it was designed: the transitive case isolates the finalization engine on DAGs, and the non-transitive case shows how condensation lifts it to arbitrary tournaments, before the resolution criterion simplifies the stopping rule. Second, the intermediate machinery – particularly the finalization threshold, the rank spectrum, the condensation projection, and the SCC refinement framework – provides the essential tools for two concrete open problems identified in §5.1: proving query complexity bounds for general $m$ (Conjecture 63, which would likely begin with the transitive case and lift via condensation) and extending the framework to noisy oracles (which requires replacing the subgraph invariant with probabilistic analogues of the finalization and resolution criteria).

## F. Problem Statement

We consider the problem of identifying top-ranked vertices in an unknown tournament graph through queries to an oracle.

### F.1. Unknown Tournaments and Oracle Model

**Tournament Graphs.** A *tournament* is a directed graph $G^* = (V, E^*)$ in which every pair of vertices is connected by exactly one directed edge: for all $u \neq v \in V$, either $(u, v) \in E^*$ or $(v, u) \in E^*$, but not both. The edge $(u, v)$ is interpreted as $u$ defeating $v$ in a comparison. Throughout, $G^*$ denotes a fixed but unknown tournament that we seek to discover through oracle queries. We use the subscript and superscript $*$ to indicate quantities associated with the tournament $G^*$.

**Neighborhoods and Degrees.** For a vertex $u$ in a directed graph $G = (V, E)$, the *out-neighbors* of $u$ is $N_G^+(u) = \{v \in V : (u, v) \in E\}$ with *out-degree* $\deg_G^+(u) = |N_G^+(u)|$, and the *in-neighbors* of $u$ is $N_G^-(u) = \{v \in V : (v, u) \in E\}$ with *in-degree* $\deg_G^-(u) = |N_G^-(u)|$.

**Oracle Model.** For a set $U$ with $|U| = n$ and a positive integer $k$, let $\binom{U}{k} := \{S \subseteq U : |S| = k\}$ denote the collection of all $k$-subsets of $U$, and let $U^{\underline{k}} := \{(u_1, \ldots, u_k) \in U^k : u_i \neq u_j \text{ for all } i \neq j\}$ denote the set of all ordered $k$-tuples of distinct elements from $U$. We assume access to an oracle $O_{G^*} : \binom{V}{k} \to 2^{E^*}$ that, given any $k$-subset $S \in \binom{V}{k}$, returns all directed edges of $G^*$ induced by $S$:

$$O_{G^*}(S) := E^* \cap S^{\underline{2}}.$$

Intuitively, given $k$ items, the oracle "organizes" a subtournament between them and reveals the result.

**Reachability.** We write $u \rightsquigarrow_G v$ to denote the existence of a directed path from $u$ to $v$ in $G$, i.e. $u \rightsquigarrow_G v$ if there exist vertices $v_0, v_1, \ldots, v_k \in V$ with $k \geq 0$, $v_0 = u$, $v_k = v$, and $(v_i, v_{i+1}) \in E$ for all $i = 0, 1, \ldots, k - 1$. The *in-reach* and *out-reach* of a vertex $v$ are defined as

$$R_G^-(v) := \{u \in V \setminus \{v\} : u \rightsquigarrow_G v\}, \tag{4}$$

$$R_G^+(v) := \{u \in V \setminus \{v\} : v \rightsquigarrow_G u\}. \tag{5}$$

In a tournament, $|R_{G^*}^-(v)|$ measures how many items transitively dominate $v$ – a reachability-based loss count. When $G^*$ is *transitive* (i.e., $(u, v)$ and $(v, p)$ in $E^*$ implies $(u, p)$ in $E^*$), in-reach coincides with in-neighborhood (Corollary 27), so $|R_{G^*}^-(v)| = \deg_{G^*}^-(v)$.

### F.2. Problem Statement: Ranking in Tournaments with $k$-wise Oracle

We formalize the primary ranking problem of interest as follows.

**Problem 1.** Given an unlabeled vertex set $V$ of size $n$, an oracle $O_{G^*}$ with query size $k \geq 2$, and a target count $m \leq n$, identify a set of top-$m$ ranked vertices $S = \{v^{(1)}, \ldots, v^{(m)}\}$ such that

$$|R_{G^*}^-(v^{(1)})| \leq |R_{G^*}^-(v^{(2)})| \leq \cdots \leq |R_{G^*}^-(v^{(m)})| \tag{6}$$

$$\text{and} \quad \max_{i \leq m} |R_{G^*}^-(v^{(i)})| \leq \min_{u \notin S} |R_{G^*}^-(u)|, \tag{7}$$

using as few oracle queries as possible.

*Remark* 2 (In-reach ranking and the decomposition ordering). Every tournament decomposes uniquely into strongly connected components (SCCs), and the condensation – the DAG obtained by collapsing each SCC to a single node – is itself a transitive tournament (Bang-Jensen & Gutin, 2008). Vertices in the same SCC have identical in-reach, so the in-reach ranking coincides with the *decomposition ordering* (also called the *Smith set stratification*) (Laslier, 1997): it ranks vertices by the position of their SCC in the condensation, declaring vertices within the same SCC tied. When $G^*$ is transitive, every SCC is a singleton, in-reach reduces to in-degree, and the ranking is a total order identical to the Copeland ranking. In the non-transitive case, ties may cause the top-$m$ set (and its internal ordering) to be non-unique. Refining the within-SCC ordering to match the full Copeland ranking requires additionally discovering the induced subtournament on each output SCC, at a cost of at most $\binom{|C|}{2}$ further queries per component $C$.

### F.3. Theoretical Foundations and Related Work

**Tournament solutions and query complexity.** Tournaments are well-studied in computational social choice (Brandt et al., 2016; Laslier, 1997). Beyond the Copeland and decomposition orderings used here, classical solutions include the Slater ranking (minimum feedback arc set) and Kemeny ranking (minimum Kendall-tau distance); both are NP-hard (Alon, 2006; Bartholdi III et al., 1989). Dey (2017) established query complexity bounds for tournament solutions under a pairwise oracle, proving that most require $\Omega(n^2)$ queries. When the top cycle has bounded size $c$, they show all solutions can be found in $O(nc + n \log n / \log(1 - 1/c))$ queries; however, this identifies only the first SCC (the top cycle) and does not recover the full condensation ordering over all components. No existing work, to our knowledge, studies the query complexity of recovering the complete SCC stratification.

The decomposition ordering is a natural target for our framework for two reasons. First, the finalization engine (introduced in §F.4 and detailed in §G) determines SCC positions without resolving internal structure, so the algorithm terminates as soon as tier membership is established. Second, it gives the honest answer when preferences are cyclic. In our primary application – LLM-as-a-judge reranking – cycles arise not from noise but from genuine ambiguity: the oracle cannot consistently order those items, and reporting them as tied is more faithful than imposing an artificial distinction. When a total ordering is needed, ties may be broken by a secondary signal (e.g., the original retrieval score) or additional fine-grained queries.

**Multi-wise comparisons.** The information-theoretic advantage of $k$-wise comparisons is established in Ren et al. (2021), who prove that full-ranking feedback improves sample complexity by a factor of $k \log k$ over winner feedback for ranking recovery. Saha & Gopalan (2019) show similar gains under the Plackett–Luce model. Jang et al. (2017) and Chen et al. (2018) provide near-optimal algorithms for top-$m$ selection under parametric models. For noisy comparisons, Feige et al. (1994) established $O(n \log n)$ bounds for sorting, tightened by Gu & Xu (2023). Our framework operates non-parametrically, using graph structure rather than score estimation, and addresses the query-efficient top-$m$ selection problem that arises when each comparison is expensive. More specifically, our setting differs from prior work in three respects: (i) we assume a deterministic tournament rather than a stochastic comparison model, (ii) we use a $k$-wise oracle ($k \geq 2$) rather than restricting to pairwise queries, and (iii) we target the SCC decomposition ordering rather than a total ranking or a single tournament solution set. To our knowledge, no existing algorithm addresses this combination.

## F.4. Technical Overview and Roadmap

The appendix provides full proofs of correctness and termination for the algorithms introduced in the main text. The central question throughout is: *given only the edges revealed so far, when can we certify that the current top-$m$ vertices are correct with respect to the unknown tournament $G^*$?* We answer this first for transitive tournaments, then lift the answer to arbitrary tournaments.

**Two-layer architecture.** The proofs separate cleanly into two layers, and keeping this separation in mind is a useful guide for reading the appendix.

The first layer is a *finalization engine* that works on any directed graph and is completely agnostic to the existence of an underlying tournament. It takes a partially revealed graph, sorts vertices by in-reach into a rank spectrum, and identifies a prefix of vertices whose relative positions are already forced by reachability alone. Informally, this layer answers: *given what we have seen so far, which vertices can we already certify as top-ranked?* The most interesting pieces apply to DAGs (subgraphs of transitive tournaments). In the general case, we will utilize condensation machinery (c.f. §G.2) to reduce subgraphs of non-transitive tournaments to DAGs to lift the finalization engine to general directed graphs.

The second layer is a *tournament bridge* that connects the finalization engine back to the ground truth. It uses properties specific to the underlying tournament to show that whenever the engine certifies a prefix in the revealed graph, the same prefix is correct in the full tournament. Informally: *why does certification in the subgraph imply correctness in $G^*$?*

**Transitive versus non-transitive case.** In the transitive case, the revealed graph $G_t$ at a round $t$ is always a DAG (every subgraph of a transitive tournament is acyclic), so the finalization engine is applied directly to $G_t$ and the bridge bounds discovered in-reach by true in-degree.

In the general case, $G_t$ may contain cycles, so the engine cannot be applied to $G_t$ directly. Instead, we move to the condensation $[G_t]$, which collapses cycles into tied tiers and is always a DAG. The finalization engine is reused without modification on $[G_t]$, but the bridge must be rebuilt: a projection map induced by SCC refinement relates $[G_t]$ to $[G^*]$ and transfers rank certificates from the discovered graph to the true tournament.

**Roadmap.** The appendix proceeds as follows.

- **Section G** develops the reusable graph-theoretic machinery: the rank spectrum, known relationships, condensation, and the finalization engine. Nothing here assumes a tournament.
- **Section H** builds the tournament bridge for transitive tournaments, derives the greedy query schedule, and proves correctness and termination of the transitive algorithm.
- **Section I** lifts the same engine to condensation graphs, rebuilds the bridge via SCC refinement and projection, and proves correctness and termination for arbitrary tournaments.
- **Section J** introduces the resolution-based stopping rule used by the practical algorithm (BLITZRANK – Algorithm 1) and proves its correctness and termination.
- **Section K** discusses query complexity bounds.

Readers interested primarily in the algorithm may begin with Section H and return to the non-transitive extension afterward; Section G serves as a reference for both.

## G. Finalization Framework

This section builds the graph-theoretic objects that constitute the finalization engine described in the roadmap (§F.4). Nearly everything here is defined for arbitrary directed graphs; the few results that specialize to tournaments are marked. The section is organized in four parts:

- **Reachability framework** (§G.1.1): transitive closure, in-reach, and the rank spectrum.
- **Condensation** (§G.2): strongly connected components, the condensation graph, and their interaction with in-reach.
- **Known relationships** (§G.3): the resolved-vertex notion used by the practical algorithm's stopping rule.
- **Finalization engine** (§G.4.2): the finalization threshold, finalized set, and their structural properties.

## G.1. Reachability Framework

**Transitive closure.** Given a graph $G = (V, E)$, its transitive closure $G^+ = (V, E^+)$ is constructed by adding an edge $(u, v)$ to $E^+$ if $u \rightsquigarrow_G v$ i.e.,

$$E^+ := \{(u, v) \in V \times V \mid u \neq v \text{ and } u \rightsquigarrow_G v\}.$$

Some observations:

1. Under transitive closure, in-reach and out-reach reduce to in-neighborhood and out-neighborhood, respectively.
2. If $G$ is a DAG, then $R_G^-(v) \cap R_G^+(v) = \emptyset$. Otherwise, the intersection is not necessarily empty if $G$ is not a DAG.

**Lemma 3** (Downward Closure of In-Reach). *Let $G = (V, E)$ be a directed graph. If $x \in R_G^-(y)$, then $R_G^-(x) \cup \{x\} \subseteq R_G^-(y)$.*

*Proof.* Let $z \in R_G^-(x)$. By definition, $z \rightsquigarrow_G x$. Since $x \in R_G^-(y)$, we have $x \rightsquigarrow_G y$. Concatenating these paths gives $z \rightsquigarrow_G y$, so $z \in R_G^-(y)$. $\square$

The in-reach has useful properties when the graph $G$ under consideration is a directed acyclic graph (DAG).

**Lemma 4** (Strict Decrease of In-Reach Along Edges). *Let $G = (V, E)$ be a DAG. If $x \in R_G^-(y)$, then $|R_G^-(x)| < |R_G^-(y)|$.*

*Proof.* By Lemma 3, $R_G^-(x) \subseteq R_G^-(y)$. Since $x \in R_G^-(y)$ but $x \notin R_G^-(x)$ and $y \notin R_G^-(x)$ (due to otherwise being cyclic), the inclusion is strict. $\square$

### G.1.1. RANK SPECTRUM

We now define the central objects of our framework. By analogy with the singular value decomposition, where the singular values $\sigma_1(A) \geq \sigma_2(A) \geq \cdots$ are uniquely determined while their associated singular vectors may admit rotational freedom within tied singular subspaces, we pair in-reach values with their corresponding vertices. The rank spectrum $R(G)$ is uniquely determined regardless of tie-breaking, while the rank basis $V(G)$ is unique only up to permutation of vertices sharing the same in-reach.

*Definition* 5 (Rank Spectrum). Let $G = (V, E)$ be a directed graph on $n$ vertices. An *in-reach ordering* of $G$ is a sequence $v^{(1)}(G), v^{(2)}(G), \ldots, v^{(n)}(G)$ of all vertices in $V$ satisfying

$$r^{(1)}(G) \leq r^{(2)}(G) \leq \cdots \leq r^{(n)}(G), \quad \text{where } r^{(i)}(G) := |R_G^-(v^{(i)}(G))|.$$

We call the sequence $R(G) := (r^{(1)}(G), \ldots, r^{(n)}(G))$ the *rank spectrum* of $G$. Similarly, we call the sequence $V(G) = (v^{(1)}(G), \ldots, v^{(n)}(G))$ the *rank basis* of $G$. When ties exist, the ordering among tied vertices may be broken arbitrarily; the rank spectrum $R(G)$ is uniquely determined regardless of tie-breaking.

The rank spectrum forms crucial connections from subgraphs of tournaments to the true underlying tournament.

**Lemma 6** (Rank Spectrum Upper Bound for DAGs). *Let $G = (V, E)$ be a DAG on $n$ vertices. Then*

$$r^{(i)}(G) \leq i - 1, \quad \text{for all } i \in [n].$$

*Proof.* Let $v = v^{(i)}(G)$ with in-reach $r := r^{(i)}(G) = |R_G^-(v)|$. By Lemma 4, every vertex in $R_G^-(v)$ has in-reach strictly less than $r$. Since $|R_G^-(v)| = r$, at least $r$ vertices have in-reach less than $r$, so $v$ occupies position $i \geq r + 1$ in the non-decreasing in-reach ordering. Hence $r^{(i)}(G) = r \leq i - 1$. $\square$

## G.2. Condensation Graphs

For non-transitive tournaments, condensation plays a crucial role as in Section I, where it allows reduction from the general directed graph case to the DAG/transitive case by collapsing cycles into equivalence classes.

**Strongly Connected Components (SCC).** Let $G = (V, E)$ be a directed graph. A *strongly connected component* (SCC) of $G$ is a maximal subset $C \subseteq V$ such that for all $u, v \in C$, there exist directed paths $u \rightsquigarrow_G v$ and $v \rightsquigarrow_G u$. We define the equivalence relation[3] $\sim_G$ on $V$ by

$$u \sim_G v \iff u \text{ and } v \text{ are in the same SCC of } G.$$

For $u \in V$, we denote by $[u]_G$ the equivalence class of $u$ under $\sim_G$.

This equivalence is important in the context of Problem 1 where equivalence implies equal in-reach.

**Lemma 7** (Equal In-Reach Within SCCs). *Let $G = (V, E)$ be a directed graph. If $u \sim_G v$, then $|R_G^-(u)| = |R_G^-(v)|$.*

*Proof.* Since $u \sim_G v$, we have $u \rightsquigarrow_G v$ and $v \rightsquigarrow_G u$. For any $w \in V \setminus \{u, v\}$:

$$w \in R_G^-(u) \iff w \rightsquigarrow_G u \iff w \rightsquigarrow_G v \iff w \in R_G^-(v),$$

where the middle equivalence follows by concatenating with the path $u \rightsquigarrow_G v$ or $v \rightsquigarrow_G u$, respectively. Hence $R_G^-(u) \setminus \{v\} = R_G^-(v) \setminus \{u\}$, which gives $|R_G^-(u)| = |R_G^-(v)|$. $\square$

*Definition* 8 (Condensation Graph). The *condensation graph* of a directed graph $G$, denoted $[G]$, is the directed graph with vertex set $[V]_G := V / \sim_G$ and edge set

$$[E]_G := \{([u]_G, [v]_G) \mid \exists x \in [u]_G, y \in [v]_G \text{ such that } (x, y) \in E \text{ and } [u]_G \neq [v]_G\}.$$

We will also omit the subscript $G$ and use $[V]$, $[E]$, and $[u] \in [V]$ when it is clear from the context.

**Lemma 9** (Reachability Respects Condensation). *Let $G = (V, E)$ be a directed graph. Let $C, D \in [V]_G$ be distinct vertices in $[G]$. Then*

$$C \rightsquigarrow_{[G]} D \iff u \rightsquigarrow_G v, \quad \text{for all } u \in C \text{ and } v \in D.$$

*Proof.* ($\Leftarrow$) Fix any $u \in C$ and $v \in D$ with $u \rightsquigarrow_G v$. The path from $u$ to $v$ passes through a sequence of SCCs, inducing a path from $C$ to $D$ in $[G]$ by definition of the condensation edges.

($\Rightarrow$) Let $C = C_0 \to C_1 \to \cdots \to C_\ell = D$ be a path in $[G]$. For each edge $(C_i, C_{i+1}) \in [E]_G$, there exist $x_i \in C_i$ and $y_i \in C_{i+1}$ with $(x_i, y_i) \in E$. Since $C_{i+1}$ is a strongly connected component, $y_i$ and $x_{i+1}$ are mutually reachable within $C_{i+1}$, so $y_i \rightsquigarrow_G x_{i+1}$. Now let $u \in C$ and $v \in D$ be arbitrary. Since $u, x_0 \in C_0$ and $x_\ell, v \in C_\ell$, mutual reachability within these SCCs gives $u \rightsquigarrow_G x_0$ and $x_\ell \rightsquigarrow_G v$. Concatenating:

$$u \rightsquigarrow_G x_0 \to y_0 \rightsquigarrow_G x_1 \to y_1 \rightsquigarrow_G \cdots \rightsquigarrow_G x_\ell \to y_\ell \rightsquigarrow_G v. \qquad \square$$

The condensation of a directed graph is always a DAG (Lemma 10). When the graph is a tournament, the condensation is moreover a transitive tournament (Proposition 11), which pins down the in-reach of every vertex (Lemma 12). These two tournament-specific results are included here because they complete the condensation picture and are referenced by multiple later sections.

**Lemma 10** (Condensation is a DAG). *For any directed graph $G$, the condensation $[G]$ is a DAG.*

*Proof.* Suppose $[G]$ contains a cycle of distinct nodes $[u_1] \to [u_2] \to \cdots \to [u_\ell] \to [u_1]$ with $\ell \geq 2$. Then for any $x \in [u_1]$ and $y \in [u_2]$, we have paths $x \rightsquigarrow y$ and $y \rightsquigarrow x$ (via the cycle), implying $x \sim_G y$. This contradicts $[u_1] \neq [u_2]$. $\square$

**Proposition 11** (Condensation of Tournament is Transitive Tournament). *If $G^*$ is a tournament, then $[G^*]$ is a transitive tournament.*

*Proof.* Let $C_1, C_2$ be distinct SCCs of $G^*$. Since $G^*$ is a tournament, for any $u \in C_1$ and $v \in C_2$, either $(u, v) \in E^*$ or $(v, u) \in E^*$. We claim all edges between $C_1$ and $C_2$ go in the same direction. Suppose not: let $(u_1, v_1), (v_2, u_2) \in E^*$ with $u_1, u_2 \in C_1$ and $v_1, v_2 \in C_2$. Since $u_1 \sim_{G^*} u_2$ and $v_1 \sim_{G^*} v_2$, we have paths $u_2 \rightsquigarrow_{G^*} u_1$ and $v_1 \rightsquigarrow_{G^*} v_2$. Then

$$u_1 \to v_1 \rightsquigarrow_{G^*} v_2 \to u_2 \rightsquigarrow_{G^*} u_1$$

forms a cycle containing both $u_1$ and $v_1$, contradicting $C_1 \neq C_2$. Thus $[G^*]$ is a tournament. Furthermore, since every acyclic tournament is transitive, $[G^*]$ is a transitive tournament. $\square$

---

[3]It is well-known that SCC form an equivalence relation. See for example Wikipedia's SCC page.

**Lemma 12** (In-Reach Within an SCC). *Let $G^* = (V, E^*)$ be a tournament with condensation SCCs $C_1, \ldots, C_L$ in condensation order. Then for any $v \in C_j$,*

$$|R_{G^*}^-(v)| = \sum_{i=1}^{j} |C_i| - 1.$$

*In particular, all vertices in the same SCC share the same in-reach, and the common in-reach is strictly increasing in $j$.*

*Proof.* Since $C_j$ is strongly connected, every vertex in $C_j \setminus \{v\}$ reaches $v$, contributing $|C_j| - 1$. For $i < j$, the condensation path $C_i \rightsquigarrow_{[G^*]} C_j$ and Lemma 9 imply every vertex in $C_i$ reaches $v$, contributing $|C_i|$. For $i > j$, the acyclicity of $[G^*]$ implies no vertex in $C_i$ reaches $v$. Summing gives the result. Strict monotonicity in $j$ follows from $|C_j| \geq 1$. $\qquad \square$

### G.3. Known Relationships

Here, we introduce the *known relationship* set, which provides a simpler finalization criterion used in Algorithm 1.

*Definition* 13 (Known relationships). The *known relationship set* of $v$ in $G$ is

$$K_G(v) := R_G^-(v) \cup R_G^+(v), \tag{8}$$

and we write

$$\kappa_G(v) := |K_G(v)|$$

for its cardinality.

A vertex $v$ is *resolved* in $G$ if $\kappa_G(v) = n - 1$, i.e., every other vertex is reachable from or to $v$.

The known relationship has an interesting connection to the condensation graph.

**Lemma 14** (Known Relationships Lift Between Graph and Condensation). *Let $G = (V, E)$ be a directed graph with $n$ vertices. Let $[G]$ be its condensation with $n' := |[V]_G|$ vertices. For any SCC $C \in [V]_G$, the following are equivalent:*

1. *$\kappa_{[G]}(C) = n' - 1$,*
2. *$\kappa_G(u) = n - 1$ for all $u \in C$,*
3. *$\kappa_G(u) = n - 1$ for some $u \in C$.*

*Proof.* **(i)⇒(ii).** Let $u \in C$ and $v \in V \setminus \{u\}$. If $v \in C$, then $u \sim_G v$, so $v \in R_G^-(u) \cap R_G^+(u)$. If $v \in D$ for some SCC $D \neq C$, then $\kappa_{[G]}(C) = n' - 1$ implies $D \in R_{[G]}^-(C) \cup R_{[G]}^+(C)$. Lemma 9 gives $v \rightsquigarrow_G u$ or $u \rightsquigarrow_G v$, so $v \in R_G^-(u) \cup R_G^+(u)$. Since $v$ was arbitrary, $\kappa_G(u) = n - 1$.

**(ii)⇒(iii).** Immediate (since $C \neq \emptyset$).

**(iii)⇒(i).** Let $u \in C$ satisfy $\kappa_G(u) = n - 1$. Take any SCC $D \neq C$ and pick any $v \in D$. Since $v \in R_G^-(u) \cup R_G^+(u)$, either $v \rightsquigarrow_G u$ or $u \rightsquigarrow_G v$. By Lemma 9, either $D \rightsquigarrow_{[G]} C$ or $C \rightsquigarrow_{[G]} D$, so $D \in R_{[G]}^-(C) \cup R_{[G]}^+(C)$. Since $D$ was arbitrary, $\kappa_{[G]}(C) = n' - 1$. $\qquad \square$

*Remark* 15. The equivalence (ii)⇔(iii) says that resolution is an *all-or-nothing* property within an SCC: if any single vertex in $C$ has all its relationships determined, then every vertex in $C$ does. This is immediate from the proof, but can also be seen directly: for $u, w \in C$, mutual reachability gives $R_G^-(u) \cup R_G^+(u) \supseteq \left( R_G^-(w) \cup R_G^+(w) \right) \setminus \{u\}$, since any vertex reaching $w$ also reaches $u$ (via $w \rightsquigarrow u$), and any vertex reachable from $w$ is reachable from $u$ (via $u \rightsquigarrow w$).

### G.4. Finalization Engine

We now introduce machinery for measuring how much of the ranking has been determined in a directed graph. The definitions below apply to any directed graph; they will later be instantiated with the revealed subgraphs arising from oracle queries.

*Definition* 16 (Finalization Threshold and Finalized Set). Let $G = (V, E)$ be a directed graph on $n$ vertices with rank spectrum $R(G) = (r^{(1)}(G), \ldots, r^{(n)}(G))$ and corresponding rank basis $(v^{(1)}(G), \ldots, v^{(n)}(G))$.

1. The *finalization threshold* $m(G)$ is the largest $j \geq 0$ such that:

$$\text{(i)} \quad r^{(i)}(G) = i - 1 \text{ for all } i = 1, \ldots, j,$$
$$\text{(ii)} \quad j \in \{0, n\} \text{ or } r^{(j)}(G) < r^{(j+1)}(G). \tag{9}$$

2. The *finalized set* contains the finalized vertices:

$$\text{TOP}(G) := \{v^{(1)}(G), \ldots, v^{(m(G))}(G)\}. \tag{10}$$

3. The *candidate set* is the complement of the finalized set:

$$\text{CAND}(G) := V \setminus \text{TOP}(G) = \{v^{(m(G)+1)}(G), \ldots, v^{(n)}(G)\}. \tag{11}$$

Informally, $m(G)$ measures how much of the rank spectrum has "crystallized" into the pattern $0, 1, 2, \ldots$ that characterizes a fully discovered transitive tournament. The finalized set $\text{TOP}(G)$ contains the vertices whose positions in this prefix are fully determined.

### G.4.1. GENERAL PROPERTIES OF $\text{TOP}(G)$

Note that $v^{(j)}(G) \in \text{TOP}(G)$ if and only if $j \leq m(G)$. We establish some properties about nodes in $\text{TOP}(G)$ for general directed graphs.

**Lemma 17** (Exact In-Reach of Top Vertices). *Let $G = (V, E)$ be a directed graph. If $m(G) \geq 1$, we have $R_G^-(v^{(1)}(G)) = \emptyset$ and for $1 < j \leq m(G)$, we have*

$$R_G^-(v^{(j)}(G)) = \{v^{(1)}(G), \ldots, v^{(j-1)}(G)\}.$$

*Proof.* Let $j \leq m(G)$ and write $v^{(j)} = v^{(j)}(G)$ for brevity. The $j = 1$ case is trivial. Consider $j \geq 2$. By the finalization threshold, $|R_G^-(v^{(j)})| = r^{(j)}(G) = j - 1$.

Let $u \in R_G^-(v^{(j)})$. By Lemma 3, $R_G^-(u) \subseteq R_G^-(v^{(j)})$, so $|R_G^-(u)| \leq j - 1$. Since the finalization threshold requires the top $j$ vertices to have distinct in-reach sizes and $v^{(j)}$ already has in-reach $j - 1$, we must have strict inequality. Hence, $|R_G^-(u)| \leq j - 2$, which places $u \in \{v^{(1)}, \ldots, v^{(j-1)}\}$. This means that $R_G^-(v^{(j)}) \subseteq \{v^{(1)}, \ldots, v^{(j-1)}\}$. Both sets have cardinality $j - 1$, so they are equal. $\square$

**Corollary 18** (Reachability of Top Vertices). *Let $G = (V, E)$ be a directed graph. If $1 \leq i < j \leq m(G)$, then we have*

$$v^{(i)}(G) \rightsquigarrow_G v^{(j)}(G).$$

*Proof.* Write $v^{(j)} = v^{(j)}(G)$ for brevity. By Lemma 17, $R_G^-(v^{(j)}) = \{v^{(1)}, \ldots, v^{(j-1)}\}$. Since $i < j$, we have $v^{(i)} \in \{v^{(1)}, \ldots, v^{(j-1)}\} = R_G^-(v^{(j)})$, which by definition means $v^{(i)} \rightsquigarrow_G v^{(j)}$. $\square$

**Lemma 19** (Vertices Outside of TOP Have Large In-Reach). *Let $G = (V, E)$ be a directed graph. If $u \notin \text{TOP}(G)$, then $|R_G^-(u)| \geq m(G)$.*

*Proof.* Note that this trivially holds if $m(G) = 0$. Consider $m(G) > 0$. Let $m := m(G)$ for short and let $u := v^{(k)}(G)$ for some $k > m$, so that $u \notin \text{TOP}(G)$. From the finalization criterion (Definition 16), we have $r^{(m)}(G) = m - 1$ and $r^{(m)}(G) < r^{(m+1)}(G)$. Since the rank spectrum is non-decreasing and $k > m$:

$$|R_G^-(u)| = r^{(k)}(G) \geq r^{(m+1)}(G) > r^{(m)}(G) = m - 1.$$

Since in-reach sizes are integers, $|R_G^-(u)| \geq m = m(G)$. $\square$

The next Lemma requires $G$ to be a DAG.

**Lemma 20** (Top Vertices Reach All Non-Top Vertices). *Let $G = (V, E)$ be a DAG. For any $w \in \text{TOP}(G)$ and any $u \in \text{CAND}(G)$, we have $w \rightsquigarrow_G u$.*

*Proof.* Let $u \in \text{CAND}(G) \implies u \notin \text{TOP}(G)$. We construct a sequence $(x_i)_{i \geq 0}$ iteratively from $u$ to a node to $\text{TOP}(G)$:

- Set $x_0 = u$.
- While $R_G^-(x_i) \cap \mathrm{CAND}(G) \neq \emptyset$, set $x_{i+1} \in R_G^-(x_i) \cap \mathrm{CAND}(G)$. Note that $x_{i+1} \rightsquigarrow x_i \rightsquigarrow x_{i-1} \rightsquigarrow \cdots \rightsquigarrow x_0 = u$.
- We terminate the sequence when $R_G^-(x_i) \cap \mathrm{CAND}(G) = \emptyset$.

Each $x_i$ lies in $\mathrm{CAND}(G) = V \setminus \mathrm{TOP}(G)$, so by Lemma 19:

$$|R_G^-(x_i)| \geq m(G), \quad \text{for all } i.$$

By Lemma 4, since $G$ is a DAG by assumption, whenever $x_{i+1}$ exists:

$$|R_G^-(x_{i+1})| < |R_G^-(x_i)|.$$

The in-reach sizes form a strictly decreasing sequence of non-negative integers bounded below by $m(G)$. Therefore, the sequence must terminate at some finite index $k$ and each element $x_i$ for $i = 0, \ldots, k$ is unique. We show that at termination, the in-reach of the termination node $x_k$ equals $\mathrm{TOP}(G)$ i.e., $R_G^-(x_k) = \mathrm{TOP}(G)$.

At termination time $k$, we have $R_G^-(x_k) \cap \mathrm{CAND}(G) = \emptyset$. That implies

$$R_G^-(x_k) \subseteq \mathrm{TOP}(G).$$

Since $x_k \in \mathrm{CAND}(G) \implies x_k \notin \mathrm{TOP}(G)$, Lemma 19 gives $|R_G^-(x_k)| \geq m(G)$. Combined with $|\mathrm{TOP}(G)| = m(G)$ and the inclusion above implies:

$$R_G^-(x_k) = \mathrm{TOP}(G).$$

Since $x_{i+1} \in R_G^-(x_i)$ for all $i = 0, 1, \ldots, k-1$, by Lemma 3, we have $R_G^-(x_{i+1}) \subseteq R_G^-(x_i)$. Applying this repeatedly:

$$\mathrm{TOP}(G) = R_G^-(x_k) \subseteq R_G^-(x_{k-1}) \subseteq \cdots \subseteq R_G^-(x_0) = R_G^-(u).$$

Hence, for any $w \in \mathrm{TOP}(G)$, we have $w \in R_G^-(u)$, which by definition means $w \rightsquigarrow_G u$. □

*Remark* 21. Unlike most results in this section, the DAG assumption in Lemma 20 is essential. Cycles can inflate the in-reach of vertices outside $\mathrm{TOP}(G)$ without any connection to the finalized vertices. For instance, consider $V = \{a, b, c\}$ with only edges $b \to c$ and $c \to b$. Then $\mathrm{TOP}(G) = \{a\}$, yet $a$ reaches neither $b$ nor $c$: the cycle between $b$ and $c$ alone is responsible for their in-reach meeting the threshold $m(G) = 1$.

### G.4.2. STRUCTURAL PROPERTIES OF THE FINALIZATION THRESHOLD

The following lemmas establish structural properties of the finalization threshold.

**Lemma 22** (Tied Candidates After Finalization Threshold). *Let $G = (V, E)$ be a DAG on $n$ vertices. Let $m' := m(G)$ be the finalization threshold of $G$. If $0 < m' \leq n - 2$, then*

$$r^{(m'+1)}(G) = r^{(m'+2)}(G).$$

*Proof.* Suppose for contradiction that $r^{(m'+1)}(G) < r^{(m'+2)}(G)$. Since $0 < m' \leq n - 2$, the finalization conditions give $r^{(m')}(G) = m' - 1$ and $r^{(m')}(G) < r^{(m'+1)}(G)$. By Lemma 6, $r^{(m'+1)}(G) \leq m'$. Hence

$$m' - 1 = r^{(m')}(G) < r^{(m'+1)}(G) \leq m',$$

forcing $r^{(m'+1)}(G) = m'$. Thus $r^{(i)}(G) = i - 1$ holds for all $i \leq m' + 1$, and the strict inequality $r^{(m'+1)}(G) < r^{(m'+2)}(G)$ satisfies condition (9)(ii) for $j = m' + 1$, contradicting the maximality of $m'$. □

Tied candidates have a useful structural property: no edge can exist between two vertices with the same in-reach in a DAG.

**Lemma 23** (Tied Candidates Have Unknown Edge). *Let $G = (V, E)$ be a DAG. If $u, v \in V$ are distinct nodes satisfying $|R_G^-(u)| = |R_G^-(v)|$, then neither $(u, v)$ nor $(v, u)$ belong to $E$.*

*Proof.* WLOG, suppose for contradiction that $(u, v) \in E$. Thus, $u \in R_G^-(v)$ which implies $|R_G^-(u)| < |R_G^-(v)|$ by Lemma 4. This is a contradiction to $|R_G^-(u)| = |R_G^-(v)|$. □

**Lemma 24** (No Penultimate Finalization). *Let $G = (V, E)$ be a directed graph on $n \geq 2$ vertices. Let $m' := m(G)$ be the finalization threshold of $G$. Then*

$$m' \neq n - 1.$$

*Consequently, $m' \in \{0, 1, \ldots, n-2, n\}$.*

*Proof.* Suppose for contradiction that $m' = n - 1$. Since $n \geq 2$, we have $n - 1 \notin \{0, n\}$, so Definition 16 requires:

1. $r^{(i)}(G) = i - 1$ for all $i \in [n-1]$, and
2. $r^{(n-1)}(G) < r^{(n)}(G)$.

From (i), $r^{(n-1)}(G) = n - 2$, so (ii) gives $r^{(n)}(G) \geq n - 1$. Since $r^{(n)}(G) = |R_G^-(v^{(n)}(G))| \leq |V \setminus \{v^{(n)}(G)\}| = n - 1$, we obtain $r^{(n)}(G) = n - 1$. Hence $r^{(i)}(G) = i - 1$ for all $i \in [n]$, and $j = n$ satisfies the finalization conditions in (9) (since $n \in \{0, n\}$), contradicting the maximality of $m' = n - 1$. $\qquad\square$

# H. Transitive Tournaments

We analyze Problem 1 for the special case of *transitive tournaments*: A graph $G^* = (V, E^*)$ is a transitive tournament if it is a tournament and it is transitive i.e., if $(u, v) \in E^*$ and $(v, w) \in E^*$, then $(u, w) \in E^*$. We provide an algorithm for this special case with correctness guarantees.

## H.1. Algorithm Framework

We adopt an iterative approach: at each round, we query a carefully chosen subset of vertices, update our knowledge of the graph, and terminate once we have sufficient information to identify the top-$m$ vertices.

**Notation.** At each round $t = 1, 2, \ldots, T$:

- $Q_t \in \binom{V}{k}$ denotes the queried $k$-subset.
- $\hat{E}_t := O_{G^*}(Q_t)$ denotes the edges revealed by the oracle.
- $E_t := E_{t-1} \cup \hat{E}_t$ denotes the cumulative set of revealed edges, with $E_0 := \emptyset$.
- $G_t := (V, E_t)$ denotes the *revealed graph* at round $t$.

By construction, $\emptyset = E_0 \subseteq E_1 \subseteq \cdots \subseteq E_T \subseteq E^*$. A good algorithm should ideally produce a strictly increasing chain.

The rank spectrum (Definition 5) of the revealed graph at round $t$ is the main object of interest.

**Discovered Rankings.** At round $t$, we order the vertices by non-decreasing in-reach in $G_t$, breaking ties arbitrarily:

$$v_t^{(1)}, v_t^{(2)}, \ldots, v_t^{(n)} \quad \text{with} \quad r_t^{(1)} \leq r_t^{(2)} \leq \cdots \leq r_t^{(n)},$$

where $r_t^{(i)} := |R_{G_t}^-(v_t^{(i)})|$. The sequence $R_t := (r_t^{(1)}, \ldots, r_t^{(n)})$ is the *discovered rank sequence* at round $t$. We use these as notational shorthands for Definition 16, where

$$v_t^{(i)} := v^{(i)}(G_t) \quad \text{and} \quad r_t^{(i)} := r^{(i)}(G_t).$$

Initially, $R_0 = (0, 0, \ldots, 0)$ since no edges have been revealed.

## H.2. Basic Facts about Transitive Tournaments

We review basic facts about transitive tournaments here.

*Remark* 25 (Transitive Tournaments are Acyclic). A transitive tournament contains no directed cycles. Indeed, if $v_1 \to v_2 \to \cdots \to v_\ell \to v_1$ were a cycle with $\ell \geq 2$, transitivity would give $(v_1, v_\ell) \in E^*$, while the cycle requires $(v_\ell, v_1) \in E^*$, contradicting the tournament property. In particular, every transitive tournament is a DAG.

**Lemma 26** (Transitive Tournaments are their own Transitive Closure). *Let $G^*$ be a transitive tournament. Then $(G^*)^+ = G^*$; that is, $u \rightsquigarrow_{G^*} v$ if and only if $(u, v) \in E^*$.*

*Proof.* The backward direction is immediate. For the forward direction, suppose $u \rightsquigarrow_{G^*} v$ via a path $u = w_0 \to w_1 \to \cdots \to w_\ell = v$. Repeated application of transitivity $((w_0, w_1) \in E^*$ and $(w_1, w_2) \in E^*$ give $(w_0, w_2) \in E^*$; then $(w_0, w_2)$ and $(w_2, w_3)$ give $(w_0, w_3)$; and so on) yields $(u, v) \in E^*$. $\qquad\square$

**Corollary 27** (In-Reach Equals In-Neighborhood). *In a transitive tournament $G^*$,*

$$R_{G^*}^-(v) = N_{G^*}^-(v) := \{u \in V \setminus \{v\} : (u, v) \in E^*\}$$

*for every vertex $v$. In particular, $|R_{G^*}^-(v)| = \deg_{G^*}^-(v)$.*

*Proof.* Immediate from Lemma 26: $u \in R_{G^*}^-(v)$ iff $u \rightsquigarrow_{G^*} v$ iff $(u, v) \in E^*$ iff $u \in N_{G^*}^-(v)$. □

**Lemma 28** (Strict In-Degree Separation in Transitive Tournaments). *Let $G^*$ be a transitive tournament. If $(u, v) \in E^*$, then $\deg_{G^*}^-(v) \geq \deg_{G^*}^-(u) + 1$. In particular, no two vertices share the same in-degree.*

*Proof.* Suppose $(u, v) \in E^*$. For any in-neighbor $w$ of $u$, we have $(w, u) \in E^*$ and $(u, v) \in E^*$, so transitivity gives $(w, v) \in E^*$. Hence $N_{G^*}^-(u) \subseteq N_{G^*}^-(v)$. Since $u \in N_{G^*}^-(v) \setminus N_{G^*}^-(u)$, the inclusion is strict, giving $\deg_{G^*}^-(v) \geq \deg_{G^*}^-(u) + 1$.

For uniqueness: given any two distinct vertices $u \neq v$, the tournament property forces $(u, v) \in E^*$ or $(v, u) \in E^*$, so the above gives $\deg_{G^*}^-(u) \neq \deg_{G^*}^-(v)$. □

The following is a classical result; see, e.g., Chapter 1 of (Moon, 2015).

**Corollary 29** (In-Degree Sequence of Transitive Tournaments). *A transitive tournament on $n$ vertices has in-degree multiset $\{0, 1, \ldots, n-1\}$.*

*Proof.* By Lemma 28, the $n$ in-degrees are pairwise distinct. Since each in-degree lies in $\{0, 1, \ldots, n-1\}$ by the tournament property, the in-degrees must be exactly $\{0, 1, \ldots, n-1\}$. □

### H.3. Rank Spectrum of Transitive Tournaments

The rank spectrum (Definition 5) of a transitive tournament admits some special properties.

**Proposition 30** (Rank Spectrum of Transitive Tournaments). *Let $G^*$ be a transitive tournament of $n$ nodes. We have:*

$$r^{(i)}(G^*) = i - 1, \quad for\ i = 1, 2, \ldots, n.$$

*Proof.* By Corollary 27, $|R_{G^*}^-(v)| = \deg_{G^*}^-(v)$ for every $v$. By Corollary 29, the in-degrees are exactly $\{0, 1, \ldots, n-1\}$. Sorting vertices by non-decreasing in-reach (Definition 5) therefore gives $r^{(i)}(G^*) = i - 1$. □

**Notation.** Since the transitive tournament $G^*$ is fixed throughout this section, we write

$$r_*^{(i)} := r^{(i)}(G^*) = i - 1 \quad \text{and} \quad v_*^{(i)} := v^{(i)}(G^*) \tag{12}$$

for the rank spectrum and rank basis of $G^*$, respectively.

The rank spectrum (Definition 5) allows us to make important connections between subgraphs (e.g. our discovered graph) of tournament graphs to the potential underlying unknown tournament graph.

**Lemma 31.** *Let $G^* = (V, E^*)$ be a transitive tournament. Let $G' = (V, E')$ be a subgraph of $G^*$ where $E' \subseteq E^*$. Then for all $v \in V$:*

$$|R_{G'}^-(v)| \leq \deg_{G^*}^-(v).$$

*Proof.* Since $E' \subseteq E^*$, we have $R_{G'}^-(v) \subseteq R_{G^*}^-(v)$ for all $v \in V$. Since $G^*$ is a transitive tournament, in-reach coincides with in-neighborhood by Corollary 27, hence $|R_{G^*}^-(v)| = \deg_{G^*}^-(v)$. Combining these inequalities yields the result:

$$\left|R_{G'}^-(v)\right| \leq \left|R_{G^*}^-(v)\right| = \deg_{G^*}^-(v). \qquad \square$$

Another connection we can make is by applying Lemma 6 to the rank spectrum of subgraphs of the transitive tournament.

**Corollary 32** (Rank Upper Bound for Subgraphs of Transitive Tournaments). *Let $G^* = (V, E^*)$ be a transitive tournament on $n$ vertices. Let $G' = (V, E')$ be a subgraph of $G^*$ where $E' \subseteq E^*$. Then*

$$r^{(i)}(G') \leq r_*^{(i)} = i - 1, \quad \text{for all } i \in [n].$$

*Proof.* By Remark 25, $G^*$ is acyclic. Since $E' \subseteq E^*$, the subgraph $G'$ is also acyclic. The bound $r^{(i)}(G') \leq i - 1$ then follows from Lemma 6, and $r_*^{(i)} = i - 1$ is Proposition 30. □

### H.4. Eliminations and Finalization

*Remark* 33 (A Filtering Criterion). Let $G^* = (V, E^*)$ be a transitive tournament. Let $G' = (V, E')$ be a subgraph of $G^*$ where $E' \subseteq E^*$. If a vertex $v \in V$ satisfies $|R_{G'}^-(v)| \geq m$, then $v$ is *not* in the final top $m$ vertices of $G^*$ i.e., $v \notin \{v_*^{(1)}, \ldots, v_*^{(m)}\}$.

*Proof.* In a transitive tournament on $n$ vertices, the true in-degrees are exactly $\{0, 1, \ldots, n - 1\}$, so $r_*^{(m)} = m - 1$ (Proposition 30). By Corollary 32, $|R_{G'}^-(v)| \leq \deg_{G^*}^-(v)$. Thus $|R_{G'}^-(v)| \geq m$ implies $\deg_{G^*}^-(v) \geq m > r_*^{(m)}$, so $v$ cannot be among the top $m$ vertices. □

We say that a vertex $v$ in a subgraph $G'$ of an underlying tournament $G^*$ is *finalized* when we have sufficient information to confirm $v$'s membership in the top-$m$ set of $G^*$. The following proposition characterizes when this occurs.

**Proposition 34** (Top-$j$ Finalization Criterion in Transitive Tournaments). *Let $G^* = (V, E^*)$ be a transitive tournament. Let $G' = (V, E')$ be a subgraph of $G^*$ where $E' \subseteq E^*$. Suppose there exists $j \in [n - 1]$ such that $r^{(1)}(G') < r^{(2)}(G') < \cdots < r^{(j)}(G') < r^{(j+1)}(G')$. Then the top $j$ rank spectrum of $G'$ aligns with $G^*$ i.e.,*

1. *$r^{(i)}(G') = i - 1$ for all $i \leq j$, and*
2. *$\left(v^{(1)}(G'), v^{(2)}(G'), \ldots, v^{(j)}(G')\right) = \left(v_*^{(1)}, v_*^{(2)}, \ldots, v_*^{(j)}\right)$.*

*Proof.* We prove (a) by induction. For the base case, by Corollary 32, $r^{(1)}(G') \leq r_*^{(1)} = 0$. Furthermore, $r^{(1)}(G') = |R_{G'}^-(v^{(1)}(G'))| \geq 0$ implies that $r^{(1)}(G') = 0$. For the inductive step, assuming $r^{(i-1)}(G') = i - 2$, the strict inequality $r^{(i)}(G') > r^{(i-1)}(G') = i - 2$ combined with $r^{(i)}(G') \leq r_*^{(i)} = i - 1$ from Corollary 32 forces $r^{(i)}(G') = i - 1$.

For (b), since $r^{(j+1)}(G') > r^{(j)}(G') = j - 1$ from (a), we have $r^{(j+1)}(G') \geq j$. All vertices $v^{(j+1)}(G'), \ldots, v^{(n)}(G')$ have discovered in-reach at least $r^{(j+1)}(G') \geq j$. Hence, vertices $v^{(j+1)}(G'), \ldots, v^{(n)}(G')$ cannot be among the top $j$ in $G^*$ following Proposition 30. This establishes the set equality. The ordering within the set follows by matching ranks. □

Proposition 34 provides the key bridge for transitive tournaments: whenever a strictly increasing prefix appears in the discovered rank spectrum, that prefix is correct with respect to the true tournament. Combined with the finalization threshold (Definition 16), which captures the maximal extent of this crystallization, this drives both the algorithm design and its termination analysis.

### H.5. A Greedy Algorithm for Ranking Transitive Tournaments

*Remark* 35. When $|\text{CAND}(G_t)| < k$, the padding vertices in $P_t \subseteq \text{TOP}(G_t)$ are already finalized; edges revealed involving $P_t$ are redundant by Corollary 18 and Lemma 20, so padding preserves correctness while satisfying the oracle's requirement that $|Q_t| = k$.

We now present Algorithm 2 that iteratively queries carefully and adaptively chosen sets of $k$ vertices until the top-$m$ vertices are finalized. The key insight is to always query candidates with the smallest discovered in-reach to guarantee that we always add an edge to make progress, where we terminate when there is enough information to finalize all top candidates. The correctness proof is presented in Theorem 37.

To establish correctness, we show that Algorithm 2 adds at least one new edge every round (for at most $\binom{n}{2}$ rounds), guaranteeing that we do not get stuck and that we terminate eventually when we have enough information.

**Theorem 36** (Termination of Algorithm 2). *Algorithm 2 terminates in at most $\binom{n}{2}$ rounds.*

---

**Algorithm 2** Greedy Tournament Sort (Transitive Case)

---

**Require:** Vertex set $V$ with $|V| = n$, oracle $O_{G^*}$ with query size $k$, target count $m \leq n$
**Ensure:** Top-$m$ vertices $(v_*^{(1)}, \ldots, v_*^{(m)})$ in ascending in-reach order
1: $E_0 \leftarrow \emptyset$, $G_0 \leftarrow (V, E_0)$, $t \leftarrow 0$
2: **while** true **do**
3:     Compute rank spectrum $R_t = (r_t^{(1)}, \ldots, r_t^{(n)})$ of $G_t$ per Definition 5
4:     Compute rank basis $(v_t^{(1)}, \ldots, v_t^{(n)})$ of $G_t$ per Definition 5
5:     Compute finalization threshold $m_t := m(G_t)$ per Definition 16
6:     **if** $m_t \geq m$ **then**
7:         {termination condition}
8:         **break**
9:     **end if**
10:    $k' \leftarrow \min(k, \ n - m_t)$ {$n - m_t \geq 2$ by Lemma 24}
11:    $S_t \leftarrow \{v_t^{(m_t+1)}, \ldots, v_t^{(m_t+k')}\}$ {$k'$ candidates with smallest in-reach}
12:    **if** $k' < k$ **then**
13:       Pad: $Q_t \leftarrow S_t \cup P_t$, where $P_t \subseteq \text{TOP}(G_t)$ with $|P_t| = k - k'$
14:    **else**
15:       $Q_t \leftarrow S_t$
16:    **end if**
17:    Query oracle: $\hat{E}_t \leftarrow O_{G^*}(Q_t)$
18:    Update: $E_{t+1} \leftarrow E_t \cup \hat{E}_t$, $G_{t+1} \leftarrow (V, E_{t+1})$
19:    $t \leftarrow t + 1$
20: **end while**
21: **return** $(v_t^{(1)}, \ldots, v_t^{(m)})$

---

*Proof.* At a non-terminal round $t$, we have $m_t < m \leq n$. By Lemma 24, $m_t \neq n - 1$, so $m_t \leq n - 2$ and thus $|\text{CAND}(G_t)| = n - m_t \geq 2$. We claim $r_t^{(m_t+1)} = r_t^{(m_t+2)}$. If $m_t > 0$, Lemma 22 implies that $r_t^{(m_t+1)} = r_t^{(m_t+2)}$. If $m_t = 0$, then the finalization condition $r_t^{(1)} < r_t^{(2)}$ fails, so $r_t^{(1)} = r_t^{(2)}$ (since ranks are non-decreasing). In any case, $r_t^{(m_t+1)} = r_t^{(m_t+2)}$ implies that $v_t^{(m_t+1)}$ and $v_t^{(m_t+2)}$ are tied candidates i.e., $|R_{G_t}^-(v_t^{(m_t+1)})| = |R_{G_t}^-(v_t^{(m_t+2)})|$. By Lemma 23, the edge between $v_t^{(m_t+1)}$ and $v_t^{(m_t+2)}$ is not in $E_t$. Since both vertices are in $Q_t$, querying the oracle reveals this edge, ensuring $|E_{t+1}| > |E_t|$. $\qquad\square$

The algorithm terminates and outputs something. We now prove that its output correctly solves Problem 1.

**Theorem 37** (Correctness of Algorithm 2). *Algorithm 2 correctly identifies the top-$m$ vertices of a transitive tournament graph $G^*$.*

*Proof.* At termination round $T$, $m_T \geq m$. We verify that the output $(v_T^{(1)}, \ldots, v_T^{(m)})$ satisfies the two conditions of Problem 1. Since $m_T \geq m \geq 1$, the finalization threshold (Definition 16) provides: (i) $r_T^{(i)} = i - 1$ for all $i \leq m_T$, and (ii) $m_T = n$ or $r_T^{(m_T)} < r_T^{(m_T+1)}$. Applying Proposition 34 with $j = m_T$ (using (i), (ii), and the fact that $G_T$ is a subgraph of the transitive tournament $G^*$) yields:

$$v_T^{(i)} = v_*^{(i)} \quad \text{for all } i \leq m_T. \tag{13}$$

In particular, the first $m \leq m_T$ output vertices satisfy $v_T^{(i)} = v_*^{(i)}$ for $i = 1, \ldots, m$. By Corollary 27, in-reach coincides with in-degree in a transitive tournament, and by Proposition 30, $|R_{G^*}^-(v_*^{(i)})| = i - 1$. Therefore:

***Internal ordering*** *(6).* For $i = 1, \ldots, m$:

$$|R_{G^*}^-(v_T^{(i)})| = |R_{G^*}^-(v_*^{(i)})| = i - 1,$$

which is strictly increasing in $i$.

***Rank dominance*** *(7)*. Let $S = \{v_T^{(1)}, \ldots, v_T^{(m)}\}$. The maximum in-reach within $S$ is $|R_{G^*}^-(v_T^{(m)})| = m - 1$. For any $u \notin S$, we have $u = v_*^{(j)}$ for some $j > m$ (by (13) and the uniqueness of rank basis in transitive tournaments), so $|R_{G^*}^-(u)| = j - 1 \geq m > m - 1$. Hence

$$\max_{i \leq m} |R_{G^*}^-(v_T^{(i)})| = m - 1 < m \leq \min_{u \notin S} |R_{G^*}^-(u)|. \qquad \square$$

## I. Non-Transitive Tournaments

We now generalize Algorithm 2 to arbitrary tournaments. When $G^*$ is not transitive, two difficulties arise.

1. **Finalization breaks down.** The finalization threshold (Definition 16) requires the rank spectrum to form a strictly increasing prefix $0, 1, 2, \ldots$ – the signature of a transitive tournament (Proposition 30). In a non-transitive tournament, vertices sharing an SCC have equal in-reach (Lemma 7 and 12), so ties are intrinsic and this prefix pattern need not appear in $G_t$ itself.
2. **Scheduling breaks down.** Algorithm 2 guarantees progress by querying tied candidates, which must have an unknown edge between them (Lemma 23). In the general case, tied vertices may already share a known edge via a cycle, so the same scheduling rule can stall.

Both difficulties are resolved by a single observation: the condensation $[G]$ of any directed graph is a DAG (Lemma 10), and the condensation of any tournament is a transitive tournament (Proposition 11). This lets us reuse the finalization engine from Section G without modification by applying it to $[G_t]$ rather than to $G_t$ directly.

The tournament bridge, however, must be rebuilt. In-reach in a subgraph $G_t$ no longer directly bounds in-degree in $G^*$, so we instead relate the two condensations $[G_t]$ and $[G^*]$ through the natural projection map induced by SCC refinement (Lemma 39). This projection preserves reachability (Lemma 41) and yields a new bridge lemma (Lemma 42) that transfers rank information from the revealed subgraph to the true tournament via the condensation. Adding edges can only merge SCCs and never split them, so the projection is monotone and the algorithm makes steady progress.

Algorithm 3 implements this strategy. The remainder of the section develops the necessary machinery: Section I.1 establishes SCC refinement and the projection map, Section I.2 connects the rank spectra of $G_t$ and $[G_t]$ to that of $G^*$, and Section I.3 proves termination and correctness.

*Remark* 38 (Equivalence for Transitive Case). When the underlying tournament $G^*$ is transitive, Algorithm 3 reduces to Algorithm 2. Specifically:

1. All SCCs are singletons: $[u]_{G_t} = \{u\}$ for all $u \in V$ and round $t$.
2. $[G_t] \cong G_t$ canonically.
3. In-reach in $[G_t]$ equals in-reach in $G_t$.

*Proof.* In a transitive tournament, there are no cycles, so each SCC is a singleton. The condensation is isomorphic to the original graph, and all operations coincide. $\qquad \square$

### I.1. SCC Refinement and Projection

**Lemma 39** (SCC Refinement). *Let $G = (V, E)$ be a directed graph. Let $G' = (V, E')$ be a subgraph of $G$ where $E' \subseteq E$. Then, we have*

$$u \sim_{G'} v \implies u \sim_G v.$$

*Equivalently, each SCC of subgraphs of $G$ is contained in some SCC of $G$ i.e.,*

$$G' \subseteq G \implies [u]_{G'} \subseteq [u]_G, \text{ for all } u \in V.$$

*Proof.* If $u \sim_{G'} v$, then there exist paths $u \rightsquigarrow_{G'} v$ and $v \rightsquigarrow_{G'} u$. Since $E' \subseteq E$, these paths exist in $G$, so $u \sim_G v$. $\qquad \square$

*Definition* 40 (Projection Map). Let $G = (V, E)$ be a directed graph. Let $G' = (V, E')$ be a subgraph of $G$ where $E' \subseteq E$. The refinement induces a natural projection $\phi_G : [V]_{G'} \to [V]_G$ defined by

$$\phi_G([u]_{G'}) = [u]_G.$$

---

**Algorithm 3** Tournament Sort General

---

**Require:** Vertex set $V$ with $|V| = n$, oracle $O_{G^*}$ with query size $k$, target count $m \leq n$
**Ensure:** A sequence of top-$m$ vertices $(v_*^{(1)}, \ldots, v_*^{(m)})$ in non-decreasing in-reach order.
 1: $E_0 \leftarrow \emptyset$, $G_0 \leftarrow (V, E_0)$, $t \leftarrow 0$
 2: **while** true **do**
 3:     Compute the condensation graph $[G_t] = ([V_t], [E_t])$ with $n_t := |[V_t]|$ per Definition 8.
 4:     Compute rank spectrum $\widetilde{R_t} := (r^{(1)}([G_t]), \ldots, r^{(n_t)}([G_t]))$ of $[G_t]$ per Definition 5
 5:     Compute the corresponding rank basis $(C_t^{(1)}, \ldots, C_t^{(n_t)})$ of $[G_t]$ given $\widetilde{R_t}$ per Definition 5 where $r^{(i)}([G_t]) := R_{[G_t]}^{-}\left(C_t^{(i)}\right)$ for $C_t^{(i)} \in [V_t]$.
 6:     Compute finalization threshold $\widetilde{m_t} := m([G_t])$ per (9).
 7:     Compute finalized SCCs $\widetilde{\text{TOP}}_t \leftarrow \left\{C_t^{(1)}, \ldots, C_t^{(\widetilde{m_t})}\right\} = \text{TOP}([G_t])$ per (10).
 8:     $\text{TOP}_t \leftarrow \bigcup_{i=1}^{\widetilde{m_t}} C_t^{(i)}$ {lift finalized nodes to graph}
 9:     **if** $|\text{TOP}_t| \geq m$ **then**
10:         **break** {we found enough top nodes}
11:     **end if**
12:     $k' \leftarrow \min(k, n_t - \widetilde{m_t})$
13:     $\widetilde{\text{CAND}}_t \leftarrow \{C_t^{(\widetilde{m_t}+1)}, \ldots, C_t^{(\widetilde{m_t}+k')}\}$ {$k'$ candidates with smallest in-reach}
14:     **if** $k' < k$ **then**
15:         Pad: $\widetilde{Q_t} \leftarrow \widetilde{\text{CAND}}_t \cup P_t$, where $P_t \subseteq \text{TOP}([G_t])$ with $|P_t| = k - k'$
16:     **else**
17:         $\widetilde{Q_t} \leftarrow \widetilde{\text{CAND}}_t$
18:     **end if**
19:     Pick representative: $Q_t \leftarrow \left\{\text{rep}(C) : C \in \widetilde{Q_t}\right\}$ where rep picks an element from a set.
20:     Query oracle: $\hat{E}_t \leftarrow O_{G^*}(Q_t)$
21:     Update: $E_{t+1} \leftarrow E_t \cup \hat{E}_t$, $G_{t+1} \leftarrow (V, E_{t+1})$
22:     $t \leftarrow t + 1$
23: **end while**
24: Let $j^* \leftarrow \min\{j \leq \widetilde{m_t} : \sum_{i=1}^{j} |C_t^{(i)}| \geq m\}$
25: Let $S_{j^*}$ be any subset of $C_t^{(j^*)}$ with $|S_{j^*}| = m - \sum_{i=1}^{j^*-1} |C_t^{(i)}|$
26: **return** the solution sequence $S := (v^{(1)}, \ldots, v^{(m)})$ obtained by concatenating $C_t^{(1)}, C_t^{(2)}, \ldots, C_t^{(j^*-1)}, S_{j^*}$ in this order.

---

This is well-defined by Lemma 39. We will often omit the subscript $G$ and domain definition when the domain and codomain are clear from the context.

**Lemma 41** (Path Preservation Under Projection). *Let $G = (V, E)$ be a directed graph. Let $G' = (V, E')$ be a subgraph of $G$ where $E' \subseteq E$. If $D \rightsquigarrow_{[G']} C$ in $[G']$, then either $\phi_G(D) = \phi_G(C)$ or $\phi_G(D) \rightsquigarrow_{[G]} \phi_G(C)$.*

*Proof.* Consider a path $D = D_0 \to D_1 \to \cdots \to D_\ell = C$ in $[G']$. For each edge $(D_i, D_{i+1}) \in [E']$, there exist $x_i \in D_i$, $y_i \in D_{i+1}$ with $(x_i, y_i) \in E' \subseteq E$.

If $\phi_G(D_i) = \phi_G(D_{i+1})$, the edge collapses to a single vertex.

If $\phi_G(D_i) \neq \phi_G(D_{i+1})$, then $[x_i]_G \neq [y_i]_G$, so $(\phi_G(D_i), \phi_G(D_{i+1})) \in [E]$.

Concatenating the non-collapsed edges yields either $\phi_G(D) = \phi_G(C)$ (if every edge along the path collapses) or a walk from $\phi_G(D)$ to $\phi_G(C)$ in $[G]$. Since $[G]$ is a DAG by Lemma 10, every walk between distinct vertices contains a path, so $\phi_G(D) \rightsquigarrow_{[G]} \phi_G(C)$. $\square$

## I.2. Rank Spectrum Under Condensation

**Lemma 42** (Discovered Reachability Respects True Rank). *Let $G = (V, E)$ be a directed graph. Let $G' = (V, E')$ be a subgraph of $G$ where $E' \subseteq E$. Let $C, D \in [V]_{G'}$ be distinct SCCs of $G'$. Then*

$$C \rightsquigarrow_{[G']} D \implies \left| R_G^-(u) \right| \leq \left| R_G^-(v) \right|, \quad \text{for all } u \in C \text{ and } v \in D,$$

*with equality if and only if $\phi_G(C) = \phi_G(D)$ (i.e. $u$ and $v$ belong to the same SCC of $G$).*

*Proof.* Let $C, D \in [V]_{G'}$ be distinct SCCs of $G'$ with $C \rightsquigarrow_{[G']} D$. Let $u \in C$ and $v \in D$. Lemma 41 implies that either $\phi_G(C) = \phi_G(D)$ or $\phi_G(C) \rightsquigarrow_{[G]} \phi_G(D)$.

In the first case, $[u]_G = [v]_G$. This means that they are in the same SCC i.e., $u \sim_G v$. Hence, Lemma (7) implies that $\left| R_G^-(u) \right| = \left| R_G^-(v) \right|$.

In the second case, we have $\phi_G(C) \neq \phi_G(D)$ and $\phi_G(C) \rightsquigarrow_{[G]} \phi_G(D)$. Since $[G]$ is a DAG (Lemma 10), there is no path from $[v]_G$ back to $[u]_G$, so $v \not\rightsquigarrow_G u$ and hence $v \notin R_G^-(u)$. Since $C \rightsquigarrow_{[G']} D$, there exists a path $u \rightsquigarrow_{G'} v$ via Lemma 9 when applied to $G'$. Since $u \rightsquigarrow_{G'} v$ and $E' \subseteq E$, we have $u \rightsquigarrow_G v$. Then by Lemma 3, we have $R_G^-(u) \subseteq R_G^-(v)$. Finally, $u \in R_G^-(v)$ but $v \notin R_G^-(u)$ as shown and $u \notin R_G^-(u)$ by definition implies $\left| R_G^-(u) \right| < \left| R_G^-(v) \right|$. The equality if and only if follows from this case. □

*Remark* 43. Lemma 42 is the key "bridge" between the discovered subgraphs ($G' = G_t$) in the algorithm and the true tournament ($G = G^*$).

Lemma 44 provides an analogous result to Lemma 20 for the condensation graph.

**Lemma 44** (Top Vertices Reach All Non-Top Vertices in Condensation Graphs). *Let $G = (V, E)$ be a directed graph. Let $G' = (V, E')$ be a subgraph of $G$ where $E' \subseteq E$. Then, for any $C \in \text{TOP}([G'])$ and $D \notin \text{TOP}([G'])$, we have*

$$\left| R_G^-(u) \right| \leq \left| R_G^-(v) \right|, \quad \text{for all } u \in C \text{ and } v \in D. \tag{14}$$

*Proof.* Since $[G']$ is a DAG, $C \rightsquigarrow_{[G']} D$ follows from 20. Equation (14) follows immediately from Lemma 42. □

**Lemma 45** (Finalized Order is Correct). *Let $G = (V, E)$ be a directed graph. Let $G' = (V, E')$ be a subgraph of $G$ where $E' \subseteq E$. If $1 \leq i < j \leq m([G'])$, then*

$$\left| R_G^-(u) \right| \leq \left| R_G^-(v) \right|, \quad \text{for all } u \in v^{(i)}([G']) \text{ and } v \in v^{(j)}([G']). \tag{15}$$

*Proof.* Since $[G']$ is a DAG, $v^{(i)}([G']) \rightsquigarrow_{[G']} v^{(j)}([G'])$ follows from Corollary 18. Then equation (15) follows immediately from Lemma 42. □

## I.3. Termination and Correctness of Algorithm 3

Now we provide a generalization of Lemma 23 to the non-transitive case with a minor adaptation to handle SCCs rather than individual vertices.

**Lemma 46** (Equal In-Reach Implies Unknown Edge). *Let $G = (V, E)$ be a directed graph. If $C, D \in [V]$ are distinct SCCs with $\left| R_{[G]}^-(C) \right| = \left| R_{[G]}^-(D) \right|$, then the edge between $C$ and $D$ is not in $[G]$. Consequently, for any $u \in C$, $v \in D$, the edge between $u$ and $v$ is not in $E$.*

*Proof.* Since $[G]$ is a DAG, Lemma 23 implies that there is no edge between $C$ and $D$ in $[G]$. This in turn implies that there cannot be any edge between any $u \in C$ and $v \in D$, because otherwise there would be an edge between $C$ and $D$ by definition. □

**Theorem 47** (Termination of Algorithm 3). *Suppose that the underlying tournament $G^* = (V, E^*)$ has at least 2 SCCs i.e., $|[V]_{G^*}| \geq 2$. Then Algorithm 3 terminates in at most $\binom{n}{2}$ rounds.*

*Proof.* We use the notation from Algorithm 3 throughout: $n_t$, $\widetilde{m}_t$, $\text{TOP}_t$, $\widetilde{\text{CAND}}_t$, and $C_t^{(i)}$ refer to the condensation quantities defined there.

If the algorithm has not terminated, $|\text{TOP}_t| < m$. Note that $n_t \geq |[V]_{G^*}| \geq 2$ for all $t$. We claim $\widetilde{m}_t < n_t$. If $\widetilde{m}_t = n_t$, then $|\text{TOP}_t| = n \geq m$, which is a contradiction. By Lemma 24 on $[G_t]$ and $\widetilde{m}_t$, we must have $\widetilde{m}_t \neq n_t - 1$. Hence, $\widetilde{m}_t \leq n_t - 2$ and thus $|\widetilde{\text{CAND}}_t| \geq 2$. We now show that there are at least 2 tied SCCs in $\widetilde{\text{CAND}}_t$, specifically $\left| R_{[G_t]}^{-} \left( C_t^{(\widetilde{m}_t+1)} \right) \right| = \left| R_{[G_t]}^{-} \left( C_t^{(\widetilde{m}_t+2)} \right) \right|$, which would allow us to make progress with Lemma 46 by querying representatives to discover at least 1 new edge. Indeed, since $[G_t]$ is a DAG with $n_t$ vertices and $\widetilde{m}_t := m([G_t])$ satisfies $\widetilde{m}_t \leq n_t - 2$ as shown, Lemma 22 implies that $r^{(\widetilde{m}_t+1)}([G_t]) = r^{(\widetilde{m}_t+2)}([G_t])$ which means $\left| R_{[G_t]}^{-} \left( C_t^{(\widetilde{m}_t+1)} \right) \right| = \left| R_{[G_t]}^{-} \left( C_t^{(\widetilde{m}_t+2)} \right) \right|$. Lemma 46 implies that there is no edge between any node in $C_t^{(\widetilde{m}_t+1)}$ and $C_t^{(\widetilde{m}_t+2)}$. Hence, since Algorithm 3 queries representatives from $C_t^{(\widetilde{m}_t+1)}$ and $C_t^{(\widetilde{m}_t+2)}$, the oracle reveals at least one new edge at round $t$ i.e., $|E_{t+1}| > |E_t|$. Since $|E_t| \leq \binom{n}{2}$, the algorithm terminates in at most $\binom{n}{2}$ rounds. $\qquad\square$

**Theorem 48** (Correctness of Algorithm 3). *If Algorithm 3 terminates, then it outputs a valid top-$m$ solution to Problem 1.*

*Remark* 49 (Why the termination theorem assumes at least two SCCs). The assumption $|[V]_{G^*}| \geq 2$ in Theorem 47 excludes the degenerate case where the underlying tournament $G^*$ consists of a single SCC. In that case every vertex reaches every other vertex, so $|R_{G^*}^{-}(v)| = n - 1$ for all $v \in V$ by Lemma 12. Hence every subset of $m$ vertices, in any order, is a valid solution to Problem 1. The interesting case is therefore when the condensation has multiple components, so that the SCC decomposition induces a nontrivial tiered ranking.

*Proof.* We want to show that once Algorithm 3 terminates at some time $T$, the sequence $(v^{(1)}, \ldots, v^{(m)})$ obtained by concatenating $C_T^{(1)}, C_T^{(2)}, \ldots, C_T^{(j^*-1)}, S_{j^*}$ where $S_{j^*}$ be any subset of $C_T^{(j^*)}$ with $|S_{j^*}| = m - \sum_{i=1}^{j^*-1} |C_T^{(i)}|$ with $j^* \leftarrow \min\{j \leq \widetilde{m}_T : \sum_{i=1}^{j} |C_T^{(i)}| \geq m\}$ satisfies a solution of Problem 1. We verify the two conditions of Problem 1 separately. Internal ordering follows from the SCC structure of finalized components. Rank dominance requires splitting into vertices inside vs. outside the boundary SCC $C_T^{(j^*)}$.

Let us first show internal ordering $\left| R_{G^*}^{-}\left(v^{(i)}\right) \right| \leq \left| R_{G^*}^{-}\left(v^{(j)}\right) \right|$ for all $1 \leq i \leq j \leq m$ as in (6). Let $v^{(i)} \in C_T^{(i')}$ and $v^{(j)} \in C_T^{(j')}$ for some $i', j' \in [j^*]$. Note that $i' \leq j' \leq j^* \leq \widetilde{m}_T$ by the concatenation construction and definition of $j^*$.

If $i' = j'$ then $v^{(i)} \sim_{G_T} v^{(j)}$ and thus $v^{(i)} \sim_{G^*} v^{(j)}$ by Lemma 39, which implies $\left| R_{G^*}^{-}\left(v^{(i)}\right) \right| = \left| R_{G^*}^{-}\left(v^{(j)}\right) \right|$ by Lemma 7.

Otherwise if $i' < j'$, then since $G_T$ is a subgraph of $G^*$ and $i' < j' \leq m([G_T])$, Lemma 45 implies that $\left| R_{G^*}^{-}(u) \right| \leq \left| R_{G^*}^{-}(v) \right|$ for all $u \in C_T^{(i')}$ and $v \in C_T^{(j')}$. This means that $\left| R_{G^*}^{-}\left(v^{(i)}\right) \right| \leq \left| R_{G^*}^{-}\left(v^{(j)}\right) \right|$. In either case, we have $\left| R_{G^*}^{-}\left(v^{(i)}\right) \right| \leq \left| R_{G^*}^{-}\left(v^{(j)}\right) \right|$.

We will now show that the solution *set* $S := \left\{ v^{(1)}, \ldots, v^{(m)} \right\}$ contains all the top ranks i.e., $\max \left\{ R_{G^*}^{-}\left(v^{(i)}\right) \mid i = 1, 2, \ldots, m \right\} \leq \min_{v \notin S} R_{G^*}^{-}(v)$ as in (7). We want to show that $R_{G^*}^{-}(u) \leq R_{G^*}^{-}(v)$ for all $u \in S$ and $v \notin S$. Note that $u$ is in one of $C_T^{(1)}, C_T^{(2)}, \ldots, C_T^{(j^*-1)}, S_{j^*}$. We will split the proof into 2 cases: (1) $v$ is in $C_T^{(j^*)} \setminus S_{j^*}$, or (2) $v$ is in $C_T^{(j)}$ for some $j > j^*$.

**Case 1**: Suppose $v \in C_T^{(j^*)} \setminus S_{j^*}$.

If $u \in S_{j^*} \subseteq C_T^{(j^*)}$, then $u$ and $v$ are in the same SCC i.e., $u \sim_{G_T} v$. Lemma 39 then implies that $u \sim_{G^*} v$ since $G_T$ is a subgraph of $G^*$. Thus, Lemma 7 implies that $\left| R_{G^*}^{-}(u) \right| = \left| R_{G^*}^{-}(v) \right|$.

Otherwise if $u \notin S_{j^*}$, then $u \in C_T^{(i)}$ for some $i < j^*$. Then Lemma 45 implies that $\left| R_{G^*}^{-}(u) \right| \leq \left| R_{G^*}^{-}(v) \right|$ since $v \in C_T^{(j^*)}$. Either cases, we have $\left| R_{G^*}^{-}(u) \right| \leq \left| R_{G^*}^{-}(v) \right|$ as desired.

**Case 2**: Suppose $v \in C_T^{(j)}$ for some $j > j^*$. Note that $u \in C_T^{(i)}$ for some $i \leq j^* \leq \widetilde{m}_T$. Since $j^* < j$, we have $i < j$.

If $j \leq \widetilde{m}_T = m([G_T])$, then since $i < j \leq m([G_T])$ and $G_T$ is a subgraph of $G^*$, Lemma 45 implies that $\left| R_{G^*}^{-}(u) \right| \leq \left| R_{G^*}^{-}(v) \right|$.

For the other subcase, if $j > m([G_T])$, we have $C_T^{(j)} \notin \mathrm{TOP}([G_T])$. Since $C_T^{(i)} \in \mathrm{TOP}([G_T])$ and $G_T$ is a subgraph of $G^*$, Lemma 44 implies that $\left|R_{G^*}^-(u)\right| \leq \left|R_{G^*}^-(v)\right|$. Either cases, we have $\left|R_{G^*}^-(u)\right| \leq \left|R_{G^*}^-(v)\right|$ as desired.

Combining case 1 and 2, we have $\left|R_{G^*}^-(u)\right| \leq \left|R_{G^*}^-(v)\right|$ for all $u \in S$ and $v \notin S$. Thus (7) is satisfied.

Thus, since the sequence $(v^{(1)}, \ldots, v^{(m)})$ output by Algorithm 3 satisfies both (6) and (7) of Problem 1, the algorithm is correct. $\qquad\square$

## J. Resolution-Based Finalization and BLITZRANK

Algorithms 2 and 3 terminate when the finalization threshold of the condensation reaches $m$, requiring the rank spectrum to crystallize into the pattern $0, 1, 2, \ldots$ up to position $m$. While this criterion is theoretically clean, it is more elaborate than what is needed for a practical implementation. In this section, we develop the vertex-level criterion used by Algorithm 1, based on the *known-relationship count*

$$\kappa_G(v) := |K_G(v)|, \qquad K_G(v) := R_G^-(v) \cup R_G^+(v).$$

A vertex is *resolved* when $\kappa_G(v) = n - 1$, meaning that its relationship to every other vertex is already determined by the revealed graph.

The key point is that resolution is sufficient for the practical stopping rule: BLITZRANK terminates once the current top-$m$ vertices by discovered in-reach are all resolved. This criterion is simpler to check than crystallization of the full rank spectrum, while still yielding a correct output. We first relate resolution to the earlier finalization framework, then prove output correctness from a basic ordering property of resolved vertices, and finally use the condensation only in the progress and termination argument.

### J.1. Properties of Known Relationship

**Proposition 50** (Known Relationships As a Necessary Condition for Finalization). *Let $G = (V, E)$ be a DAG with $m(G) \geq 1$. Then we have*

$$v \in \mathrm{TOP}(G) \implies \left|R_G^-(v) \cup R_G^+(v)\right| = n - 1.$$

*Proof.* We write $v^{(j)} := v^{(j)}(G)$ for brevity. Let $v = v^{(j)}$ for some $j \leq m(G)$ i.e., $v \in \mathrm{TOP}(G)$. We show that every vertex in $V \setminus \{v\}$ belongs to either $R_G^-(v)$ or $R_G^+(v)$. First, by Lemma 17, we know exactly the in-reaches of $v$:

$$R_G^-(v) = \begin{cases} \emptyset, & \text{if } j = 1, \\ \{v^{(1)}, \ldots, v^{(j-1)}\}, & \text{if } j > 1. \end{cases}$$

Now, for any $v^{(i)} \in \mathrm{TOP}(G)$ with $i > j$, Corollary 18 gives $v^{(j)} \rightsquigarrow_G v^{(i)}$, so:

$$\{v^{(j+1)}, \ldots, v^{(m(G))}\} \subseteq R_G^+(v).$$

Finally, by Lemma 20, for any $u \in \mathrm{CAND}(G) = V \setminus \mathrm{TOP}(G)$, we have $v \rightsquigarrow_G u$, so:

$$\mathrm{CAND}(G) \subseteq R_G^+(v).$$

The sets $R_G^-(v)$ and $R_G^+(v)$ are disjoint (since $G$ is a DAG and any vertex in their intersection would create a cycle through $v$). Combining the results above:

$$R_G^-(v) = \{v^{(1)}, \ldots, v^{(j-1)}\},$$
$$R_G^+(v) \supseteq \{v^{(j+1)}, \ldots, v^{(m(G))}\} \cup \mathrm{CAND}(G).$$

These sets partition $V \setminus \{v\}$:

$$V \setminus \{v\} = \{v^{(1)}, \ldots, v^{(j-1)}\} \cup \{v^{(j+1)}, \ldots, v^{(m(G))}\} \cup \mathrm{CAND}(G).$$

Therefore:

$$R_G^-(v) \cup R_G^+(v) \supseteq V \setminus \{v\}.$$

Trivially, we have $R_G^-(v) \cup R_G^+(v) \subseteq V \setminus \{v\}$. We conclude:

$$\left|R_G^-(v) \cup R_G^+(v)\right| = n - 1. \qquad\square$$

*Remark* 51. The converse of Proposition 50 does not hold; see the remark following Proposition 52 for a counterexample and the additional condition that yields a full characterization.

We can obtain a characterization of the known relationships by strengthening the assumption a bit. This helps with understanding the finalization criterion and will be useful for implementing a practical version of the algorithm.

**Proposition 52** (Characterization of Finalized Vertices). *Let $G = (V, E)$ be a DAG with $m(G) \geq 1$. Then*

$$v \in \text{TOP}(G) \iff \kappa_G(v) = n - 1 \text{ and } |R_G^-(v)| < m(G).$$

*Proof.* ($\Rightarrow$) Let $v = v^{(j)}(G)$ for some $j \leq m(G)$. By Proposition 50, $\kappa_G(v) = n - 1$. By the finalization threshold (Definition 16), $|R_G^-(v)| = r^{(j)}(G) = j - 1 < m(G)$.

($\Leftarrow$) Suppose $|R_G^-(v)| < m(G)$. By the contrapositive of Lemma 19, $v \notin \text{CAND}(G)$, so $v \in \text{TOP}(G)$. $\square$

*Remark* 53. The in-reach condition $|R_G^-(v)| < m(G)$ alone suffices for the backward direction; the known-relationship condition $\kappa_G(v) = n - 1$ is redundant for that implication. We retain it in the biconditional because it is the computationally natural quantity to track: the algorithm accumulates known relationships incrementally, and the in-reach condition provides a cheap secondary check once resolution is established.

Note also that the converse of Proposition 50 fails in general: $\kappa_G(v) = n - 1$ does not imply $v \in \text{TOP}(G)$. Consider $V = \{a, b, c\}$ with $E = \{(a, c), (b, c)\}$. Here $m(G) = 1$ with $\text{TOP}(G) = \{a\}$, yet $\kappa_G(c) = 2 = n - 1$. Proposition 52 correctly excludes $c$ because $|R_G^-(c)| = 2 \geq m(G) = 1$, violating the in-reach condition.

## J.2. Correctness of Algorithm 1

We now establish correctness and termination of Algorithm 1. The key observation is that the output validity depends on a single elementary property of *resolved* vertices, while the condensation machinery is needed only for proving termination.

### J.2.1. OUTPUT CORRECTNESS

Recall that the algorithm computes, for each vertex $u \in V$ in the revealed graph $G = (V, E)$ with $E \subseteq E^*$, the loss count $|R_G^-(u)|$ and the known relationship $\kappa_G(u)$. The algorithm terminates when the set $T$ of $m$ vertices with smallest discovered in-reach is contained in $F := \{u \in V : \kappa_G(u) = n - 1\}$.

**Lemma 54** (Resolved Vertex Ordering). *Let $G^* = (V, E^*)$ be a tournament and $G = (V, E)$ a subgraph with $E \subseteq E^*$. Let $u, v \in V$ be distinct. If $u$ is resolved (i.e., $\kappa_G(u) = n - 1$) and $v \notin R_G^-(u)$, then*

$$|R_{G^*}^-(u)| \leq |R_{G^*}^-(v)|.$$

*Proof.* Since $R_G^-(u) \cup R_G^+(u) = V \setminus \{u\}$ and $v \notin R_G^-(u)$, we have $v \in R_G^+(u)$, so $u \rightsquigarrow_G v$ and hence $u \rightsquigarrow_{G^*} v$ (as $E \subseteq E^*$). In $G^*$, exactly one of the following holds:

1. $v \rightsquigarrow_{G^*} u$ as well. Then $u \sim_{G^*} v$, so $u$ and $v$ belong to the same SCC of $G^*$, giving $|R_{G^*}^-(u)| = |R_{G^*}^-(v)|$ by Lemma 7.
2. $v \not\rightsquigarrow_{G^*} u$. Then $[u]_{G^*}$ and $[v]_{G^*}$ are distinct SCCs with $[u]_{G^*} \rightsquigarrow_{[G^*]} [v]_{G^*}$. Since the condensation of a tournament is a transitive tournament (Proposition 11), $[u]_{G^*}$ strictly precedes $[v]_{G^*}$ in condensation order. Lemma 12 then gives $|R_{G^*}^-(u)| < |R_{G^*}^-(v)|$.

In both cases, $|R_{G^*}^-(u)| \leq |R_{G^*}^-(v)|$. $\square$

The following observation connects the algorithm's vertex-level quantities to the hypothesis of Lemma 54.

**Lemma 55** (Smaller Discovered Loss Excludes In-Reach Membership). *Let $G = (V, E)$ be a directed graph. Let $u, v \in V$ be distinct. If $u$ is resolved (i.e., $\kappa_G(u) = n - 1$) and $|R_G^-(u)| \leq |R_G^-(v)|$, then $v \notin R_G^-(u)$.*

*Proof.* Suppose for contradiction that $v \in R_G^-(u)$. Lemma 3 gives $R_G^-(v) \cup \{v\} \subseteq R_G^-(u)$, so $|R_G^-(v)| + 1 \leq |R_G^-(u)|$, i.e., $|R_G^-(v)| < |R_G^-(u)|$, contradicting $|R_G^-(u)| \leq |R_G^-(v)|$. $\square$

Combining these 2 lemmas provide an important bridge from the resolved vertices' in-reaches of the discovered graph to the in-reaches of the underlying tournament.

**Corollary 56** (Resolved In-Reaches are True In-Reaches). *Let $G^* = (V, E^*)$ be a tournament and $G = (V, E)$ a subgraph with $E \subseteq E^*$. Let $u, v \in V$ be distinct. We have,*

$$\kappa_G(u) = n - 1 \text{ and } \left|R_G^-(u)\right| \leq \left|R_G^-(v)\right| \implies \left|R_{G^*}^-(u)\right| \leq \left|R_{G^*}^-(v)\right|.$$

*Proof.* The proof immediately follows from combining Lemma 54 and Lemma 55. □

**Theorem 57** (Output Correctness of Algorithm 1). *If Algorithm 1 terminates, its output is a valid solution to Problem 1.*

*Proof.* At termination with discovered graph $G \subseteq G^*$, $T \subseteq F$, meaning every $u \in T$ satisfies $\kappa_G(u) = n - 1$, and $T$ consists of the $m$ vertices with smallest in-reach.

**Internal ordering (6).** Let $v^{(1)}, \ldots, v^{(m)}$ be the vertices of $T$ sorted by ascending in-reach. For any $1 \leq i < j \leq m$, we have $\left|R_G^-(v^{(i)})\right| \leq \left|R_G^-(v^{(j)})\right|$ with $\kappa_G(v^{(i)}) = n - 1$. Then Corollary 56 implies that $\left|R_{G^*}^-(v^{(i)})\right| \leq \left|R_{G^*}^-(v^{(j)})\right|$.

**Rank dominance (7).** Let $u \in T$ and $v \notin T$. Since $T$ contains the $m$ vertices with smallest in-reaches, we have $\left|R_G^-(u)\right| \leq \left|R_G^-(v)\right|$. Since $\kappa_G(u) = n - 1$, Corollary 56 implies that $|R_{G^*}^-(u)| \leq |R_{G^*}^-(v)|$. Taking the maximum over $u \in T$ and minimum over $v \notin T$:

$$\max_{u \in T} |R_{G^*}^-(u)| \leq \min_{v \notin T} |R_{G^*}^-(v)|.$$

□

*Remark* 58. Lemma 54 is one-sided: only $u$ needs to be fully resolved ($\kappa_G(u) = n - 1$); no assumption is made on $v$. This is why the proof requires only $T \subseteq F$ and places no constraint on vertices outside $T$.

### J.2.2. TERMINATION

Output correctness requires no condensation machinery. Termination, however, uses the condensation to guarantee that every non-terminal round discovers at least one new edge.

**Lemma 59** (Tied SCCs Contain Unresolved Vertices). *Let $G = (V, E)$ be a directed graph. If $C, D \in [V]$ are distinct SCCs with $|R_{[G]}^-(C)| = |R_{[G]}^-(D)|$, then for any $u \in C$ and $v \in D$, we have $\kappa_G(u) < n - 1$ and $\kappa_G(v) < n - 1$.*

*Proof.* By Lemma 46, there is no edge between any vertex in $C$ and any vertex in $D$ in $E$. In particular, for any $u \in C$ and $v \in D$: $v \notin R_G^-(u) \cup R_G^+(u)$, so $\kappa_G(u) \leq n - 2 < n - 1$. The same argument gives $\kappa_G(v) < n - 1$. □

**Theorem 60** (Termination of Algorithm 1). *Algorithm 1 terminates in at most $\binom{n}{2}$ rounds.*

*Proof.* At any non-terminal round, $T \not\subseteq F$: some vertex $u \in T$ has $\kappa_G(u) < n - 1$. We show the scheduled query $Q$ discovers at least one new edge.

Let $\widetilde{m} := m([G])$ denote the finalization threshold of $[G]$, and let $n' := |[V]|$. We first establish $\widetilde{m} \leq n' - 2$. If $\widetilde{m} = n'$, then every SCC is finalized, and Proposition 50 applied to the DAG $[G]$ gives $\kappa_{[G]}(C) = n' - 1$ for every $C \in \text{TOP}([G]) = [V]$. This implies $\kappa_G(u) = n - 1$ for all $u \in V$ by Lemma 14, contradicting the existence of an unresolved vertex. Hence $\widetilde{m} < n'$. Lemma 24 then gives $\widetilde{m} \neq n' - 1$, so $\widetilde{m} \leq n' - 2$.

By Lemma 22 applied to the DAG $[G]$, the SCCs at positions $\widetilde{m} + 1$ and $\widetilde{m} + 2$ in the condensation rank spectrum are tied:

$$|R_{[G]}^-(C^{(\widetilde{m}+1)})| = |R_{[G]}^-(C^{(\widetilde{m}+2)})|.$$

By Lemma 59, both SCCs contain vertices with $\kappa_G(\cdot) < n - 1$, so both appear in the candidate set $\mathcal{C}$ of the algorithm. Since they share the same in-reach value in $[G]$, they are among the earliest elements of $\mathcal{C}$ (sorted by ascending in-reach). The algorithm selects representatives from each, and by Lemma 46, no edge exists between these representatives in $E$. The oracle query therefore reveals at least one new edge: $|E_{t+1}| > |E_t|$.

Since $|E_t| \leq \binom{n}{2}$ for all $t$, the algorithm terminates in at most $\binom{n}{2}$ rounds. □

*Remark* 61 (Comparison with Algorithms 2 and 3). The theoretical algorithms terminate when the finalization threshold of the condensation reaches $m$, requiring the rank spectrum to crystallize into the pattern $0, 1, 2, \ldots$ up to position $m$. Algorithm 1 terminates when the $m$ lowest-loss vertices are individually resolved ($\kappa_G(\cdot) = n - 1$), a vertex-level condition that can be satisfied strictly earlier than full crystallization. The condensation enters Algorithm 1 only in the scheduling step (for ordering candidate SCCs) and in the termination proof (for guaranteeing progress). The output correctness proof is independent of the condensation and relies solely on Lemma 54.

## K. Query Complexity Discussion

### K.1. Setup and Notation

We measure *query complexity* as the number of calls made to the $k$-wise oracle. Formally, an algorithm $\mathcal{A}$ adaptively selects query sets $Q_1, Q_2, \ldots$ with $Q_t \subseteq V$ and $|Q_t| \leq k$, observes $\mathcal{O}(Q_t)$, and terminates once it can certify the desired output (top-$m$ selection).

**Instance-wise and worst-case complexity.** For a fixed underlying tournament $G^*$, let $T_\mathcal{A}(G^*)$ denote the (random) number of oracle calls made by $\mathcal{A}$ until termination.[4] We define the worst-case query complexity over a class $\mathcal{G}$ as

$$Q_\mathcal{A}(\mathcal{G}; n, k, m) := \sup_{G^* \in \mathcal{G}} T_\mathcal{A}(G^*).$$

In particular, we will distinguish the transitive class $\mathcal{T}_n^{\mathrm{tr}}$ (total orders) from the general class $\mathcal{T}_n$ (all tournaments).

**Sequential vs. parallel depth.** The complexity above counts total oracle calls. If multiple disjoint query sets are evaluated in parallel, a second measure is the number of *rounds* (parallel depth). We defer a formal treatment of parallel depth to future work.

### K.2. Transitive Case

The $\binom{n}{2}$ bound from Theorem 36 is a worst-case upper bound that does not exploit the structure of greedy selection. We now derive tighter bounds.

**Proposition 62** (Top-1 Query Complexity). *When $m = 1$, Algorithm 2 identifies the top vertex in at most $\left\lceil \frac{n-1}{k-1} \right\rceil$ queries.*

*Proof.* In the transitive setting, the restriction of $G^*$ to any queried set $Q$ is a total order, hence it has a unique best element (the one with zero losses within $Q$). By transitivity, any other element of $Q$ loses to this best element and therefore cannot be the global best. Thus each query can eliminate at least $|Q| - 1 \geq k - 1$ candidates from being the top vertex. Starting from $n$ candidates, after $q$ queries at most $n - q(k - 1)$ candidates remain; requiring $n - q(k - 1) \leq 1$ yields the bound. □

### K.3. A Conjecture for Top-$m$ Query Complexity

Beyond $m = 1$, a tight query complexity analysis remains open. The proved termination bound is $O(n^2)$ (Theorem 60), obtained by showing that each non-terminal round reveals at least one new edge. This existential progress argument does not exploit the greedy structure of the scheduler, which empirically produces query counts far below $O(n^2)$. For $m = 1$, Proposition 62 gives a tight $O(n/k)$ bound via a clean elimination argument: each query eliminates $k - 1$ candidates. For $m > 1$, the algorithm must simultaneously refine multiple rank positions, and the interaction between candidate elimination and frontier advancement resists a simple recurrence. We state a conjectural bound motivated by the structure of the algorithm and validated empirically in Section K.6.

*Conjecture* 63 (Top-$m$ Query Complexity). Let BLITZRANK$^\dagger$ denote Algorithm 1 with candidate SCCs ordered lexicographically by ascending condensation in-reach, then ascending condensation out-reach, then fixed ID; and with each $\mathrm{rep}(C)$ chosen by minimum $\kappa_G$, then fixed ID. Then

$$Q_{\text{BLITZRANK}^\dagger}(\mathcal{T}_n; n, k, m) = O\left( \frac{n-1}{k-1} + \frac{m-1}{k-1} \log_k m \right).$$

---

[4]In this paper the oracle is deterministic; randomness, if any, comes from tie-breaking or implementation choices.

*Remark* 64 (Decomposition). The conjectured form decomposes into two terms:

- A *candidate reduction* term $(n-1)/(k-1)$: the cost to winnow $n$ candidates down to a single maximum, matching Proposition 62.
- A *frontier refinement* term $(m-1)/(k-1) \cdot \log_k m$: the additional cost to extract elements $2, \ldots, m$ from the remaining candidates, reflecting the logarithmic depth of repeated selection.

The $O(\cdot)$ notation hides a constant factor. Empirically (Section K.6), we observe that this constant is at most 1.25 across all tested configurations, suggesting the functional form is tight up to lower-order terms. The specific tie-breaking rules in BLITZRANK$^\dagger$ pin down a single deterministic algorithm, avoiding ambiguity from arbitrary implementation choices.

*Remark* 65 (Lower bounds). Standard arguments show the conjectured form is tight up to constants. Elimination gives $\Omega((n-1)/(k-1))$ for finding the maximum (Proposition 62); a decision-tree bound gives $\Omega(\log(m!)/\log(k!)) = \Omega(m \log m/(k \log k))$ for ordering $m$ items with $k!$-outcome queries. When $m = n$ the latter recovers the classical $\Omega(n \log n/(k \log k))$ sorting lower bound. Since $\max(a, b) = \Theta(a + b)$ for non-negative $a, b$, the maximum of the two bounds already matches the conjectured $(n-1)/(k-1) + (m-1)/(k-1) \cdot \log_k m$ up to constant factors; the open challenge is the upper bound.

*Remark* 66 (Difficulty of a formal proof). The gap between the proved $O(n^2)$ termination bound and Conjecture 63 reflects a fundamental limitation of the existing analysis, which is built around *correctness and eventual progress* rather than a sharp accounting of how quickly the greedy scheduler resolves the top-$m$ frontier. In Algorithm 1, the condensation is used only to order candidate SCCs and to guarantee that every non-terminal iteration reveals at least one new edge (Theorem 60) – far weaker than the conjectured scaling. Proving the stronger bound would require a new structural invariant showing that repeated $k$-wise queries not only make progress, but concentrate information efficiently on the unresolved top-$m$ boundary.

Two structural mismatches compound the difficulty. First, the algorithm schedules queries at the SCC level of the condensation, while termination is a vertex-level condition ($\kappa_G(v) = n - 1$); bridging these levels requires tracking how SCC-level queries translate into vertex-level resolution. Second, transitive-closure effects make the dynamics nonlocal: a single new edge can trigger cascading updates to in-reach sets and SCC structure, resolving vertices far from the query set. The 25-horses example (Figure 2) illustrates a third difficulty: the per-query information gain is inherently non-uniform. The execution exhibits three distinct phases: (i) a *skeleton-building* phase (Rounds 1–5) where disjoint groups are sorted independently and each query reveals exactly $k - 1$ new edges; (ii) a *connection* phase (Round 6) where group winners are compared, transitive closure propagates across previously isolated components, and vertices resolve en masse; (iii) a *saturation* phase (Round 7) where the remaining top candidates require targeted comparisons with diminishing transitive-closure leverage. Any tight analysis must likely account for these phase transitions, as no single amortization rate can capture the varying progress. Establishing the conjecture would likely require a phase-aware potential function that tracks the rate at which the top-$m$ frontier crystallizes – a substantial extension of the present framework.

### K.4. Why the Bound Should Extend to All Tournaments

Conjecture 63 states the bound over the full class $\mathcal{T}_n$. The following remark provides structural intuition for why non-transitive tournaments should not be harder than transitive ones.

*Remark* 67 (Why Cycles Help). The intuition is that cycles can only make top-$m$ selection *easier*. When vertices form a cycle in $G^*$, they belong to the same SCC and thus share a common rank in the condensation DAG. Once the algorithm discovers such a cycle, all vertices in the SCC finalize simultaneously – they need not be distinguished from one another. In contrast, the transitive case requires establishing pairwise orderings among all vertices: no shortcuts exist.

More precisely, for any tournament $G^*$, one can construct a "shadow" transitive tournament $\tilde{G}$ that respects the SCC tier structure of $G^*$: vertices in better-ranked tiers beat vertices in worse-ranked tiers, with intra-tier orderings assigned arbitrarily. Inter-tier edges in $\tilde{G}$ are determined by the same tier structure as in $G^*$, so any query sequence reveals the same inter-tier relationships in both. The difference lies within tiers: in $\tilde{G}$, all vertices are singletons requiring full resolution; in $G^*$, cycles merge vertices into SCCs that finalize together. Thus the query count on $G^*$ should never exceed that on $\tilde{G}$. A formal proof of this reduction remains open.

### K.5. Empirical Methodology

We empirically validate Conjecture 63 by measuring the query complexity of BLITZRANK$^\dagger$ on synthetic transitive instances – the conjectured worst case (Remark 67).

**Instance generation.** A transitive tournament on $n$ vertices corresponds to a total order, uniquely determined by a permutation $\pi : [n] \to [n]$. The underlying tournament $G_\pi^*$ has edge $u \to v$ iff $\pi^{-1}(u) < \pi^{-1}(v)$. The oracle $\mathcal{O}$ applied to any query set $Q \subseteq V$ returns the restriction of $G_\pi^*$ to $Q$ – equivalently, the total order on $Q$ induced by $\pi$.

We test three classes of initial orderings:

1. *Random*: $\pi$ sampled uniformly at random (20 independent seeds per $(n, k)$ pair).
2. *Identity*: $\pi = \mathrm{id}$, i.e. items presented in sorted order. The best items appear first in the input, giving the algorithm early access to them.
3. *Reverse*: $\pi(i) = n - i$, i.e. items presented in reverse sorted order. The best items appear last, forcing the algorithm to scan the entire input before encountering them.

The identity and reverse permutations are deterministic (one run per $(n, k)$ pair).

**Measurement protocol.** For each parameter triple $(n, k, m)$, we execute BLITZRANK$^\dagger$ with target count $m$ and record $T_{\mathcal{A}}(G_\pi^*)$, the number of oracle calls until termination. Rather than running separate experiments for each $m \in \{1, \ldots, n\}$, we exploit a key observation: a single execution with $m = n$ (full sort) reveals $Q(n, k, m')$ for all $m' \le n$ as a byproduct. Specifically, when the $m'$-th vertex is finalized at round $t$, we record $T_{m'} := t$ as the query count for top-$m'$ selection. This yields the full curve $m \mapsto Q(n, k, m)$ from one run.

**Experimental grid.** We test $n \in \{100, 200, 400, 800\}$ and $k \in \{2, 5, 10, 20, 50\}$, extracting $m = 1, \ldots, n$ from each full-sort run. For each $(n, k, m)$, we report the median, 10th–90th percentile range, and maximum of $T_m$ across seeds (for random permutations) or the single observed value (for identity and reverse).

### K.6. Empirical Results

Figure 10 presents the empirical query counts alongside the concrete reference curve

$$B(n, k, m) = \left\lceil \frac{n-1}{k-1} \right\rceil + \frac{m-1}{k-1} \cdot (1 + \log_k(m)), \tag{16}$$

a non-asymptotic instantiation of the $O(\cdot)$ form in Conjecture 63 (the additive 1 inside the parenthesis is absorbed by the big-$O$).

**Main observations.**

1. *The conjectured form captures the scaling.* Across all tested $(n, k)$ pairs, the empirical curves follow the shape predicted by $B(n, k, m)$: sublinear in $k$, with a candidate-reduction plateau for small $m$ transitioning to logarithmic growth.
2. *The constant is small.* The maximum observed query count satisfies $\max_\pi T_{\mathcal{A}}(G_\pi^*) \le 1.25 \cdot B(n, k, m)$ for all tested configurations. The median typically lies below $B(n, k, m)$ itself.
3. *Variance is low.* The 10th–90th percentile bands are narrow, indicating that query complexity is stable across random permutations and not dominated by rare worst-case instances within our sample.
4. *Structured orderings are easier than random.* The identity and reverse permutations – representing the two extremes of initial ordering – produce query counts that lie uniformly below the random baseline across all tested $(n, k, m)$ configurations (Figure 10). Both stay well within $B(n, k, m)$ itself, far from the 1.25 envelope. This suggests that random permutations, which maximally mix rank positions, are harder for the algorithm than structured orderings.

*Table 11.* Full-sort query counts ($m$=$n$, $k$=10) for three initial orderings. Random reports [min, max] over 20 seeds.

| $n$ | Identity | Reverse | Random [min, max] | $1.25 \cdot B(n, k, n)$ |
|---|---|---|---|---|
| 100 | 25 | 25 | [35, 39] | 55 |
| 200 | 53 | 53 | [86, 89] | 120 |
| 400 | 109 | 109 | [204, 212] | 255.9 |
| 800 | 221 | 222 | [470, 480] | 544.4 |

Table 11 summarizes full-sort query counts ($m$=$n$) at $k$=10. The gap between structured and random permutations widens as $n$ increases: at $n$=800, identity and reverse require 221–222 queries while random requires 470–480 – all well within

the $1.25 \cdot B(n, k, n)$ envelope. Notably, identity and reverse permutations yield nearly identical query counts despite representing opposite extremes of the initial ordering, suggesting the algorithm's efficiency depends on the inherent structure of the permutation rather than its alignment with the target ranking.

**Limitations.** These experiments provide evidence for, but do not prove, Conjecture 63. Key caveats:

- We test identity, reverse, and random permutations; a worst-case permutation engineered with knowledge of the algorithm's internal scheduling could conceivably yield higher query counts. Transitive tournaments are the conjectured worst case (Remark 67), but this reduction itself remains unproved.
- The parameter range is bounded ($n \leq 800$, $k \leq 50$); extrapolation to larger scales is uncertain.
- The 1.25 multiplicative gap may reflect lower-order terms not captured by the asymptotic form, or slack in our analysis.

See Remark 66 for a discussion of the obstacles to a formal proof.

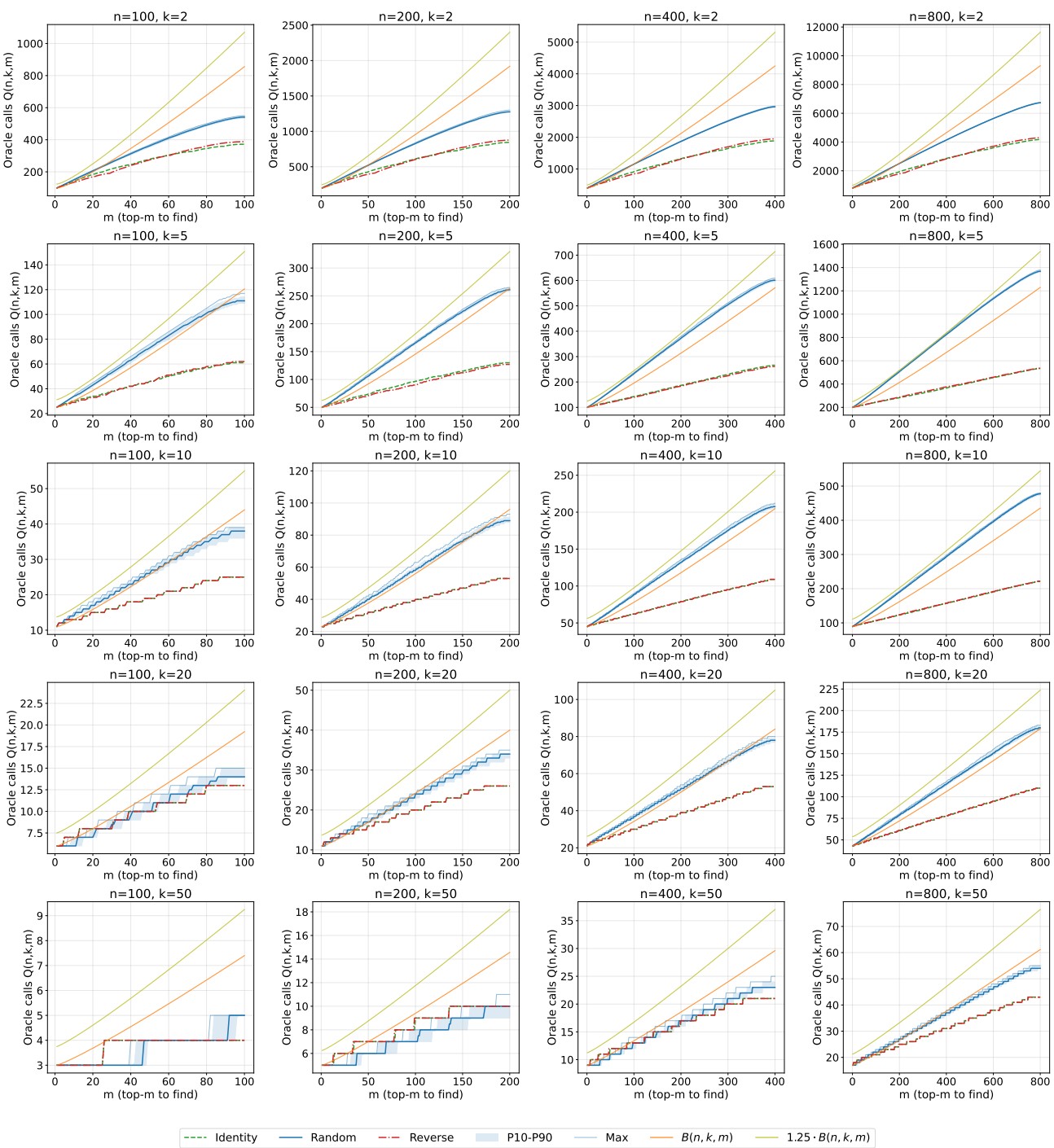

*Figure 10.* Empirical query counts $Q(n, k, m)$ versus target count $m$ for three initial orderings: random (blue, 20 seeds with P10–P90 band and max), identity (green, sorted input), and reverse (red, reverse-sorted input). The conjectured bound $B(n, k, m)$ (orange) closely tracks the empirical complexity; $1.25 \cdot B(n, k, m)$ (yellow-green) upper-bounds all observed counts. Identity and reverse permutations lie uniformly below the random baseline.

