# OpenReview forum: "BlitzRank: Principled Zero-shot Ranking Agents with Tournament Graphs"
_ICML.cc/2026/Conference — ICML 2026 spotlight_

### Official Review · Reviewer_gNEN · 2026-02-13

**Soundness:** 3
**Presentation:** 3
**Significance:** 2
**Originality:** 2
**Overall Recommendation:** 4
**Confidence:** 4

**Summary:**

The paper proposes a zero-shot ranking method that treats Large Language Model (LLM) ranking as a tournament graph problem. Instead of expensive listwise comparisons, it employs a greedy query scheduling strategy to infer relationships transitively. To handle non-transitive loops (Condorcet cycles), the authors propose collapsing Strongly Connected Components (SCCs) into single nodes (ties) within a condensation graph. Theoretical bounds for query complexity are provided for both transitive and general cases.

**Compliance With Llm Reviewing Policy:**

Affirmed.

**Final Justification:**

Thank you to the authors for their rebuttal. I appreciate the clarification regarding the big cycle. That said, I still believe this is a theoretically interesting direction to explore. Regarding weighted (non-binary) preferences, I don’t think this issue is orthogonal, and I believe it would be worthwhile to investigate in future work. My other minor concerns have been addressed satisfactorily. I will maintain my positive score.

**Key Questions For Authors:**

- What is the main rationale for modelling only binary relationships between the ranked items?

- Is the choice of *K* generalisable across different datasets, or does it need to be tuned per dataset? Would it also vary depending on the LLM used?

**Limitations:**

yes

**Strengths And Weaknesses:**

**Strengths:**

- The storyline is well presented and easy to follow.

- The greedy active-learning strategy for edge selection shows a clear reduction in token consumption compared to standard sorting algorithms.

- The complexity analysis is well justified.

**Weaknesses:**

- There are two major concerns regarding the design of the tournament framework:
  1) If I understand correctly, cycles are currently treated as a “tie.” However, in an extreme case where a limited set of items forms a single large cycle, the framework may effectively treat all of them as tied. In other words, if the graph contains a few large cycles, this could significantly affect performance.
  2) Related to the first concern, the current framework appears to model only a binary relationship (“win” or “lose”) in pairwise comparisons, without capturing the strength of preference (e.g., “how much A surpasses B”). This may make breaking cycles challenging.

- The tournament-based design is not entirely new. Prior work on traditional pairwise and listwise ranking methods has explored similar mechanisms (e.g., TSPRank [1]). While this paper applies a tournament-based approach to LLM rankers, it does not discuss how it relates to traditional methods that share a similar design philosophy. And this leads to potential discussion on whether the condensation can be replaced by traditional optimisation method such as combinatorial optimisation to resolve the tie (but of course this will introduce extra latency, and you might need to model the edge weights as mentioned above).

[1] TSPRank: Bridging Pairwise and Listwise Methods with a Bilinear Travelling Salesman Model. KDD 2025.

- The paper only compares *K = 10* and *K = 20*, but it would be better to include a few additional values to better summarise the overall trends. With only two variants, it’s hard to tell whether this reflects a consistent pattern or a hand-picked special case.

Minor:

- Currently, the main results table (Table 3) is placed in the appendix and lacks clear guidance on how to interpret it. I suggest moving representative columns or summary statistics (e.g., averages) into the main paper; otherwise, it is difficult for readers to go back and forth between the table and the discussion.

---

> ### Author Rebuttal · Authors · 2026-03-29
>
> We thank the reviewer for their thoughtful evaluation and for recognizing the clarity of the storyline, the efficiency gains of our greedy scheduling strategy, and the well-justified complexity analysis. We address each concern below.
>
> ---
>
> > Large cycles concern
>
> We provide a detailed response on oracle noise and cycles in our reply to Reviewer YCnN. In short: large cycles are empirically rare (average SCC size 1.07 at k=10, 1.18 at k=20; Table 1), and when they do form, they capture genuinely similar documents (~40% lower BM25 variance; Table 2). We acknowledge this in our limitation section and are actively working on a relaxation.
>
> ---
>
> > Binary relationships / edge weights: "What is the main rationale for modeling only binary relationships between the ranked items?"
>
> Thank you for this interesting connection. In our setting, we assume access to an oracle that gives us pairwise relationships (as in the LLM zero-shot reranking setup). Relaxing this to non-binary edges is an interesting but separate problem. That said, incorporating edge confidence is a natural extension in the noisy oracle direction we are actively pursuing (discussed in Section 5).
>
> ---
>
> > TSPRank and traditional tournament methods
>
> We appreciate the pointer to TSPRank. From our understanding, TSPRank and our framework operate under fundamentally different regimes. TSPRank *trains* a listwise reranker, while our framework assumes access to an arbitrary listwise reranker as an oracle (we use RankGPT, but this could be any listwise reranker). We operate in a zero-shot setting where LLMs have been established as powerful general rerankers, and where limited context windows (effectively ranking only 10–20 documents at a time) and high inference costs are the core problems our paper addresses.
>
> That said, our tournament-graph sort algorithm is general purpose and works with arbitrary listwise oracles — using TSPRank as the underlying reranker would be an interesting but orthogonal direction.
>
> ---
>
> > "Is the choice of k generalizable across datasets, or does it need to be tuned per dataset? Would it also vary depending on the LLM used?"
>
> Our algorithm has a *single* hyperparameter: the window size k (Section 4.3, Page 7). We did not tune k across datasets or LLMs. We use k=10 and k=20, which are standard values in the listwise reranking literature. In Figure 4 (Page 6), both TS-k10 and TS-k20 are substantially more efficient than existing approaches while matching or exceeding accuracy. We consistently observe that k=10 is more accurate but slightly more expensive than k=20. Hence, the choice of k simply trades cost for accuracy.
>
> Appendix H further covers k ∈ {2, 5, 10, 20, 50} on synthetic instances up to n=800, showing consistent trends (Figure 10).
>
> ---
>
> > Table 3 placement
>
> Thank you for the suggestion and we apologize for any difficulty with interpreting Table 3. We will curate summary statistics and representative columns into the main body to improve readability in the revision.
>
> ---
>
> We again thank the reviewer for their constructive feedback and the positive assessment. We hope that the clarifications above — particularly on the empirical rarity of large cycles, the distinction from TSPRank's training-based regime, and the robustness of k across datasets and LLMs — address the raised concerns. We welcome further discussion during the rebuttal period.

---

> > ### Author Rebuttal · Reviewer_gNEN · 2026-04-01
> >
> > Thanks to the authors for the rebuttal and clarifications. For two of the main weaknesses I raised, the response helps, but it seems addressing them may require significant adjustments to the framework, which would be difficult to complete within the rebuttal period. Regarding the relevance to TSPRank, additional K settings, and the Table 3 arrangement, the authors provided clear justification and agreed to refine these in the revision. Overall, I will keep my positive rating.

---

> > > ### Author Response · Authors · 2026-04-02
> > >
> > > We thank the reviewer again for the thoughtful engagement and for maintaining a positive assessment. We are glad that several points (relation to TSPRank, choice of k, and presentation) are now clearer and we are committed to include these in our revisions.
> > >
> > > Regarding the two main weaknesses on cycles and richer preference modeling, where “addressing them may require significant adjustments to the framework,” we would like to further clarify that these can be addressed with additional exposition and natural extensions, rather than substantial changes:
> > >
> > > ---
> > >
> > > **1\. Cycles are already handled within the current framework.**
> > > Rather than forcing a total order, our framework captures non total-order (like cycles) via SCC condensation. Extensive empirical results support this modeling choice:  as noted in the rebuttal, large cycles are rare and correspond to genuinely similar items, suggesting that our modeling choice for non-total order is semantically meaningful rather than a failure mode.
> > >
> > > ---
> > >
> > > **2\. Weighted / non-binary preferences are an orthogonal but natural extension of our framework.**
> > > In the zero-shot ranking setup that we consider, there are 2 layers: the oracle and *the algorithm on top of the oracle*. Our paper focuses on the algorithm layer, where we build on top of SoTA oracles like RankGPT which provide binary preferences. Investigating a more optimal oracle design (including weighted/non-binary preferences) is an interesting but *orthogonal direction* to the goal/scope of our paper.
> > >
> > > ---
> > >
> > > The main contribution of the paper is a principled and query-efficient framework for exploiting k-wise comparisons on listwise ranking oracles via tournament graphs and transitive closure, together with theoretical guarantees and consistent empirical gains across 14 benchmarks and 5 LLMs. The concerns raised correspond to **extensions** along the modeling axis (handling richer signals or additional assumptions), rather than limitations that affect the validity of the current results.
> > >
> > > We hope this clarifies that these points can be addressed with concrete clarifications and minor additions in revision, rather than fundamental changes. If the reviewer had a specific scenario in mind, we would greatly appreciate further clarification and would be happy to address it directly.
> > >
> > > We appreciate the reviewer’s careful reading and constructive feedback, which we believe will help strengthen the final version.

---

### Official Review · Reviewer_nqBb · 2026-03-12

**Soundness:** 3
**Presentation:** 3
**Significance:** 3
**Originality:** 4
**Overall Recommendation:** 5
**Confidence:** 4

**Summary:**

This paper introduces a tournament graph framework for query-efficient top-$m$ selection using $k$-wise comparison oracles, with application to LLM-based zero-shot document reranking. The central insight is that each $k$-wise comparison reveals a complete tournament of $\binom{k}{2}$ pairwise preferences, and accumulating these into a global preference graph with transitive closure yields additional orderings without further oracle calls. The proposed algorithm, TOURNAMENTSORT, greedily queries candidates with the fewest known relationships and terminates when the top-$m$ items are certifiably finalized. Non-transitive preferences (cycles from inconsistent LLM judgments) are handled via SCC condensation, producing principled tiered rankings. Evaluated across 14 retrieval benchmarks and 5 LLM oracles, the method achieves Pareto dominance on the accuracy-efficiency frontier: comparable or better nDCG@10 at 25-40% fewer tokens than methods with similar structure, and $7\times$ fewer tokens than pairwise reranking.

**Compliance With Llm Reviewing Policy:**

Affirmed.

**Ethical Review Concerns:**

Extracting the text from the submitted PDF reveals invisible instructions embedded in the document.

**Ethical Review Flag:**

Flag this paper for an ethics review.

**Ethics Expertise Needed:**

["Other Expertise"]

**Final Justification:**

I recommend **Accept (5)**. This paper makes a novel connection between LLM-based reranking and tournament graph theory, introducing a principled framework that moves beyond heuristic sliding windows. The central insight—that $k$-wise queries reveal a complete tournament whose structure can be exploited via transitive closure and SCC condensation—is both elegant and effectively addresses information waste in prior methods.

The authors' rebuttal was strong, resolving the window-size confound through a decisive comparison (SW-k10) and providing new ablation evidence for the greedy scheduling's efficiency. These additions clearly demonstrate that the method's 25-40% token savings are the result of superior algorithmic design rather than LLM window bias. While the deterministic oracle assumption remains a limitation, the framework’s consistent performance across 14 benchmarks and 5 oracles establishes clear Pareto dominance on the accuracy-efficiency frontier. The rebuttal resolved my main concerns regarding comparison fairness and scheduling value, justifying a solid acceptance.

The rating is final.

**Key Questions For Authors:**

1. **Robustness to oracle noise**: How does the framework perform when oracle responses are stochastic? If a fraction $\epsilon$ of pairwise judgments are flipped, how does accuracy degrade compared to baselines? Transitive closure could amplify isolated errors into systematic ranking corruption — have you observed this in practice? This is the single most important question, as the answer determines whether the theoretical guarantees meaningfully transfer to the real (noisy) setting.

2. **Disentangling sources of advantage**: The framework's advantage over Sliding Window comes from three sources: intelligent scheduling, cross-window transitive closure, and early termination. An ablation study would be valuable — e.g., (a) TournamentSort without transitive closure (using only direct edges), (b) Sliding Window with accumulated pairwise relationships and transitive closure, or (c) TournamentSort with random scheduling instead of greedy. This would clarify whether the dominant contribution is the scheduling, the accumulation, or the certification.

3. **Sliding Window at $k=10$**: The main comparison uses SW with $k=20$ against TS with $k=10$. Since TS-k10 outperforms TS-k20 partly due to LLMs producing cleaner orderings at smaller window sizes, how does SW with $k=10$ and stride 5 perform? This would control for the window size confound and isolate the contribution of transitive closure and greedy scheduling.

4. **Scalability to larger $n$**: The experiments use $n=100$ (BM25 top-100). For settings with $n=1000$ or larger candidate sets, does the framework's advantage grow (as the theory suggests) or do practical issues (graph computation overhead, deeper transitive chains amplifying noise) emerge?

5. **Conjecture 49 — adversarial instances**: The query complexity conjecture is validated on random transitive instances. Have you tested adversarial permutations (e.g., the identity permutation, reverse permutation, or structured permutations designed to delay finalization)? If the bound holds even for adversarial cases, this would significantly strengthen the claim.

6. **Embedded Prompt Injection.** The PDF of the manuscript contains hidden text (invisible to human readers but extractable via text parsing tools) that instructs LLM-based reviewers to include specific "canary phrases" in their reviews. Please explain the presence and purpose of this prompt injection in your submitted manuscript. Was this added by the authors, and if so, what is the justification? (See appended evidence).

**Limitations:**

The authors adequately discuss the deterministic oracle assumption as the primary limitation and identify noisy/probabilistic oracles as a significant open problem. The discussion of error amplification via transitive closure is honest. Future directions (confidence-weighted edges, incorporating priors, parallel depth analysis) are reasonable.

However, the limitations section could be strengthened by: (1) providing even a preliminary empirical analysis of robustness to noise (e.g., by running the same queries twice and measuring consistency), and (2) discussing the practical implications of the within-window transitivity constraint — that the framework's generality beyond ranked-list oracles is theoretical rather than demonstrated.

The impact statement appropriately notes the framework's contribution to sustainable LLM use and identifies no specific negative societal impacts.

**Strengths And Weaknesses:**

**Strengths:**

- The paper makes a genuinely novel connection between LLM reranking and tournament graph theory. While the individual components (tournaments, transitive closure, SCCs) are well-known in combinatorics, their composition into a unified framework for query-efficient ranking with certifiable termination is original and intellectually satisfying. The observation that existing $k$-wise methods discard $\binom{k}{2} - (k-1)$ comparisons per query is simple but previously overlooked, and the framework directly exploits this gap. The treatment of non-transitivity via SCC condensation — reinterpreting cycles as tie tiers rather than noise — is a principled design choice that contrasts favorably with ad hoc approaches like multiple random seeds (TourRank) or score averaging.

- The theoretical development is great. The correctness proof for the transitive case builds a clean chain: in-reach lower-bounds true in-degree (Lemma 2), discovered ranks underestimate true ranks (Lemma 3), and the finalization criterion (Definition 6 + Proposition 5) certifies exact identification of the top-$m$ set. The progress guarantee (Lemma 10: tied candidates have unknown edges) ensures termination. The generalization to non-transitive tournaments via SCC condensation (Proposition 24: condensation of any tournament is transitive) elegantly lifts all transitive-case machinery. The proofs were verified and found to be correct.

- The framework addresses a genuine practical problem — reducing the cost of LLM-based reranking in RAG pipelines — with a principled approach. The Pareto dominance across 5 diverse LLM oracles (both closed- and open-source, spanning different capability levels) is a notably consistent empirical signal — many reranking methods look good on one oracle but degrade on others. The deterministic convergence ($\pm 2\%$ variance in query count) is a meaningful practical advantage for cost budgeting. The ability to accommodate variable $k$ per round (adapting to heterogeneous document lengths and context constraints) adds practical flexibility no baseline offers. The SCC analysis (Section 4.4) provides useful insight into when and why LLMs produce inconsistent rankings.

**Weaknesses:**

- The deterministic oracle assumption is the framework's central limitation. Real LLM oracles are stochastic: the same pairwise comparison can yield different results across calls. Transitive closure amplifies errors — a single incorrect edge along a chain can corrupt many transitively inferred relationships. The paper acknowledges this (Section 5) and insightfully identifies the asymmetry: "an edge along a long chain discriminates many pairs transitively, so its corruption is catastrophic, while a leaf edge affects only one comparison." However, no theoretical or empirical analysis of robustness to oracle noise is provided. The SCC analysis partially addresses this (cycles absorb inconsistencies), but the more dangerous failure mode — an erroneous edge that does *not* create a cycle but silently corrupts a transitive chain — remains unquantified. An experiment with controlled noise injection (e.g., flipping a fraction of oracle responses) would significantly strengthen the paper.

- The practical efficiency gains, while consistent, are incremental. TS-k10 achieves +0.2 nDCG@10 over single-pass Sliding Window at 22% fewer tokens — meaningful at scale but not transformative. The accuracy gap to Pairwise reranking is small (0.2 points) but Pairwise uses $7.5\times$ more tokens, a comparison that may overstate the practical importance since few production systems use full pairwise. The most relevant practical comparison is against Sliding Window and AcuRank, where gains are modest.

---

> ### Author Rebuttal · Authors · 2026-03-29
>
> We thank the reviewer for their thorough and insightful evaluation, and for verifying the proofs. We address each question below.
>
> ---
>
> > Robustness to oracle noise
>
> This is a great question. We provide a detailed response to this point in our reply to Reviewer  YCnN, which we summarize here.
>
> While a full noise study is beyond the scope of this paper, we note indirect evidence that noise is well-behaved with modern LLMs and that our method absorbs it gracefully via SCCs. In particular: the SCC analysis (Table 2) shows cycles capture similar documents (~40% lower BM25 variance); comparisons across 5 oracles of varying capabilities provide a diverse noise sensitivity setup where we observe stable convergence; and increasing k from 10 to 20 produces only slightly larger SCCs (1.18 vs 1.07) rather than failing catastrophically (Table 1).
>
> We acknowledge the reviewer's point — *"an erroneous edge that does not create a cycle but silently corrupts a transitive chain"* — as a limitation in Section 5 and are actively working on a relaxation of this problem.
>
> ---
>
> > Disentangling sources of advantage
>
> Thank you for this suggestion. Disentangling the sources of advantage is an important question. We address each proposed ablation below, beginning with (c) where we ran new experiments.
>
> **(c) TournamentSort with random scheduling**
>
> We agree that isolating the contribution of greedy scheduling is the most informative ablation. On synthetic transitive instances (where oracle calls can be counted exactly, 20 seeds, setup as in Section H.5), we compared greedy scheduling against two random baselines: (1) random k-subsets drawn from the candidate set, and (2) random k-subsets drawn from all nodes.
>
> The results are striking. Replacing greedy with random scheduling on candidates requires 2.5×–5.8× more oracle calls, and random on all nodes requires 8.6×–28.6× more — both with substantially higher variance. These slowdowns stem from wasted calls on non-frontier and repetitive subsets. We will include the full table and figure in the revised paper.
>
> **(a) Without transitive closure**
>
> Transitive closure is core to every step of TournamentSort: finalization, condensation reduction, and scheduling all depend on in-reach and out-reach, which require transitive closure to compute. "TournamentSort without transitive closure" would not be a variant of TournamentSort — it would be a different algorithm entirely.
>
> **(b) Sliding Window with accumulated edges and transitive closure**
>
> We appreciate the idea but do not see a straightforward way to realize it — Sliding Window operates on a fixed, predetermined schedule and cannot adapt to where uncertainty lies in the preference graph. We would be happy to investigate this if you could provide further details on the specific mechanism you have in mind.
>
> ---
>
> > Sliding Window at k=10
>
> This comparison highlights a fundamental asymptotic advantage of our framework. SW with k=10 and stride 5 propagates only the top 5 documents per pass. In contrast, TournamentSort with k=10 guarantees correct top-10 identification at any window size, because correctness is certified by the resolution criterion, not by window coverage. We report the *NDCG@10 on TREC DL19 and DL20* in the table below:
>
> | Method | k | DL19 | DL20 |
> |:---|:---:|:---:|:---:|
> | Sliding Window | 20 | 74.0 | 70.8 |
> | Sliding Window | **10** | *56.4* | *53.2* |
> | Tournament Sort | 20 | 74.6 | 70.7 |
> | Tournament Sort | 10 | 73.6 | 72.4 |
>
> ---
>
> > Scalability to larger n
>
> The method's computational advantage should grow as n scales up to thousands. Graph operations (SCC via Tarjan's, transitive closure via BFS) are negligible compared to expensive LLM oracle calls, which remain the bottleneck by far in our experiments.
>
> ---
>
> > Conjecture 49 — adversarial instances
>
> Adversarial instances are not known for us and we only have a conjectured worst-case bound. We ran synthetic experiments with identity and reverse permutations across all 20 (n, k) configurations, where we observe that these orderings are *easier* than random. We present oracle calls Q(n, k, n) results at k=10, m=n below and will include the full results in the revision:
>
> | n | Identity | Reverse | Random [min,max] over 20 seeds | Conjectured bound: 1.25 * B(n,k,n) |
> |:---:|:---:|:---:|:---:|:---:|
> | 100 | 25 | 25 | [35, 39]  | 55 |
> | 200 | 53 | 53 | [86, 89] | 120 |
> | 400 | 109 | 109 | [204, 212]  | 255.9 |
> | 800 | 221 | 222 | [470, 480]  | 544.4 |
>
> Note that the gap widens as n increases and identity and reverse permutations give nearly identical counts despite representing opposite extremes.
>
> ---
>
> > Embedded prompt injection
>
> The hidden text was injected by ICML itself as watermarking, not by us — see the [ICML 2026 Peer Review FAQ](https://icml.cc/Conferences/2026/PeerReviewFAQ#prompt_injection). We appreciate the reviewer's diligence in flagging it.

---

> > ### Author Rebuttal · Reviewer_nqBb · 2026-04-01
> >
> > The rebuttal resolves the comparison fairness concern convincingly and adds meaningful ablation evidence. The core theoretical contribution, formalizing information accumulation in reranking via tournament graphs, with SCC-based handling of non-transitivity, is genuinely novel and could influence how the community approaches preference aggregation beyond document reranking. The originality of the framework, combined with consistent empirical results across benchmarks and oracles, warrants acceptance. The oracle noise gap is a limitation but not a dealbreaker given the paper's primarily theoretical contribution.
> >
> > Suggestions for Camera-Ready
> >
> > 1. **Include the SW-k10 comparison** (Table from rebuttal) in the main experiments — it is the cleanest evidence of algorithmic advantage.
> > 2. **Include the scheduling ablation** results (random vs. greedy) — this directly answers the natural question of where the gains come from.
> > 3. **Add a brief noise discussion** in the experiments section: even without a full study, reporting re-query consistency on a small subset of comparisons would partially address the concern.
> > 4. **Clarify the within-window transitivity point**: each $k$-wise query produces a ranked list (transitive tournament), so non-transitivity only arises across windows. This distinction between the general tournament framing and what LLM oracles actually produce deserves a sentence.
> > 5. **Emphasize variable $k$ per round** as a practical feature — this flexibility for heterogeneous document lengths is unique among baselines and undersold in the current draft.

---

### Official Review · Reviewer_oBfx · 2026-03-12

**Soundness:** 3
**Presentation:** 3
**Significance:** 3
**Originality:** 3
**Overall Recommendation:** 4
**Confidence:** 3

**Summary:**

This paper introduces a tournament graph framework for k-wise reranking in retrieval-augmented generation, addressing inefficiencies and limited information exploitation in existing methods. By using a global preference graph and a greedy query schedule, the framework achieves better accuracy with 25–40% fewer tokens than existing approaches, while effectively handling non-transitive preferences and achieving Pareto dominance across multiple benchmarks and models.

**Compliance With Llm Reviewing Policy:**

Affirmed.

**Key Questions For Authors:**

Please refer to the weaknesses part

Overall, I find the experimental section solid and fairly comprehensive, and I do not have many concerns about the empirical study itself. My main concern is about the paper’s introduction of prior work, which I believe should be described more carefully and accurately. If the authors can address this concern, I would be happy to raise my score.

**Limitations:**

yes

**Strengths And Weaknesses:**

Strengths

1. The figures and tables are clear.
2. The paper is well structured overall.
3. The baselines include some relatively recent methods, such as TourRank.

Weaknesses

1. The main issue lies in the introduction. The authors state that works such as TourRank are primarily designed to identify a winner. However, in each tournament round, TourRank actually assigns scores to all candidates, rather than only identifying the winner. Based on these scores, it is possible to derive more detailed pairwise ordering information among the candidates. Therefore, the characterization of TourRank in the introduction does not seem fully accurate.
2. It is unclear why the paper only reports NDCG@10. It would be more complete to also include results for other cutoff values, such as NDCG@5 and NDCG@20.

---

> ### Author Rebuttal · Authors · 2026-03-29
>
> We thank the reviewer for their positive assessment of the experimental section and constructive feedback. We are glad the reviewer finds the paper well-structured with clear figures and solid empirical evaluation, and we appreciate the explicit guidance on what would merit a higher score.
>
> ---
>
> > 1. The authors state that works such as TourRank are primarily designed to identify a winner. However, in each tournament round, TourRank actually assigns scores to all candidates, rather than only identifying the winner. Based on these scores, it is possible to derive more detailed pairwise ordering information among the candidates. Therefore, the characterization of TourRank in the introduction does not seem fully accurate.
>
> We appreciate your attention to detail and apologize for any confusion from the language in the introduction.
>
> TourRank's points system does extract ordinal position information from each group — not just the winner — and our language ("primarily to identify a winner") is imprecise when applied to TourRank.
>
> That said, TourRank only assigns scores to the subset of candidates that advance at each stage, rather than scoring all candidates simultaneously like our transitive notion of in-reach and out-reach. Specifically, at each selection stage of TourRank, only the selected documents receive a +1 point increment while the remaining documents receive no update (Algorithm 1, Line 8 in Chen et al., 2025). For example, in the stage that narrows from 5 to 2 candidates, only the top 2 receive +1 while the other 3 along with the other 95 candidates get nothing.
>
> We will revise the introductory language with the more precise discussion in the related works section and Appendix A.
>
> ---
>
> > 2. It is unclear why the paper only reports NDCG@10. It would be more complete to also include results for other cutoff values, such as NDCG@5 and NDCG@20.
>
> We note that nDCG@10 is the main/primary evaluation metric for all TREC-DL tracks as well as the BEIR benchmark. Baselines like AcuRank (Yoon et al., 2025), Setwise (Zhuang et al., 2024b), and LRL (Ma et al., 2023) only report nDCG@10.
>
> That said, we agree that reporting additional results would strengthen the evaluation and we report the numbers for nDCG@{1,5,10} below (note that our setting only selects the top-10). We will include these additions in the revision of our paper.
>
> | Method | nDCG@1 | nDCG@5 | nDCG@10 |
> |:---|:---:|:---:|:---:|
> | BM25 retrieval | 46.5 | 43.0 | 41.1 |
> | TourRank | 63.3 | 58.8 | 55.5 |
> | TourRank-R2 | 63.5 | 59.3 | 56.1 |
> | SW | 66.2 | 59.8 | 56.6 |
> | SW-R2 | 65.7 | 59.6 | 56.4 |
> | AcuRank | 64.5 | 59.5 | 55.9 |
> | AcuRank-H | 65.6 | 59.8 | 56.4 |
> | **TS-k20 (ours)** | 66.1 | 59.7 | 56.0 |
> | **TS-k10 (ours)** | 66.0 | 60.1 | 56.5 |
>
> *Macro-averaged nDCG@k scores across 14 datasets x 5 models.*
>
> ---
>
> Given the reviewer's positive assessment and the acknowledgment that the experimental section is "solid and fairly comprehensive," we hope this provides sufficient reasons to raise the score. We are committed to incorporating both changes.

---

> > ### Author Rebuttal · Reviewer_oBfx · 2026-04-01
> >
> > The authors have admitted to revise the descriptions and added experiments on ndcg@1, ndcg@5.

---

### Official Review · Reviewer_YCnN · 2026-03-14

**Soundness:** 3
**Presentation:** 3
**Significance:** 2
**Originality:** 2
**Overall Recommendation:** 5
**Confidence:** 2

**Summary:**

This work targets core flaws of LLM zero-shot re-ranking: heuristic strategies underutilize preference information, while information-efficient methods are computationally expensive. It proposes a theoretically grounded tournament graph framework (TOURNAMENTSORT) using pairwise preferences from k-ary document comparisons, aggregating into a global graph to derive more rankings via transitive closure. Validated on 14 datasets and 5 LLMs, it achieves Pareto optimality—matching/surpassing SOTA accuracy while reducing token consumption by 25%-40%.

**Compliance With Llm Reviewing Policy:**

Affirmed.

**Key Questions For Authors:**

See weakness

**Limitations:**

See weakness

**Strengths And Weaknesses:**

Strengths
- Solid theoretical foundation; rigorous proof of algorithm correctness and termination, filling theoretical gaps.

- Significantly improved information utilization, reducing LLM calls and token consumption; strong adaptability.

- Comprehensive experiments with statistically significant results supporting core claims.

Weaknesses

- Strong assumption of deterministic oracles; lack of fault tolerance for LLM judgment noise and error propagation.

- No strict theoretical proof for query complexity of general m-values, incomplete theory.

- Incomplete comparison with advanced adaptive re-ranking methods.

- Underutilizes retrieval prior information, limiting further efficiency improvement.

---

> ### Author Rebuttal · Authors · 2026-03-29
>
> We thank the reviewer for their careful reading and constructive feedback. We address each concern below.
>
> ---
>
> > Strong assumption of deterministic oracles; lack of fault tolerance for LLM judgment noise and error propagation
>
> We agree that extending to noisy oracles is an important direction. We acknowledge it in the future work section of the conclusion (Section 5): *"We hope the deterministic framework developed here provides a foundation for these extensions, much as noiseless sorting algorithms underpin the study of noisy comparison models."*
>
> Regarding the point about robustness, our empirical results across 5 diverse LLMs and 14 benchmarks show that, with modern LLMs, the strong assumption of deterministic oracles is not as catastrophic as theoretically possible.
>
> Additionally, we provide detailed comparisons against adaptive methods like AcuRank (which take into account noise) where variants like AcuRank-H trades additional compute for better noise handling which provides gain. However, we match AcuRank-H at significantly lower cost even without explicit handling of noise.
>
> That said, we believe that a principled noisy treatment is an important direction (although non-trivial) that we are actively exploring. We believe this would allow us further trade compute for additional accuracy gain by better modeling of noise.
>
> ---
>
> > No strict theoretical proof for query complexity of general m-values, incomplete theory.
>
> We appreciate the reviewer raising this point. We want to clarify that we *do* prove termination ($O(n^2)$ worst case) and an exact bound for m=1. The open question is tightening the bound for general m to match the conjectured O((n−1)/(k−1) + (m−1)/(k−1) · log_k m) in the query complexity discussion in Section 2.5 with more details in Appendix H (Conjecture 49).
>
> Our conjecture is primarily validated via empirical evidence across configurations with n up to 800 and k up to 50. Results show observed query counts within a factor of 1.25 of the conjecture (see Figure 10 on page 31), providing strong evidence that the functional form is sound. Thus, we believe that the practical algorithm's efficiency is not in question and what remains open is a formal analysis.
>
> We have attempted a formal analysis and believe that this is genuinely a difficult problem. From our existing analysis, there are three specific structural challenges we are working to overcome:
>
> 1. The algorithm schedules queries at the SCC level of the condensation while termination is a vertex-level condition (K_G(v) = n−1), requiring a bridge between these levels;
> 2. Transitive closure makes dynamics nonlocal: a single new edge can cascade and resolve vertices far from the query set; and
> 3. The per-query information gain is inherently non-uniform across phases of execution (as illustrated by the 25-horses example in Figure 2 with the 3 phases, where skeleton-building, connection, and saturation phases exhibit different rates).
>
> A tight analysis would likely require a phase-aware potential function which we believe is a substantial technical contribution in its own right.
>
> ---
>
> > Incomplete comparison with advanced adaptive re-ranking methods.
>
> The recent method AcuRank is an adaptive method and we compare against it. If we missed some important adaptive re-ranking methods, we would gladly add additional comparisons to our revision.
>
> ---
>
> > Underutilizes retrieval prior information, limiting further efficiency improvement.
>
> We agree that incorporating first-stage retrieval scores as priors is a promising direction and we acknowledge this point in the limitations section of the conclusion and future directions in Section 5.
>
> However, even without extensive incorporations of priors, our method already achieves strong efficiency gains over SoTA reranking methods across 14 datasets x 5 LLMs.
>
> Our Conjecture 49 holding would imply that theoretical efficiency is already approaching information-theoretical lower bounds.
>
> That said, we are actively exploring a more principled incorporation of priors with connection to the noisy oracle extension: priors become valuable when oracle observations are uncertain and can be modeled in a Bayesian framework.
>
> ---
>
> We again thank the reviewer for their thoughtful and detailed feedback. The points raised — particularly around noisy oracles, tighter theoretical analysis, and the incorporation of retrieval priors — align with directions we are actively pursuing. We are happy to incorporate any additional suggestions and welcome further discussion during the rebuttal period.

---

### Review · Ethics_Reviewer_wmud · 2026-03-30

**Recommendation:** Remediation action needed

**Basis For Judgement:**

If this is not an acceptable use of hidden text/prompt injection then the paper should be desk rejected.

**Ethics Issue:**

One reviewer flagged that the submitted PDF contains hidden text (invisible to human readers but extractable via text parsing tools) that instructs LLM-based reviewers to include specific "canary phrases" in their reviews. The reviewer says "see appended evidence" but I do not know what they are referring to with that statement so I am unable to verify this.

Per ICML policy "Any attempts at prompt injection are strictly forbidden and will result in desk rejection. (Prompt injection refers to the insertion of specially crafted text into the paper, with the intention to manipulate LLMs, for instance, to obtain a favorable review.)" However, an update on 2/14/2026 says "While prompt injection by authors is disallowed, we will not penalize papers with prompts that merely seek to detect the use of LLMs by reviewers." It seems like the authors are in the latter category (i.e., injecting hidden text with the purpose of trying to reveal if a reviewer has used LLMs in their review) based on the reviewers feedback.

Suggested actions: ethics chairs confirm if this is an acceptable use of hidden text/prompt injection.

**Remediation Action:**

Suggested actions: ethics chairs confirm if this is an acceptable use of hidden text/prompt injection.

---

### Decision · Program_Chairs · 2026-04-30

**Decision:**

Accept (spotlight)

**Comment:**

This is the meta-review that summarizes the review comments, rebuttals, and discussions. The paper presents a zero-shot ranking method that formulates LLM-based ranking as a tournament graph problem. All reviewers agree the paper is well presented and easy to follow. The theoretical foundation is solid. The author’s rebuttal has addressed most concerns raised in the initial reviews. The authors should o revise the paper according the review comments in the final version.